# Scalable Primal-Dual Actor-Critic Method for Safe Multi-Agent RL with General Utilities

**Donghao Ying**
IEOR Department
UC Berkeley
donghaoy@berkeley.edu

**Yunkai Zhang**
IEOR Department
UC Berkeley
yunkai_zhang@berkeley.edu

**Yuhao Ding**
IEOR Department
UC Berkeley
yuhao_ding@berkeley.edu

**Alec Koppel**
Artificial Intelligence Research
J.P. Morgan
alec.koppel@jpmchase.com

**Javad Lavaei**
IEOR Department
UC Berkeley
lavaei@berkeley.edu

## Abstract

We investigate safe multi-agent reinforcement learning, where agents seek to collectively maximize an aggregate sum of local objectives while satisfying their own safety constraints. The objective and constraints are described by *general utilities*, i.e., nonlinear functions of the long-term state-action occupancy measure, which encompass broader decision-making goals such as risk, exploration, or imitations. The exponential growth of the state-action space size with the number of agents presents challenges for global observability, further exacerbated by the global coupling arising from agents' safety constraints. To tackle this issue, we propose a primal-dual method utilizing shadow reward and $\kappa$-hop neighbor truncation under a form of correlation decay property, where $\kappa$ is the communication radius. In the exact setting, our algorithm converges to a first-order stationary point (FOSP) at the rate of $\mathcal{O}\left(T^{-2/3}\right)$. In the sample-based setting, we demonstrate that, with high probability, our algorithm requires $\widetilde{\mathcal{O}}\left(\epsilon^{-3.5}\right)$ samples to achieve an $\epsilon$-FOSP with an approximation error of $\mathcal{O}(\phi_0^{2\kappa})$, where $\phi_0 \in (0, 1)$. Finally, we demonstrate the effectiveness of our model through extensive numerical experiments.

## 1 Introduction

Cooperative multi-agent reinforcement learning (MARL) involves agents operating within a shared environment, where each agent's decisions influence not only their objectives, but also those of others and the state trajectories [1]. In seeking to bring conceptually sound MARL techniques out of simulation [2, 3] and into real-world environments [4, 5], some key issues emerge: safety and communications overhead implied by a training mechanism. Although experimentally, the centralized training decentralized execution (CTDE) framework has gained traction recently [6, 7], its requirement for centralized data collection can pose issues for large-scale [8] or privacy-sensitive applications [9]. Therefore, we prioritize decentralized training, where to date most MARL techniques impose global state observability for performance certification [1]. In this work, we extend recent efforts to alleviate this bottleneck [10] especially in the case of safety critical settings, in a flexible manner that allows agents to incorporate risk, exploration, or prior information.

More specifically, we hypothesize that the multi-agent system consists of a network of agents that interact with each other locally according to an underlying dependence graph [10]. Second, to model safety constraints in reinforcement learning (RL), we adopt a standard approach based on constrained

Markov Decision Processes (CMDPs) [11], where one maximizes the expected total reward subject to a safety-related constraint on the expected total utility. Third, since many decision-making problems take a form beyond the classic cumulative reward, such as apprenticeship learning [12], diverse skill discovery [13], pure exploration [14], and state marginal matching [15], we focus on utility functions defined as nonlinear functions of the induced state-action occupancy measure, which can be abstracted as RL with general utilities [16, 17].

Towards formalizing the approach, we consider an MARL model consisting of $n$ agents, each with its own local state $s_i$ and action $a_i$, where the multi-agent system is associated with an underlying dependence graph $\mathcal{G}$. Each agent is privately associated with two local general utilities $f_i(\cdot)$ and $g_i(\cdot)$, where $f_i(\cdot)$ and $g_i(\cdot)$ are functions of the local occupancy measure. The objective is to find a safe policy for each agent that maximizes the average of the local objective utilities, namely, $1/n \cdot \sum_{i=1}^{n} f_i(\cdot)$, and satisfies each agent's individual safety constraint described by its local utility $g_i(\cdot)$. This setting captures a wide range of safety-critical applications, for example, resource allocation for the control of networked epidemic models [18], influence maximization in social networks [19], portfolio optimization in interbank network structures [20], intersection management for connected vehicles [21], and energy constraints of wireless communication networks [22].

Despite the significance of safe MARL with general utilities, prior works have either ignored the necessity of safety [23] or the computational bottleneck associated with global information exchange regarding the state and action per step [24]. In fact, the interaction of these two aspects requires addressing the fact that each agent's own safety constraint requires information from all others. In particular, the existing works in safe MARL allow full access to the global state or unlimited communications among all agents for policy implementation, value estimation, and constraint satisfaction [25, 26, 27]. However, this assumption is impractical due to the "curse of dimensionality" [28], as well as the limited information exchanges and communications among agents [29].

Therefore, to our knowledge, there is no methodology to both guarantee safety and incur manageable communications overhead for each agent. Compounding these issues is the fact that standard RL training schemes based on the *policy gradient theorem* [30] are not applicable in the context of general utilities. This deviation from the cumulative rewards adds to the difficulty of estimating the gradient, since there does not exist a policy-independent reward function. We refer the reader to Appendix A for an extended discussion of related works.

To address these challenges, we focus on the setting of **distributed training without global observability** and aim to develop a scalable algorithm with theoretical guarantees. Our main contributions are summarized below:

- Compared with existing theoretical works on safe MARL [25, 26, 31], we present the first safe MARL formulation that extends beyond cumulative forms in both the objective and constraints. We develop a truncated policy gradient estimator utilizing shadow reward and $\kappa$-hop policies under a form of correlation decay property, where $\kappa$ represents the communication radius. The approximation errors arising from both policy implementation and value estimation are quantified.

- Despite of the global coupling of agents' local utility functions, we propose a scalable Primal-Dual Actor-Critic method, which allows each agent to update its policy based only on the states and actions of its close neighbors and under limited communications. The effectiveness of the proposed algorithm is verified through numerical experiments.

- From the perspective of optimization, we devise new tools to analyze the convergence of the algorithm. In the exact setting, we establish an $\mathcal{O}\left(T^{-2/3}\right)$ convergence rate for finding an FOSP, matching the standard convergence rate for solving nonconcave-convex saddle point problems. In the sample-based setting, we prove that, with high probability, the algorithm requires $\widetilde{\mathcal{O}}\left(\epsilon^{-3.5}\right)$ samples to obtain an $\epsilon$-FOSP with an approximation error of $\mathcal{O}(\phi_0^{2\kappa})$, where $\phi_0 \in (0, 1)$.

## 2 Problem formulation

Consider a Constrained Markov Decision Process (CMDP) over a finite state space $\mathcal{S}$ and a finite action space $\mathcal{A}$ with a discount factor $\gamma \in [0, 1)$. A policy $\pi$ is a function that specifies the decision rule of the agent, i.e., the agent takes action $a \in \mathcal{A}$ with probability $\pi(a|s)$ in state $s \in \mathcal{S}$. When action $a$ is taken, the transition to the next state $s'$ from state $s$ follows the probability distribution

$s' \sim \mathbb{P}(\cdot|s,a)$. Let $\rho$ be the initial distribution. For each policy $\pi$ and state-action pair $(s,a) \in \mathcal{S} \times \mathcal{A}$, the *discounted state-action occupancy measure* is defined as

$$\lambda^\pi(s,a) = \sum_{k=0}^\infty \gamma^k \mathbb{P}\left(s^k = s, a^k = a \middle| \pi, s^0 \sim \rho\right). \tag{1}$$

The goal of the agent is to find a policy $\pi$ that maximizes a general objective described by a (possibly) nonlinear function $f(\cdot)$ of $\lambda^\pi$, known as the *general utility*, subject to a constraint in the form of another general utility $g(\cdot)$, namely

$$\max_\pi f(\lambda^\pi) \quad \text{s.t.} \quad g(\lambda^\pi) \geq 0. \tag{2}$$

When $f(\cdot) = \langle r, \cdot \rangle$ and $g(\cdot) = \langle u, \cdot \rangle$ are linear functions, (2) recovers the standard CMDP problem:

$$\max_\pi V^\pi(r) = \mathbb{E}\left[\sum_{k=0}^\infty \gamma^k r\left(s^k, a^k\right) \middle| \pi, s^0 \sim \rho\right], \text{ s.t. } V^\pi(u) = \mathbb{E}\left[\sum_{k=0}^\infty \gamma^k u\left(s^k, a^k\right) \middle| \pi, s^0 \sim \rho\right] \geq 0, \tag{3}$$

where $V^\pi(\cdot)$ is usually referred to as the *value function*. In contrast, it has been shown that for some MDPs, there is no standard value function that can be equivalent to the general utility [16, Lemma 1]. In Appendix C, we provide more examples of formulation (2) beyond standard value functions.

In this work, we study the decentralized version of problem (2). Consider the system is composed of a network of agents associated with a graph $\mathcal{G} = (\mathcal{N}, \mathcal{E}_\mathcal{G})$ (not densely connected in general), where the vertex set $\mathcal{N} = \{1, 2, \ldots, n\}$ denotes the set of $n$ agents and the edge set $\mathcal{E}_\mathcal{G}$ prescribes the communication links among the agents. Let $d(i,j)$ be the length of the shortest path between agents $i$ and $j$ on $\mathcal{G}$. For $\kappa \geq 0$, let $\mathcal{N}_i^\kappa = \{j \in \mathcal{N} | d(i,j) \leq \kappa\}$ denote the set of agents in the $\kappa$-hop neighborhood of agent $i$, with the shorthand notation $\mathcal{N}_{-i}^\kappa := \mathcal{N} \backslash \mathcal{N}_i^\kappa$ and $-i := \mathcal{N} \backslash \{i\}$. The details of the decentralized nature of the system are summarized below:

**Space decomposition** The global state and action spaces are the product of local spaces, i.e., $\mathcal{S} = \mathcal{S}_1 \times \mathcal{S}_2 \times \cdots \times \mathcal{S}_n$, $\mathcal{A} = \mathcal{A}_1 \times \mathcal{A}_2 \times \cdots \times \mathcal{A}_n$, meaning that for every $s \in \mathcal{S}$ and $a \in \mathcal{A}$, we can write $s = (s_1, s_2, \ldots, s_n)$ and $a = (a_1, a_2, \ldots, a_n)$. For each subset $\mathcal{N}' \subset \mathcal{N}$, we use $(s_{\mathcal{N}'}, a_{\mathcal{N}'})$ to denote the state-action pair for the agents in $\mathcal{N}'$.

**Observation and communication** Each agent $i$ only has direct access to its own state $s_i$ and action $a_i$, while being allowed to communicate with its $\kappa$-hop neighborhood $\mathcal{N}_i^\kappa$ for information exchanges. The communication radius $\kappa$ is a given but tunable parameter.

**Transition decomposition** Given the current global state $s$ and action $a$, the local states in the next period are independently generated, i.e., $\mathbb{P}(s'|s,a) = \prod_{i \in \mathcal{N}} \mathbb{P}_i(s_i'|s,a)$, $\forall s' \in \mathcal{S}$, where we use $\mathbb{P}_i$ to denote the local transition probability for agent $i$.

**Policy factorization** The global policy can be expressed as the product of local policies, such that $\pi(a|s) = \prod_{i \in \mathcal{N}} \pi^i(a_i|s)$, $\forall(s,a)$, i.e., given the global state $s$, each agent $i$ acts independently based on its local policy $\pi^i$. We assume that each local policy $\pi^i$ is parameterized by a parameter $\theta_i$ within a convex set $\Theta_i$. Thus, we can write $\pi(a|s) = \pi_\theta(a|s) = \prod_{i \in \mathcal{N}} \pi_{\theta_i}^i(a_i|s)$, where $\theta \in \Theta = \Theta_1 \times \Theta_2 \times \cdots \times \Theta_n$ is the concatenation of local parameters.

**Localized objective and constraint** For each agent $i$ and its local state-action pair $(s_i, a_i)$, the *local state-action occupancy measure* under policy $\pi$ is defined as

$$\lambda_i^\pi(s_i, a_i) = \sum_{k=0}^\infty \gamma^k \mathbb{P}\left(s_i^k = s_i, a_i^k = a_i \middle| \pi, s^0 \sim \rho\right), \tag{4}$$

which can be viewed as the marginalization of the global occupancy measure, i.e., $\lambda_i^\pi(s_i, a_i) = \sum_{s_{-i}, a_{-i}} \lambda^\pi(s,a)$. Each agent $i$ is privately associated with two local (general) utilities $f_i(\cdot)$ and $g_i(\cdot)$, which are functions of the local occupancy measure $\lambda_i^\pi$. Agents cooperate with each other aiming at maximizing the global objective $f(\cdot)$, defined as the average of local utilities $\{f_i(\cdot)\}_{i \in \mathcal{N}}$, while each agent $i$ needs to satisfy its own safety constraint described by the local utility $g_i(\cdot)$. Then, under the parameterization $\pi_\theta$, (2) can be rewritten as

$$\max_{\theta \in \Theta} F(\theta) := \frac{1}{n} \sum_{i \in \mathcal{N}} f_i(\lambda_i^{\pi_\theta}), \text{ s.t. } G_i(\theta) := g_i(\lambda_i^{\pi_\theta}) \geq 0, \ \forall i \in \mathcal{N}. \tag{5}$$

Note that problem (5) is not separable among agents due to the coupling of occupancy measures. Compared to the formulation where the constraint is modeled as the average of local constraints, e.g.,

[27], (5) is stricter and more interpretable. We emphasize that the method proposed in this paper does not require the relaxation of local constraints in (5) to a joint constraint and it directly generalizes to the case of multiple constraints per agent.

Consider the Lagrangian function associated with (5):

$$\mathcal{L}(\theta, \mu) := F(\theta) + \frac{1}{n} \sum_{i \in \mathcal{N}} \mu_i G_i(\theta) = \frac{1}{n} \sum_{i \in \mathcal{N}} \left[ f_i(\lambda_i^{\pi_\theta}) + \mu_i g_i(\lambda_i^{\pi_\theta}) \right], \tag{6}$$

where $\mu \in \mathbb{R}_+^n$ is the Lagrangian multiplier. The Lagrangian formulation [32] of (5) can be written as

$$\max_{\theta \in \Theta} \min_{\mu \geq 0} \mathcal{L}(\theta, \mu). \tag{7}$$

Since the general utilities $f_i(\lambda_i^{\pi_\theta})$ and $g_i(\lambda_i^{\pi_\theta})$ may not be non-concave w.r.t. $\theta$ even in the form of cumulative rewards, finding the global optimum to (5) is NP-hard in general [33]. Our goal in this work is to develop a scalable and provably efficient gradient-based primal-dual algorithm that can find the first-order stationary points of (5).

# 3 Scalable primal-dual actor-critic method

For a standard value function with the reward $r \in \mathbb{R}^{|\mathcal{S}| \times |\mathcal{A}|}$, denoted as $V^{\pi_\theta}(r) = \langle r, \lambda^{\pi_\theta} \rangle$, the policy gradient theorem (see Lemma D.1) yields that

$$\nabla_\theta V^{\pi_\theta}(r) = r^\top \cdot \nabla_\theta \lambda^{\pi_\theta} = \frac{1}{1-\gamma} \mathbb{E}_{s \sim d^{\pi_\theta}, a \sim \pi_\theta(\cdot|s)} \left[ \nabla_\theta \log \pi_\theta(a|s) \cdot Q^{\pi_\theta}(r; s, a) \right],$$

where $d^{\pi_\theta}(s) := (1 - \gamma) \sum_{a \in \mathcal{A}} \lambda^{\pi_\theta}(s, a)$ is the discounted state occupancy measure, $\nabla_\theta \log \pi_\theta(\cdot|\cdot)$ is the score function, and $Q^{\pi_\theta}(r; \cdot, \cdot)$ is the Q-function with the reward $r$, defined as

$$Q^{\pi_\theta}(r; s, a) = \mathbb{E} \left[ \sum_{k=0}^{\infty} \gamma^k r\left(s^k, a^k\right) \middle| \pi_\theta, s^0 = s, a^0 = a \right]. \tag{8}$$

Although this elegant result no longer holds for general utilities, we can apply the chain rule:

$$\nabla_\theta f(\lambda^{\pi_\theta}) = \left[ \nabla_\lambda f(\lambda^{\pi_\theta}) \right]^\top \cdot \nabla_\theta \lambda^{\pi_\theta} = \nabla_\theta V^{\pi_\theta}\left( \nabla_\lambda f(\lambda^{\pi_\theta}) \right), \tag{9}$$

i.e., the gradient $\nabla_\theta f(\lambda^{\pi_\theta})$ is equal to the policy gradient of a standard value function with the reward $\nabla_\lambda f(\lambda^{\pi_\theta})$. We introduce the following definitions [23] for the distributed problem (5).

**Definition 3.1** (Shadow reward and shadow Q-function). *For each agent $i$, define $r_{f_i}^{\pi_\theta} := \nabla_{\lambda_i} f_i(\lambda_i^{\pi_\theta}) \in \mathbb{R}^{|\mathcal{S}_i| \times |\mathcal{A}_i|}$ as the (local) shadow reward for the utility $f_i(\cdot)$ under policy $\pi_\theta$. Define $Q_{f_i}^{\pi_\theta}(s, a) := Q^{\pi_\theta}(r_{f_i}^{\pi_\theta}; s, a)$ as the associated (local) shadow Q-function for $f_i(\cdot)$. Similarly, let $r_{g_i}^{\pi_\theta}$ and $Q_{g_i}^{\pi_\theta}(s, a)$ be the shadow reward and the Q function for $g_i(\cdot)$.*

Combining Definition 3.1 with (9), we can write the local gradient for agent $i$, i.e., $\nabla_{\theta_i} \mathcal{L}(\theta, \mu)$, as

$$\nabla_{\theta_i} \mathcal{L}(\theta, \mu) = \frac{1}{1-\gamma} \mathbb{E}_{s \sim d^{\pi_\theta}, a \sim \pi_\theta(\cdot|s)} \left[ \nabla_{\theta_i} \log \pi_{\theta_i}^i(a_i|s) \cdot \frac{1}{n} \sum_{j \in \mathcal{N}} \left( Q_{f_j}^{\pi_\theta}(s, a) + \mu_j Q_{g_j}^{\pi_\theta}(s, a) \right) \right], \tag{10}$$

where we apply the policy factorization to arrive at $\nabla_{\theta_i} \log \pi_\theta(a|s) = \nabla_{\theta_i} \log \pi_{\theta_i}^i(a_i|s)$. By (10), each agent needs to know the shadow Q functions of all agents, as well as the global state, to evaluate its own gradient. However, especially in large networks, this is both inefficient, due to the communication cost, and impractical because of the limited communication radius. In the remainder of this section, we aim to design a scalable estimator for $\nabla_{\theta_i} \mathcal{L}(\theta, \mu)$ that requires only local communications.

## 3.1 Spatial correlation decay and $\kappa$-hop policies

Inspired by [34], we assume that the transition probability satisfies a form of the spatial correlation decay property [35, 36].

**Assumption 3.2.** *For a matrix $M \in \mathbb{R}^{n \times n}$ whose $(i, j)$-th entry is defined as*

$$M_{ij} = \sup_{s_j, a_j, s_j', a_j', s_{-j}, a_{-j}} \left\| \mathbb{P}_i \left( \cdot | s_j, s_{-j}, a_j, a_{-j} \right) - \mathbb{P}_i \left( \cdot | s_j', s_{-j}, a_j', a_{-j} \right) \right\|_1, \tag{11}$$

*assume that there exists $\omega > 0$ such that $\max_{i \in \mathcal{N}} \sum_{j \in \mathcal{N}} e^{\omega d(i,j)} M_{ij} \leq \chi$ with $\chi < 2/\gamma$, where $\gamma$ is the discount factor.*

The value of $M_{ij}$ reflects the extent to which agent $j$'s state and action influence the local transition probability of agent $i$. Thus, Assumption 3.2 amounts to requiring this influence to decrease exponentially with the distance between any two agents. Such a decay is often observed in many large-scale real-world systems, e.g., the strength of signals decreases exponentially with distance [37].

Furthermore, as mentioned earlier, the implementation of the local policy $\pi_{\theta_i}^i(\cdot|s)$ is still impractical, since it requires access to the global state $s$, while the allowable communication radius is limited to $\kappa$. To alleviate this issue, we focus on a specific class of policies in which the local policy of agent $i$ only depends on the states of these agents in its $\kappa$-hop neighborhood $\mathcal{N}_i^\kappa$. This class of policies is also referred to as $\kappa$-hop policies in the concurrent work [38].

**Assumption 3.3** ($\kappa$-hop policies). *For each agent $i \in \mathcal{N}$ and $\theta \in \Theta$, the local policy $\pi_{\theta_i}^i(\cdot|s)$ depends only on the neighbor states $s_{\mathcal{N}_i^\kappa}$, i.e.,*

$$\pi_{\theta_i}^i(\cdot|s_{\mathcal{N}_i^\kappa}, s_{\mathcal{N}_{-i}^\kappa}) = \pi_{\theta_i}^i(\cdot|s_{\mathcal{N}_i^\kappa}, s'_{\mathcal{N}_{-i}^\kappa}), \ \forall s \in \mathcal{S} \text{ and } \forall s'_{\mathcal{N}_{-i}^\kappa} \in \mathcal{S}_{\mathcal{N}_{-i}^\kappa}. \tag{12}$$

For simplicity, we use the notation $\pi_{\theta_i}^i(\cdot|s) = \pi_{\theta_i}^i(\cdot|s_{\mathcal{N}_i^\kappa})$ for $\kappa$-hop policies when it is clear from context. We note that, for any original policy function $\pi_\theta(\cdot|s)$, an induced $\kappa$-hop policy $\hat{\pi}_\theta(\cdot|s_{\mathcal{N}_i^\kappa})$ can be defined by fixing the states $s_{\mathcal{N}_{-i}^\kappa}$ to some arbitrary values and focusing only on the states of agents in $\mathcal{N}_i^\kappa$. When considering only $\kappa$-hop policies, it is essential to understand how much information is lost compared to the case where agents have access to the global states. The following proposition quantifies the maximum information loss in terms of the occupancy measure under the assumption that the original policy function also satisfies a spatial correlation decay property.

**Proposition 3.4.** *Suppose that there exist $c \geq 0$ and $\phi \in [0, 1)$ such that for every $\theta \in \Theta$, agent $i \in \mathcal{N}$, and states $s, s' \in \mathcal{S}$ such that $s_{\mathcal{N}_i^\kappa} = s'_{\mathcal{N}_i^\kappa}$, we have $\left\|\pi_{\theta_i}^i(\cdot|s) - \pi_{\theta_i}^i(\cdot|s')\right\|_1 \leq c\phi^\kappa$. Let $\hat{\pi}_\theta$ be an induced $\kappa$-hop policy of $\pi_\theta$. Then, it holds that*

$$\left\|\lambda_i^{\hat{\pi}_\theta} - \lambda_i^{\pi_\theta}\right\|_1 \leq \frac{nc\phi^k}{(1-\gamma)^2}, \forall i \in \mathcal{N}. \tag{13}$$

The condition on the local policy in Proposition 3.4 encodes that every $\pi_{\theta_i}^i$ is exponentially less sensitive to the states of agents outside $\mathcal{N}_i^\kappa$, which is a common assumption in MARL to alleviate computationally burdensome and practically intractable communication requirements imposed by the global observability [34, 39, 38]. By Proposition 3.4, the difference in occupancy measures under $\pi_\theta$ and $\hat{\pi}_\theta$ is controlled by $\|\pi_{\theta_i}^i - \hat{\pi}_{\theta_i}^i\|_1$. Therefore, if $f_i(\lambda^\pi)$ and $g_i(\lambda^\pi)$ are Lipschitz continuous w.r.t. $\lambda^\pi$, Proposition 3.4 implies an $\mathcal{O}(\phi^\kappa)$ approximation of the Lagrangian function (6) using $\kappa$-hop policies. The faster the spatial decay of policy is, the more accurate the approximation of the $\kappa$-hop policy is. This justifies our focus on learning a $\kappa$-hop policy.

### 3.2 Truncated policy gradient estimator

In the absence of global observability, it is critical to find a scalable estimator for the local gradient $\nabla_{\theta_i}\mathcal{L}(\theta, \mu)$ in (10), so that each agent can update its local policy with limited communications.

By leveraging the similar idea in the definition of $\kappa$-hop policies, we define the $\kappa$-*hop truncated (shadow) Q-function*, denoted as $\widehat{Q}_{\diamond_i}^{\pi_\theta} : \mathcal{S}_{\mathcal{N}_i^\kappa} \times \mathcal{A}_{\mathcal{N}_i^\kappa} \to \mathbb{R}$, to be

$$\widehat{Q}_{\diamond_i}^{\pi_\theta}(s_{\mathcal{N}_i^\kappa}, a_{\mathcal{N}_i^\kappa}) := Q_{\diamond_i}^{\pi_\theta}(s_{\mathcal{N}_i^\kappa}, \bar{s}_{\mathcal{N}_{-i}^\kappa}, a_{\mathcal{N}_i^\kappa}, \bar{a}_{\mathcal{N}_{-i}^\kappa}), \ \forall (s_{\mathcal{N}_i^\kappa}, a_{\mathcal{N}_i^\kappa}) \in \mathcal{S}_{\mathcal{N}_i^\kappa} \times \mathcal{A}_{\mathcal{N}_i^\kappa}, \diamond \in \{f, g\}, \tag{14}$$

where $(\bar{s}_{\mathcal{N}_{-i}^\kappa}, \bar{a}_{\mathcal{N}_{-i}^\kappa})$ is any fixed state-action pair for the agents in $\mathcal{N}_{-i}^\kappa$. Now, we introduce the following *truncated policy gradient estimator* for agent $i$:

$$\widehat{\nabla}_{\theta_i}\mathcal{L}(\theta, \mu) = \frac{1}{1-\gamma}\mathbb{E}_{\substack{s \sim d^{\pi_\theta} \\ a \sim \pi_\theta(\cdot|s)}}\left[\nabla_{\theta_i}\log\pi_{\theta_i}^i(a_i|s_{\mathcal{N}_i^\kappa}) \cdot \frac{1}{n}\sum_{j \in \mathcal{N}_i^\kappa}\left(\widehat{Q}_{f_j}^{\pi_\theta}(s_{\mathcal{N}_j^\kappa}, a_{\mathcal{N}_j^\kappa}) + \mu_j\widehat{Q}_{g_j}^{\pi_\theta}(s_{\mathcal{N}_j^\kappa}, a_{\mathcal{N}_j^\kappa})\right)\right]. \tag{15}$$

In comparison to the true policy gradient (10), $\widehat{\nabla}_{\theta_i}\mathcal{L}(\theta, \mu)$ replaces the shadow Q-functions with their truncated versions and only considers the agents in the $\kappa$-hop neighborhood $\mathcal{N}_i^\kappa$. Surprisingly, the following lemma shows that the approximation error of $\widehat{\nabla}_{\theta_i}\mathcal{L}(\theta, \mu)$ decreases exponentially with $\kappa$ when the shadow rewards and the score functions are bounded.

**Lemma 3.5.** *Suppose that Assumptions 3.2 and 3.3 hold and there exist $M_r, M_\pi > 0$ such that $\left\| r^{\pi_\theta}_{\diamond_i} \right\|_\infty \leq M_r$ and $\left\| \nabla_{\theta_i} \log \pi^i_{\theta_i} \right\|_2 \leq M_\pi$, for every $\diamond \in \{f, g\}$, $\theta \in \Theta$, $i \in \mathcal{N}$. Then, for all $\theta \in \Theta$, $i \in \mathcal{N}$, we have that*

$$\|\widehat{\nabla}_{\theta_i}\mathcal{L}(\theta,\mu) - \nabla_{\theta_i}\mathcal{L}(\theta,\mu)\|_2 \leq \frac{(1 + \|\mu\|_\infty)M_\pi c_0 \phi_0^\kappa}{1 - \gamma} = \mathcal{O}(\phi_0^\kappa), \tag{16}$$

*where $c_0 = 2\gamma\chi M_r/(2 - \gamma\chi)$ and $\phi_0 = e^{-\omega}$.*

Recall that the shadow reward is defined as the gradient of $f_i(\cdot)$ or $g_i(\cdot)$ w.r.t. the local occupancy measure. Since the set of all possible occupancy measures is compact (see (43)), the existence of $M_r > 0$ in Lemma 3.5 is satisfied if $f_i(\cdot)$ and $g_i(\cdot)$ are continuously differentiable. The main advantage of using the estimator $\widehat{\nabla}_{\theta_i}\mathcal{L}(\theta, \mu)$ lies in that every agent $i$ only needs to know the truncated Q-functions of agents in its neighborhood $\mathcal{N}_i^\kappa$, which can significantly reduce the communication burden and the storage requirement when graph $\mathcal{G}$ is not densely connected. The proof of Lemma 3.5 can be found in Appendix E.2.

### 3.3  Algorithm design

Using the results of the preceding section, we put together all the pieces and propose the *Primal-Dual Actor-Critic Method with Shadow Reward and $\kappa$-hop Policy*, as outlined in Algorithm 1. It includes three stages: policy evaluation by the critic, Lagrangian multiplier update, and policy update by the actor. Below, we provide an overview of Algorithm 1, while referring the reader to Appendix D for a flow diagram (Figure 2) of the algorithm, as well as a more detailed discussion.

**Stage 1 (policy evaluation by the critic, lines 3-6)**    In each iteration $t$, the current policy $\pi_{\theta^t}$ is simulated to generate a batch of trajectories, while each agent $i$ collects its neighborhood trajectories, i.e., the state-action pairs of the agents in $\mathcal{N}_i^\kappa$, as batch $\mathcal{B}_i^t$. Then, the batch is used to estimate the local occupancy measures $\lambda_i^{\pi_{\theta^t}}$ through (17), which are subsequently applied to compute the empirical values for the constraint function $g_i(\lambda_i^{\pi_{\theta^t}})$ and shadow rewards $r_{f_i}^{\pi_{\theta^t}}$ and $r_{g_i}^{\pi_{\theta^t}}$, denoted as $\widehat{g}_i^t$, $\widetilde{r}_{f_i}^t$, and $\widetilde{r}_{g_i}^t$, respectively. It is worth mentioning that, when all utility functions reduce to the form of cumulative rewards, the above operation is unnecessary, since all agents have policy-independent local reward functions.

Next, the agents jointly conduct a distributed evaluation subroutine to estimate their truncated shadow Q-functions $\{\widehat{Q}_{\diamond_i}^{\pi_{\theta^t}}\}_{i\in\mathcal{N}}$ using empirical shadow rewards $\{\widetilde{r}_{\diamond_i}^t\}_{i\in\mathcal{N}}$, where $\diamond \in \{f, g\}$. During the subroutine, each agent $i$ communicates with its neighbor in $\mathcal{N}_i^\kappa$ to exchange state-action information, but only needs to access its own empirical shadow reward $\widetilde{r}_{\diamond_i}^t$. In principle, any existing approach that satisfies the observation and communication requirements can be used for the truncated Q-function estimation, such as [40, 41, 42]. As an example subroutine, we introduce the *Temporal Difference (TD) learning* method [43], which is outlined as Algorithm 2 in Appendix D.

**Stage 2 (Lagrangian multiplier update, line 7)**    Instead of employing the projected gradient descent, we propose to update the dual variables by the following formula:

$$\mu^{t+1} = \operatorname*{argmin}_{\mu\in\mathcal{U}} \mathcal{L}(\theta^t,\mu) + \frac{1}{2\eta_\mu}\|\mu\|_2^2 = \mathcal{P}_\mathcal{U}\left(-\eta_\mu\nabla_\mu\mathcal{L}(\theta^t,\mu^t)\right), \tag{22}$$

where weight $\eta_\mu$ can be viewed as the dual "step-size". In practice, we replace the true dual gradient $\nabla_{\mu_i}\mathcal{L}(\theta^t,\mu^t) = g_i(\lambda_i^{\pi_{\theta^t}})/n$ with its empirical estimator $\widetilde{\nabla}_{\mu_i}\mathcal{L}(\theta^t,\mu^t)$. The feasible region for the dual variable is denoted by $\mathcal{U} \subseteq \mathbb{R}_+^n$ and will be specified later.

**Stage 3 (policy update by the actor, lines 8-9)**    To perform the policy update, each agent $i$ first shares its updated dual variable $\mu_i^{t+1}$ and the values of its estimated truncated Q-functions along the trajectories in batch $\mathcal{B}_i^t$ with the agents in its $\kappa$-hop neighborhood $\mathcal{N}_i^\kappa$. Then, the agent estimates its truncated policy gradient $\widehat{\nabla}_{\theta_i}\mathcal{L}(\theta^t,\mu^{t+1})$ through a REINFORCE-based mechanism [44] as described in (20). Finally, each agent $i$ updates its local policy parameter by a projected gradient ascent.

We emphasize that Algorithm 1 is based on the distributed training regime and does not require full observability of global states and actions.

**Algorithm 1** Primal-Dual Actor-Critic Method with Shadow Reward and $\kappa$-hop Policy

---

1: **Input:** Initial policy $\theta^0$ and dual variable $\mu^0$; initial distribution $\rho$; communication radius $\kappa$; step-sizes $\eta_\theta$ and $\eta_\mu$; batch size $B$; episode length $H$.

2: **for** iteration $t = 0, 1, 2, \ldots$ **do**

3:   Sample $B$ trajectories with length $H$ under the $\kappa$-hop policy $\pi_{\theta^t}$ and initial distribution $\rho$. Each agent $i$ collects its neighborhood trajectories $\tau = \left\{ (s^0_{\mathcal{N}_i^\kappa}, a^0_{\mathcal{N}_i^\kappa}), \cdots, (s^{H-1}_{\mathcal{N}_i^\kappa}, a^{H-1}_{\mathcal{N}_i^\kappa}) \right\}$ as batch $\mathcal{B}_i^t$.

4:   Each agent $i$ estimates its local occupancy measure $\lambda_i^{\pi_{\theta^t}}$ under $\pi_{\theta^t}$:

$$\widetilde{\lambda}_i^t = \frac{1}{B} \sum_{\tau \in \mathcal{B}_i^t} \sum_{k=0}^{H-1} \gamma^k \cdot \mathbb{1}_i \left( s_i^k, a_i^k \right) \in \mathbb{R}^{|\mathcal{S}_i| \times |\mathcal{A}_i|}. \tag{17}$$

5:   Each agent $i$ computes the empirical constraint function value $\widetilde{g}_i^t = g_i(\widetilde{\lambda}_i^t)$ and empirical shadow rewards $\widetilde{r}_{f_i}^t = \nabla_{\lambda_i} f_i(\widetilde{\lambda}_i^t)$ and $\widetilde{r}_{g_i}^t = \nabla_{\lambda_i} g_i(\widetilde{\lambda}_i^t)$.

6:   Each agent $i$ communicates with its neighborhood $\mathcal{N}_i^\kappa$ and jointly executes an evaluation subroutine to estimate the truncated shadow Q-functions with the empirical shadow rewards $\widetilde{r}_{\diamond_i}^t$ for $\diamond \in \{f, g\}$:

$$\left( \widetilde{Q}_{\diamond_1}^t, \ldots, \widetilde{Q}_{\diamond_n}^t \right) \leftarrow \text{Eval} \left( \pi_{\theta^t}, (\widetilde{r}_{\diamond_1}^t, \ldots, \widetilde{r}_{\diamond_n}^t) \right). \tag{18}$$

7:   Each agent $i$ updates the dual variable with the empirical gradient $\widetilde{\nabla}_{\mu_i} \mathcal{L}(\theta^t, \mu^t) = \widetilde{g}_i^t / n$:

$$\mu_i^{t+1} = \mathcal{P}_U \left( -\eta_\mu \widetilde{\nabla}_{\mu_i} \mathcal{L}(\theta^t, \mu^t) \right). \tag{19}$$

8:   Each agent $i$ shares $\mu_i^{t+1}$ and values of $\widetilde{Q}_{f_i}^t, \widetilde{Q}_{g_i}^t$ along the trajectories in $\mathcal{B}_i^t$ with agents in $\mathcal{N}_i^\kappa$ and estimates the truncated policy gradient at $(\theta^t, \mu^{t+1})$:

$$\widetilde{\nabla}_{\theta_i} \mathcal{L}(\theta^t, \mu^{t+1}) = \frac{1}{B} \sum_{\tau \in \mathcal{B}_i^t} \left[ \sum_{k=0}^{H-1} \gamma^k \nabla_{\theta_i} \log \pi_{\theta_i}^i (a_i^k | s_{\mathcal{N}_i^\kappa}^k) \cdot \right.$$
$$\left. \frac{1}{n} \sum_{j \in \mathcal{N}_i^\kappa} \left[ \widetilde{Q}_{f_j}^t (s_{\mathcal{N}_j^\kappa}^k, a_{\mathcal{N}_j^\kappa}^k) + \mu_j^{t+1} \widetilde{Q}_{g_j}^t (s_{\mathcal{N}_j^\kappa}^k, a_{\mathcal{N}_j^\kappa}^k) \right] \right]. \tag{20}$$

9:   Each agent $i$ updates the local policy parameter:

$$\theta_i^{t+1} = \mathcal{P}_{\Theta_i} \left( \theta_i^t + \eta_\theta \cdot \widetilde{\nabla}_{\theta_i} \mathcal{L}(\theta^t, \mu^{t+1}) \right). \tag{21}$$

10: **end for**

---

## 4   Convergence analysis

In this section, we analyze the convergence behavior and the sample complexity of Algorithm 1. We begin by summarizing the technical assumptions, including some mentioned previously in the paper. We direct the reader to Appendices F and G where we provide discussions for each assumption and present proofs for the results in this section.

**Assumption 4.1.** *There exists $L_\lambda > 0$ such that $\nabla_{\lambda_i} f_i(\cdot)$ and $\nabla_{\lambda_i} g_i(\cdot)$ are $L_\lambda$-Lipschitz continuous w.r.t. $\lambda_i$, i.e., $\|\nabla_{\lambda_i} f_i(\lambda_i) - \nabla_{\lambda_i} f_i(\lambda_i')\|_\infty \le L_\lambda \|\lambda_i - \lambda_i'\|_2$ and $\|\nabla_{\lambda_i} g_i(\lambda_i) - \nabla_{\lambda_i} g_i(\lambda_i')\|_\infty \le L_\lambda \|\lambda_i - \lambda_i'\|_2$, $\forall i \in \mathcal{N}$.*

**Assumption 4.2.** *The parameterized policy $\pi_\theta$ is such that **(I)** the score function is bounded, i.e., $\exists M_\pi > 0$ s.t. $\|\nabla_{\theta_i} \log \pi_{\theta_i}^i (a_i | s_{\mathcal{N}_i^\kappa})\|_2 \le M_\pi$, $\forall (s, a) \in \mathcal{S} \times \mathcal{A}$, $\theta \in \Theta$, $i \in \mathcal{N}$. **(II)** $\exists L_\theta > 0$ s.t. the utility functions $F(\theta) = f(\lambda^{\pi_\theta})$ and $G_i(\theta) = g_i(\lambda_i^{\pi_\theta})$ are $L_\theta$-smooth w.r.t. $\theta$, $\forall i \in \mathcal{N}$.*

**Assumption 4.3.** *There exist an FOSP $(\theta^\star, \mu^\star)$ of (5) and a constant $\overline{\mu} > 0$ s.t. $\mu_i^\star < \overline{\mu}$, $\forall i \in \mathcal{N}$. Let $\mathcal{U} = U^n = [0, \overline{\mu}]^n$.*

In Lemma F.5, we summarize a few properties that are the direct consequence consequence of Assumptions 4.1-4.3. Due to the non-concavity of problem (5), our focus is to find an approximate

first-order stationary point (FOSP). A point $(\theta, \mu) \in \Theta \times \mathcal{U}$ is said to be an $\epsilon$-FOSP if

$$\mathcal{E}(\theta, \mu) \coloneqq \left[\mathcal{X}(\theta, \mu)\right]^2 + \left[\mathcal{Y}(\theta, \mu)\right]^2 \leq \epsilon, \tag{23}$$

where the metrics $\mathcal{X}(\cdot, \cdot)$ and $\mathcal{Y}(\cdot, \cdot)$ are defined as

$$\mathcal{X}(\theta, \mu) \coloneqq \max_{\theta' \in \Theta, \|\theta' - \theta\|_2 \leq 1} \langle \nabla_\theta \mathcal{L}(\theta, \mu), \theta' - \theta \rangle, \quad \mathcal{Y}(\theta, \mu) \coloneqq - \min_{\mu' \in \mathcal{U}, \|\mu' - \mu\|_2 \leq 1} \langle \nabla_\mu \mathcal{L}(\theta, \mu), \mu' - \mu \rangle. \tag{24}$$

The definitions of $\mathcal{X}(\cdot, \cdot)$ and $\mathcal{Y}(\cdot, \cdot)$ are based on the first-order optimality condition [45, 46]. Given $\theta^\star \in \Theta$ and $\mu^\star \in \mathcal{U}$, it can be shown that $\mathcal{E}(\theta^\star, \mu^\star) = 0$ implies that $(\theta^\star, \mu^\star)$ is an FOSP of (5) (see Lemma F.6). In the following, we first consider the exact setting where the agents can obtain the true values of their local occupancy measures, shadow Q-functions, and truncated policy gradients. Therefore, the only source of approximation error is the truncation of the policy gradient.

**Theorem 4.4** (Exact setting). *Let Assumptions 3.2, 3.3, 4.1-4.3 hold and suppose that the agents can accurately estimate their local occupancy measures, shadow Q-functions, and truncated policy gradients. For every $T > 0$, let $\left\{\left(\mu^t, \theta^t\right)\right\}_{t=0}^T$ be the sequence generated by Algorithm 1 with $\eta_\mu = \mathcal{O}\left(T^{1/3}\right)$ and $\eta_\theta = 1/\left(L_{\theta\theta} + 4L_{\theta\mu}^2 \eta_\mu\right)$, where $L_{\theta\theta}, L_{\theta\mu}$ are Lipschitz constants defined in Lemma F.5. Then, there exists $t^\star \in \{0, 1, \ldots, T - 1\}$ such that*

$$\mathcal{E}\left(\theta^{t^\star}, \mu^{t^\star+1}\right) = \mathcal{O}\left(T^{-2/3}\right) + \mathcal{O}\left(\phi_0^{2\kappa}\right). \tag{25}$$

Next, we delve into the sample complexity of Algorithm 1. For theoretical analysis, we assume that the estimation process for the truncated Q-function offers an approximation to the true function, with the error being associated with the magnitude of the reward function. Let $\widehat{Q}_i^{\pi_\theta}(r_i; \cdot, \cdot) \in \mathbb{R}^{|\mathcal{S}_{\mathcal{N}_i^\kappa}| \times |\mathcal{A}_{\mathcal{N}_i^\kappa}|}$ be the truncated Q-function with the reward function $r_i(\cdot, \cdot) \in \mathbb{R}^{|\mathcal{S}_i| \times |\mathcal{A}_i|}$ for agent $i \in \mathcal{N}$.

**Assumption 4.5.** *For every reward function $r_i(\cdot, \cdot)$ and $\epsilon_0 > 0$, the subroutine computes an approximation $\widetilde{Q}_i^{\pi_\theta}(r_i; \cdot, \cdot)$ to the truncated Q-function $\widehat{Q}_i^{\pi_\theta}(r_i; \cdot, \cdot)$ such that*

$$\left\|\widetilde{Q}_i^{\pi_\theta}(r_i; \cdot, \cdot) - \widehat{Q}_i^{\pi_\theta}(r_i; \cdot, \cdot)\right\|_\infty \leq \|r_i\|_\infty \epsilon_0 \tag{26}$$

*with $\mathcal{O}(1/(\epsilon_0)^2)$ samples, for every $i \in \mathcal{N}, \theta \in \Theta$.*

We comment that the sample complexity of the truncated Q-function evaluation described in Assumption 4.5 is not restrictive. It can be achieved with high probability by the TD-learning procedure outlined in Algorithm 2 when the agents have enough exploration [10, 43]. For brevity, we assume that (26) holds almost surely. The only difference in the probabilistic version would be the presence of an additional term for the failure probability, which does not affect the order of the sample complexity.

**Theorem 4.6** (Sample-based setting). *Suppose that Assumptions 3.2, 3.3, 4.1-4.3, and 4.5 hold. For every $\epsilon > 0$ and $\delta \in (0, 1)$, let $\left\{\left(\mu^t, \theta^t\right)\right\}_{t=0}^T$ be the sequence generated by Algorithm 1 with $T = \mathcal{O}\left(\epsilon^{-1.5}\right)$, $\eta_\mu = \mathcal{O}\left(\epsilon^{-0.5}\right)$, $\eta_\theta = 1/\left(L_{\theta\theta} + 4L_{\theta\mu}^2 \eta_\mu\right)$, $\epsilon_0 = \mathcal{O}\left(\sqrt{\epsilon}\right)$, $\delta_0 = \delta/(2n(T + 1))$, batch size $B = \mathcal{O}\left(\log(1/\delta_0)\epsilon^{-2}\right)$, episode length $H = \log(1/\epsilon)$, where $L_{\theta\theta}, L_{\theta\mu}$ are Lipschitz constants defined in Lemma F.5. Then, with probability $1 - \delta$, there exists $t^\star \in \{0, 1, \ldots, T - 1\}$ such that*

$$\mathcal{E}\left(\theta^{t^\star}, \mu^{t^\star+1}\right) = \mathcal{O}\left(\epsilon\right) + \mathcal{O}(\phi_0^{2\kappa}). \tag{27}$$

*The required number of samples is $\widetilde{\mathcal{O}}\left(\epsilon^{-3.5}\right)$.*

### 4.1 Technical discussions

Theorem 4.4 implies an $\mathcal{O}\left(T^{-2/3}\right)$ iteration complexity of Algorithm 1, matching the fastest convergence rate for solving nonconcave-convex maximin problems in the literature [47]. The approximation error $\mathcal{O}\left(\phi_0^{2\kappa}\right)$ decays at a linear rate w.r.t. the radius of communications. Thus, as long as the underlying network is not densely connected, such as those in wireless communication [37] and autonomous driving [48], an approximate FOSP to (5) can be efficiently computed, while each agent $i$ only needs to communicate with a small number of agents in its neighborhood. .

In Theorem 4.4, we have chosen large step-sizes for the dual variable update to achieve the best convergence rate. This aggressive update ensures that the dual metric $\mathcal{Y}(\theta^t, \mu^{t+1})$ always remains

within a small range and also provides a satisfactory ascent direction for the policy update. Then, the average primal metric $1/T \cdot \sum_{t=0}^{T-1} \left[ \mathcal{X}\left(\theta^t, \mu^{t+1}\right) \right]^2$ is upper-bounded by exploiting a recursive relation between any two consecutive dual updates. Hence, the existence of a point $\left(\theta^{t^\star}, \mu^{t^\star+1}\right)$ that satisfies (25) is guaranteed. It is worth noting that the proof of Theorem 4.4 can be easily generalized to the scenario where $T$ is unspecified, and the same convergence rate can still be achieved with adaptive step-sizes $\eta_\mu^t = \mathcal{O}\left(t^{1/3}\right)$ and $\eta_\theta^t = 1/\left(L_{\theta\theta} + 4L_{\theta\mu}^2 \eta_\mu^t\right)$.

Theorem 4.6 states that, with high probability, Algorithm 1 has an $\widetilde{\mathcal{O}}\left(\epsilon^{-3.5}\right)$ sample complexity for finding an $\epsilon$-FOSP of (5) with an approximation error $\mathcal{O}(\phi_0^{2\kappa})$. Note that we absorb the logarithmic terms in the notation $\widetilde{\mathcal{O}}(\cdot)$. The proof of Theorem 4.6 can be broken down into two parts. Firstly, we evaluate the approximation errors of the estimators used in Algorithm 1 in relation to the model parameters, as outlined in Proposition G.1. Then, we integrate these errors into the iteration complexity result established in Theorem 4.4 and optimize the selection of parameters.

# 5   Numerical experiment

In this section, we validate Algorithm 1 via numerical experiments, focusing on three key questions[1]:

- How does Algorithm 1 perform with multiple agents, and does the policy gradient truncation effectively alleviate computational load?

- While Algorithm 1 is the first approach that provably solves the safe MARL problem with general utilities, how does it compare with existing methods for standard Safe MARL?

- What benefits does the use of general utilities offer over standard cumulative rewards?

To answer these questions, we performed multiple experiments in three environments[2]. The objective functions are based on cumulative rewards, while constraint functions leverage general utilities to incentivize or dissuade agents from exploring the environments.

**Synthetic environment**   Analogous to [24, Section 5.1], where agents are linearly arranged as $1-2-\cdots-n$. Each agent $i$ has binary local state and action spaces, i.e., $\mathcal{S}_i = \mathcal{A}_i = \{0, 1\}$, and the local transition matrix $\mathbb{P}_i$ depends solely on its action $a_i$ and the state of agent $i + 1$. The reward functions are constructed such that the optimal unconstrained policy compels all agents to continuously choose action 1, irrespective of their states.

**Pistonball**   A physics-based game that emphasizes *cooperations and high-dimensional states* as illustrated in Figure 1a. Each piston represents an agent, where its local neighborhood includes adjacent pistons, and the goal is to collectively move the ball from right to left. The agent can move up, down, or remain still. We modify the original game[49] so that the agent can only observe the ball when it enters the local neighborhood, as well as the height of neighboring pistons.

**Wireless communication**   An access control problem following a similar setup as in [24, 50]. As illustrated in Figure 1b, the agents try to transmit packets to common access points, and the transmission fails if the access point receives more than one packet simultaneously. As there are more agents than access points, *some agents need to learn to forego their benefits for the collective good*.

In addition to the objective, we incorporate two types of safety constraints characterized by general utilities that cannot be easily encapsulated by standard value functions based on cumulative rewards.

- **Entropy constraints** that stimulates exploration, formalized as $\mathrm{Entropy}(\lambda_i^{\pi_\theta}) \geq c$, $\forall i \in \mathcal{N}$. The function $\mathrm{Entropy}(\lambda_i^{\pi_\theta})$ represents the local entropy, defined as $-\sum_{s \in \mathcal{S}} d_i^\pi(s) \cdot \log\left(d_i^\pi(s)\right)$, where $d_i^{\pi_\theta}(s_i) = (1 - \gamma) \sum_{a_i \in \mathcal{A}_i} \lambda_i^{\pi_\theta}(s_i, a_i)$ is the local state occupancy measure.

- **$\ell_2$-constrains** that deter agents from learning overly randomized policies, formulated as $\left\| \sum_{s_i \in \mathcal{S}_i} \lambda_i^{\pi_\theta} \right\|_2^2 \geq c$, $\forall i \in \mathcal{N}$. This constraint is beneficial in applications like autonomous driving and human-AI collaboration, where an agent's policy needs to be predictable for other agents.

---

[1]Code is available here: https://github.com/zhykoties/Decentralized-Safe-MARL-with-General-Utilities.

[2]See Appendix H for detailed descriptions and complete experimental results.

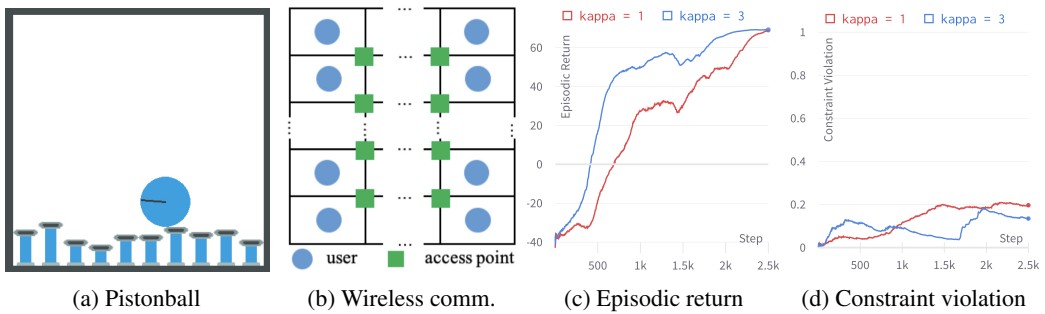

| | | | |
|---|---|---|---|
| (a) Pistonball | (b) Wireless comm. | (c) Episodic return | (d) Constraint violation |

Figure 1: (a,b) Environment illustration. (c,d) Performance of Algorithm 1 in Pistonball with 20 agents under entropy constraints.

Table 1: Comparison between Scalable Primal-Dual Actor-Critic method in our work with MAPPO-L by [31] in Pistonball and wireless communication.

| | **Pistonball** | | **Wireless Communication** | |
|---|---|---|---|---|
| **Algorithm** | **Episodic return** | **Const. vio.** | **Episodic return** | **Const. vio.** |
| Ours | **51.788 ± 1.346** | **0.04919** | **3.373 ± 0.112** | **0.1926** |
| MAPPO-L | 50.612 ± 2.118 | 0.06884 | 3.347 ± 0.131 | 0.4000 |
| Decen. Agg. MAPPO-L | 48.197 ± 6.188 | 0.2179 | 3.106 ± 0.673 | 1.1890 |
| Decen. MAPPO-L | 41.102 ± 18.769 | 0.09303 | 3.148 ± 0.614 | 1.5760 |

In Figure 1, we demonstrate the performance of Algorithm 1 in the 20-agent Pistonball environment under entropy constraints. We observe that, while the truncation with $\kappa = 3$ converges in fewer iterations, truncation with $\kappa = 1$ also yields comparable performance. This underscores the efficiency of Algorithm 1 as employing a smaller communication radius can significantly reduce the computation.

Finally, we compare Algorithm 1 with three baselines based on the MAPPO-Lagrangian method [31].

- **MAPPO-L**: the original algorithm introduced in [31]. Note that each agent has access to global information.

- **Decentralized MAPPO-L**: decentralized version of MAPPO-L, where each agent only has access to information in the local neighborhood. However, since each agent is trained to greedily maximize its individual reward, its behaviors might sacrifice the performance of other agents.

- **Decentralized Aggregate MAPPO-L**: decentralized version of MAPPO-L, where we address the aforementioned issue by redefining each agent's reward to be the sum of rewards of all agents in its local neighborhood.

For a fair comparison, we consider two standard safe MARL problems, where both objectives and constraints are shaped by cumulative rewards (see Appendix H.4). The results in Table 1 demonstrate that our method consistently outperforms both the centralized and decentralized variants of MAPPO-Lagrangian. We refer the readers to Appendix H for the comprehensive experimental results that fully answer the three questions raised at the beginning of this section.

## 6 Conclusion

In this work, we study the safe MARL with general utilities, with a focus on the setting of distributed training without global observability. To address the challenge of scalability and incorporating general utilities, we propose a primal-dual actor-critic method with shadow reward and $\kappa$-hop policy. Taking advantage of the spatial correlation decay property of the transition dynamics, we show that the proposed method achieves an $\mathcal{O}\left(T^{-2/3}\right)$ convergence rate to the FOSP of the problem in the exact setting and achieves an $\widetilde{\mathcal{O}}\left(\epsilon^{-3.5}\right)$ sample complexity, with high probability, in the sample-based setting. Finally, the effectiveness of our model and approach is verified by numerical studies. For future research, it would be interesting to develop scalable safe MARL algorithms with adaptive communication of agents' information [51] and intelligent sampling of agents' trajectories.

## Acknowledgement

This work was supported by grants from ARO, ONR, AFOSR, NSF, and the UC Noyce Initiative.

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
