$$\pi^i_{\theta_i}(\cdot|s_{\mathcal{N}^\kappa_i}, s_{\mathcal{N}^\kappa_{-i}}) = \pi^i_{\theta_i}(\cdot|s_{\mathcal{N}^\kappa_i}, s'_{\mathcal{N}^\kappa_{-i}}), \ \forall s \in \mathcal{S} \text{ and } \forall s'_{\mathcal{N}^\kappa_{-i}} \in \mathcal{S}_{\mathcal{N}^\kappa_{-i}}. \tag{12}$$

For simplicity, we use the notation $\pi^i_{\theta_i}(\cdot|s) = \pi^i_{\theta_i}(\cdot|s_{\mathcal{N}^\kappa_i})$ for $\kappa$-hop policies when it is clear from context. We note that, for any original policy function $\pi_\theta(\cdot|s)$, an induced $\kappa$-hop policy $\hat{\pi}_\theta(\cdot|s_{\mathcal{N}^\kappa_i})$ can be defined by fixing the states $s_{\mathcal{N}^\kappa_{-i}}$ to some arbitrary values and focusing only on the states of agents in $\mathcal{N}^\kappa_i$. When considering only $\kappa$-hop policies, it is essential to understand how much information is lost compared to the case where agents have access to the global states. The following proposition quantifies the maximum information loss in terms of the occupancy measure under the assumption that the original policy function also satisfies a spatial correlation decay property.

**Proposition 3.4.** *Suppose that there exist $c \geq 0$ and $\phi \in [0, 1)$ such that for every $\theta \in \Theta$, agent $i \in \mathcal{N}$, and states $s, s' \in \mathcal{S}$ such that $s_{\mathcal{N}^\kappa_i} = s'_{\mathcal{N}^\kappa_i}$, we have $\left\| \pi^i_{\theta_i}(\cdot|s) - \pi^i_{\theta_i}(\cdot|s') \right\|_1 \leq c\phi^\kappa$. Let $\hat{\pi}_\theta$ be an induced $\kappa$-hop policy of $\pi_\theta$. Then, it holds that*

$$\left\| \lambda^{\hat{\pi}_\theta}_i - \lambda^{\pi_\theta}_i \right\|_1 \leq \frac{nc\phi^k}{(1-\gamma)^2}, \forall i \in \mathcal{N}. \tag{13}$$

The condition on the local policy in Proposition 3.4 encodes that every $\pi^i_{\theta_i}$ is exponentially less sensitive to the states of agents outside $\mathcal{N}^\kappa_i$, which is a common assumption in MARL to alleviate computationally burdensome and practically intractable communication requirements imposed by the global observability [34, 39, 38]. By Proposition 3.4, the difference in occupancy measures under $\pi_\theta$ and $\hat{\pi}_\theta$ is controlled by $\|\pi^i_{\theta_i} - \hat{\pi}^i_{\theta_i}\|_1$. Therefore, if $f_i(\lambda^\pi)$ and $g_i(\lambda^\pi)$ are Lipschitz continuous w.r.t. $\lambda^\pi$, Proposition 3.4 implies an $\mathcal{O}(\phi^\kappa)$ approximation of the Lagrangian function (6) using $\kappa$-hop policies. The faster the spatial decay of policy is, the more accurate the approximation of the $\kappa$-hop policy is. This justifies our focus on learning a $\kappa$-hop policy.

### 3.2 Truncated policy gradient estimator

In the absence of global observability, it is critical to find a scalable estimator for the local gradient $\nabla_{\theta_i} \mathcal{L}(\theta, \mu)$ in (10), so that each agent can update its local policy with limited communications.

By leveraging the similar idea in the definition of $\kappa$-hop policies, we define the $\kappa$-*hop truncated (shadow) Q-function*, denoted as $\widehat{Q}^{\pi_\theta}_{\diamond_i} : \mathcal{S}_{\mathcal{N}^\kappa_i} \times \mathcal{A}_{\mathcal{N}^\kappa_i} \to \mathbb{R}$, to be

$$\widehat{Q}^{\pi_\theta}_{\diamond_i}(s_{\mathcal{N}^\kappa_i}, a_{\mathcal{N}^\kappa_i}) := Q^{\pi_\theta}_{\diamond_i}(s_{\mathcal{N}^\kappa_i}, \bar{s}_{\mathcal{N}^\kappa_{-i}}, a_{\mathcal{N}^\kappa_i}, \bar{a}_{\mathcal{N}^\kappa_{-i}}), \ \forall (s_{\mathcal{N}^\kappa_i}, a_{\mathcal{N}^\kappa_i}) \in \mathcal{S}_{\mathcal{N}^\kappa_i} \times \mathcal{A}_{\mathcal{N}^\kappa_i}, \diamond \in \{f, g\}, \tag{14}$$

where $(\bar{s}_{\mathcal{N}^\kappa_{-i}}, \bar{a}_{\mathcal{N}^\kappa_{-i}})$ is any fixed state-action pair for the agents in $\mathcal{N}^\kappa_{-i}$. Now, we introduce the following *truncated policy gradient estimator* for agent $i$:

$$\widehat{\nabla}_{\theta_i} \mathcal{L}(\theta, \mu) = \frac{1}{1-\gamma} \mathbb{E}_{\substack{s \sim d^{\pi_\theta} \\ a \sim \pi_\theta(\cdot|s)}} \left[ \nabla_{\theta_i} \log \pi^i_{\theta_i}(a_i|s_{\mathcal{N}^\kappa_i}) \cdot \frac{1}{n} \sum_{j \in \mathcal{N}^\kappa_i} \left( \widehat{Q}^{\pi_\theta}_{f_j}(s_{\mathcal{N}^\kappa_j}, a_{\mathcal{N}^\kappa_j}) + \mu_j \widehat{Q}^{\pi_\theta}_{g_j}(s_{\mathcal{N}^\kappa_j}, a_{\mathcal{N}^\kappa_j}) \

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

# Supplementary Materials

## Limitations

This is a theoretical work that concerns with algorithm design for safe multi-agent reinforcement learning with general utilities. The main results in the paper characterize the convergence rate of the proposed algorithm to an approximate first-order stationary point. The proof of the main results rely on several technical assumptions, which are detailedly discussed in Appendix F.1.

## A   Related work

**Safe MARL**   The study of provably efficient algorithms for safe RL has received considerable attention due to the crucial role of safety in autonomous systems [11, 52, 53, 54, 55, 56]. Our work is closely related to Lagrangian-based CMDP algorithms [57, 58, 59, 60, 61, 62, 63], which update the primal variable via policy gradient ascent and updates the dual variable via projected sub-gradient descent. The concept of safe RL has also been extended to multi-agent systems. Specifically, [25] study the distributed consensus CMDP with networked agents and propose a decentralized policy gradient method to perform policy optimization over a network. Furthermore, [31] propose a safe multi-agent policy iteration procedure that attains the monotonic improvement guarantee and constraints satisfaction guarantee at every iteration, but has no convergence guarantee. In addition, [26] adopt a mean-field control approach for safe MARL and provide a natural policy gradient-based algorithm. Furthermore, the Nash equilibrium for constrained Markov potential games has been studied in [64, 65, 66, 67], using the notion of constrained Nash equilibrium [68, 69, 70]. These results are not applicable to the RL setting that assumes unknown models. Recently, [27] prove the first result on the non-asymptotic convergence to the constrained Nash equilibrium by adding built-in exploration mechanisms under constraints. However, these works all assume access to the global state of all agents, whereas our method only requires the state-action information in the local neighborhood while also guaranteeing small performance loss.

**MARL with general utilities**   A series of recent works have focused on developing general approaches for RL with general utilities (also known as convex MDPs) [17, 71, 23, 16, 72, 61, 73]. In particular, our work is closely related to [23, 73] which extend RL with general utilities to multi-agent systems. Specifically, [23] propose a decentralized shadow reward actor-critic (DSAC) method in which agents alternate between policy evaluation (critic), weighted averaging with neighbors (information mixing), and local gradient updates for their policy parameters (actor). The DSAC approach augments the classic critic step by requiring the agents to estimate their local occupancy

measure in order to estimate the derivative of the local utility with respect to their occupancy measure, i.e., the "shadow reward". However, this approach assumes full observability, i.e., each agent should have access to the global states and actions of the team, which limits its application to systems with a large numbers of agents. To address this issue, [73] develop a scalable algorithm for multi-agent RL with general utilities without the full observability assumption by exploiting the spatial correlation decay property of the network structure [10]. However, these works only consider the unconstrained RL problem, which may lead to undesired policies in safety-critical applications. Therefore, additional effort is required to deal with the emerging safety constraints while guaranteeing the scalability, and our work addresses this problem.

## B  Notations

For a finite set $\mathcal{S}$, let $|\mathcal{S}|$ denote its cardinality. When the variable $s$ follows the distribution $\rho$, we write it as $s \sim \rho$. Let $\mathbb{E}[\cdot]$ and $\mathbb{E}[\cdot \mid \cdot]$, respectively, denote the expectation and conditional expectation of a random variable. Let $\mathbb{R}$ denote the set of real numbers. For a vector $x$, we use $x^\top$ to denote the transpose of $x$ and use $\langle x, y \rangle$ to denote the inner product $x^\top y$. We use the convention that $\|x\|_1 = \sum_i |x_i|$, $\|x\|_2 = \sqrt{\sum_i x_i^2}$, and $\|x\|_\infty = \max_i |x_i|$. When applied to a matrix $A$, the norms are referred to as the induced norms, e.g., $\|A\|_1 = \max_{\|x\|_1 \neq 0} \{\|Ax\|_1 / \|x\|_1\}$ is the induced 1-norm and $\|A\|_2 = \max_{\|x\|_2 \neq 0} \{\|Ax\|_2 / \|x\|_2\}$ stands for the spectral norm. For a set $X \subset \mathbb{R}^p$, let $\mathrm{cl}(X)$ denote the closure of $X$. Let $\mathcal{P}_X$ denote the projection onto $X$, defined as $\mathcal{P}_X(y) \coloneqq \arg\min_{x \in X} \|x - y\|_2$. For a function $f(x)$, let $\arg\min f(x)$ (resp. $\arg\max f(x)$) denote any global minimum (resp. global maximum) of $f(x)$ and let $\nabla_x f(x)$ denote its gradient with respect to $x$.

## C  Further details on CMDPs with general utilities

In standard CMDPs, the objective function and the constraint function take the form of discounted cumulative rewards, i.e.,

$$
\begin{aligned}
\max_\pi \; & V^\pi(r) \coloneqq \mathbb{E}\left[\sum_{k=0}^\infty \gamma^k r\left(s^k, a^k\right) \,\middle|\, a^k \sim \pi(\cdot|s^k), s^0 \sim \rho \right], \\
\text{s.t.} \; & V^\pi(u) \coloneqq \mathbb{E}\left[\sum_{k=0}^\infty \gamma^k u\left(s^k, a^k\right) \,\middle|\, a^k \sim \pi(\cdot|s^k), s^0 \sim \rho \right] \geq 0.
\end{aligned}
\tag{28}
$$

By using the definition of the occupancy measure $\lambda^\pi$ (see (1)), we can equivalently write problem (28) as

$$
\max_\pi \; f(\lambda^\pi) = \langle r, \lambda^\pi \rangle, \quad \text{s.t.} \quad g(\lambda^\pi) = \langle u, \lambda^\pi \rangle \geq 0.
\tag{29}
$$

When viewing $\lambda^\pi$ as the decision variable, (29) is known to be the linear programming formulation of the CMDP [11]. Thus, the standard CMDP problem is a special case of CMDPs with general utilities.

However, many decision-making problems of interests take a form beyond cumulative rewards. We give three examples of such problems below.

**Example C.1** (Safety-aware apprenticeship learning [74]). *In apprenticeship learning, the agent learns to mimic an expert's demonstrations instead of maximizing the long-term reward. In the presence of critical safety requirements, the learner will also strive to satisfy given constraints on the expected total cost. This problem can be formulated as*

$$
\max_\pi \; f(\lambda^\pi) = -\mathrm{dist}(\lambda^\pi, \lambda_e) \;\; s.t. \;\; g(\lambda^\pi) = \langle c, \lambda^\pi \rangle \leq 0,
$$

*where $\lambda_e$ is the occupancy measure corresponding to the expert demonstration, $c$ denotes the vector of costs, and $\mathrm{dist}(\cdot, \cdot)$ can be any distance function on the set of occupancy measures, e.g., $\ell^2$-distance or Kullback-Liebler (KL) divergence.*

**Example C.2** (Feasibility constrained MDPs [75]). *As an extension to standard CMDPs, the designer may desire to control the MDP through limiting the deviation of the learned policy from a convex feasibility region $C$, e.g., $C$ may be a single point representing the occupancy measure of a known safe policy. In this case, the problem can be cast as*

$$
\max_\pi \; f(\lambda^\pi) = \langle r, \lambda \rangle \;\; s.t. \;\; g(\lambda^\pi) = \mathrm{dist}(\lambda, C) \leq d_0,
$$

*where $r$ is the reward vector of the underlying MDP and $d_0 \geq 0$ denotes the threshold of the allowable deviation.*

**Example C.3** (Constrained entropy maximization [76])**.** *In the absence of a reward function, a suitable intrinsic objective for the agent is to maximize the speed at which it explores the environment. However, in safety-critical systems, it is important to account for the safety risks inevitably brought by the pursuit of exploration. In this scenario, one can consider the problem*

$$\max_\pi f(\lambda^\pi) = -\sum_{s \in \mathcal{S}} d^\pi(s) \cdot \log\big(d^\pi(s)\big), \ \ s.t. \ g(\lambda^\pi) = \langle c, \lambda^\pi \rangle \leq 0,$$

*where $d^\pi(s) := (1-\gamma) \sum_{a \in \mathcal{A}} \lambda^\pi(s,a)$ is the discounted state occupancy measure and $f(\lambda^\pi)$ computes the entropy of the distribution $d^\pi(\cdot)$ under policy $\pi$.*

Finally, for the distributed problem (5), we remark that it recovers the standard constrained MARL when all local utilities are linear, namely

$$\max_{\theta \in \Theta} F(\theta) = \frac{1}{n} \sum_{i \in \mathcal{N}} \langle r_i, \lambda_i^{\pi_\theta} \rangle = \frac{1}{n} \sum_{i \in \mathcal{N}} \mathbb{E}\left[ \sum_{k=0}^{\infty} \gamma^k r_i\left(s_i^k, a_i^k\right) \bigg| a^k \sim \pi_\theta(\cdot|s^k), s^0 \sim \rho \right],$$

$$\text{s.t. } G_i(\theta) = \langle u_i, \lambda_i^{\pi_\theta} \rangle = \mathbb{E}\left[ \sum_{k=0}^{\infty} \gamma^k u_i\left(s_i^k, a_i^k\right) \bigg| a^k \sim \pi_\theta(\cdot|s^k), s^0 \sim \rho \right] \geq 0, \ \forall i \in \mathcal{N},$$

(30)

where $r_i : \mathcal{S}_i \times \mathcal{A}_i \to \mathbb{R}$ and $u_i : \mathcal{S}_i \times \mathcal{A}_i \to \mathbb{R}$ are vectors of local rewards and utilities, respectively. The problem (30) is still not separable since the transition dynamics are coupled and the decisions of the agents are intertwined through their policies.

## D  Further discussions on Algorithm 1

In this section, we provide a line-by-line discussion of the algorithm to offer further clarity. A detailed flow diagram of Algorithm 1 at each iteration $t$ is shown in Figure 2.

- **Line 3 (trajectory sampling):** In order to estimate the Lagrangian $\mathcal{L}(\theta, \mu)$ and its gradients $\nabla_\theta \mathcal{L}(\theta, \mu), \nabla_\mu \mathcal{L}(\theta, \mu)$, which depend on occupancy measures, we first make the agents estimate their local occupancy measures through trajectory sampling. At the beginning of each period $t$, $B$ batches of trajectories are sampled under the $\kappa$-hop policy $\pi_{\theta^t}$. Because the local policy $\pi_{\theta_i^t}^i$ only depends on the states of agent $i$'s $\kappa$-hop neighbors, each agent $i$ only needs to communicate with these neighbors to make decisions. Specifically, in each period $k$, the environment first samples state $s^k \sim \mathbb{P}(\cdot|s^{k-1}, a^{k-1})$. Then, each agent $i$ obtains the states of its neighbors $s_{\mathcal{N}_i^\kappa}^k$ and takes action according to $a_i^k \sim \pi_{\theta_i}^i(\cdot|s_{\mathcal{N}_i^\kappa}^k)$. After the sampling procedure, each agent $i$ collects the partial trajectories within its communication radius, which are trajectories formed by the state-action pairs of the agents in $\mathcal{N}_i^\kappa$.

- **Line 4 (occupancy measure estimation):** With access to the batch of trajectories $\mathcal{B}_i^t$, each agent then forms an estimate for its local occupancy measure $\lambda_i^{\pi_{\theta^t}}$ under $\pi_{\theta^t}$ through (17). Note that $\mathbb{1}_i\left(s_i^k, a_i^k\right) \in \mathbb{R}^{|\mathcal{S}_i| \times |\mathcal{A}_i|}$ is an indicator vector, where all its entries are zero except for its $(s_i^k, a_i^k)$-th entry being one. Thus, the estimator (17) approximates the expected value (4) by counting the discounted visiting time of different state-action pairs and taking the average over a batch of trajectories. Such a Monte Carlo estimation is also used in [23, 71]. The accuracy of this occupancy measure estimator is quantified in Proposition G.1.

- **Line 5 (constraint and shadow reward evaluation):** Recall that the shadow rewards are defined as $r_{f_i}^{\pi_\theta} = \nabla_{\lambda_i} f_i(\lambda_i^{\pi_\theta})$ and $r_{g_i}^{\pi_\theta} = \nabla_{\lambda_i} g_i(\lambda_i^{\pi_\theta})$ in Definition 3.1. With empirical occupancy measures, each agent $i$ can directly compute their constraint function value as $\widetilde{g}_i^t = g_i(\widetilde{\lambda}_i^t)$ and shadow rewards as $\widetilde{r}_{f_i}^t = \nabla_{\lambda_i} f_i(\widetilde{\lambda}_i^t)$ and $\widetilde{r}_{g_i}^t = \nabla_{\lambda_i} g_i(\widetilde{\lambda}_i^t)$, where $\widetilde{g}_i^t$ is used in the dual update (Line 7) and shadow rewards are used in the Q-function evaluation (Line 6). When $f_i(\cdot)$ and $g_i(\cdot)$ satisfy proper smoothness assumptions, e.g., Assumption 4.1, the approximation errors of these estimators are proportional to the errors of empirical occupancy measures.

- **Line 6 (truncated shadow Q-function evaluation):** To compute the truncated policy gradient estimator, the agents need to estimate their truncated shadow Q-functions. For the estimation

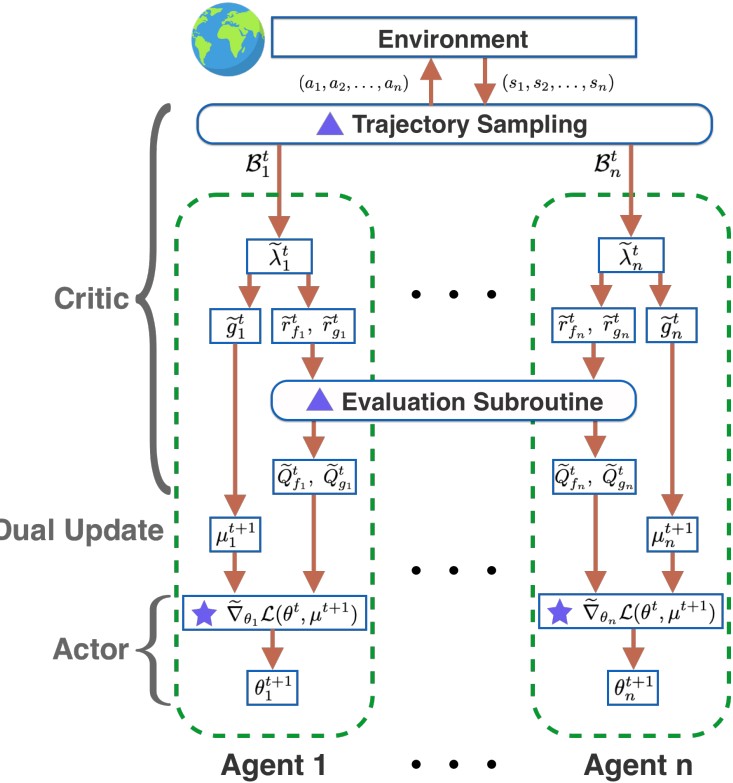

Figure 2: The flow diagram of Algorithm 1 at iteration $t$. There are three stages: policy evaluation by the critic (line 3-6); Lagrangian multiplier update (line 7); policy update by the actor (line 8-9). The steps highlighted by ▲ require each agent $i$ to access the states/actions of the agents in its $\kappa$-hop neighborhood, i.e., $\left(s_{\mathcal{N}_i^\kappa}, a_{\mathcal{N}_i^\kappa}\right)$. The step highlighted by ★ corresponds to the computation of policy gradient, which requires each agent $i$ to share its local shadow Q-functions and dual variable with the agents in $\mathcal{N}_i^\kappa$.

process, we introduce the distributed TD-learning algorithm [43], which is a model-free method as outlined in Algorithm 2. In each iteration, a new state $s^k$ is sampled by the environment according to the transition probability $\mathbb{P}(\cdot|s^{k-1}, a^{k-1})$. Then, each agent $i$ exchanges its state information with agents in the neighborhood $\mathcal{N}_i^\kappa$ and makes a decision using its $\kappa$-hop local policy $\pi_{\theta_i}^i$, i.e., sampling an action $a_i^k \sim \pi_{\theta_i}^i(\cdot|s_{\mathcal{N}_i^\kappa}^k)$. Finally, the existing estimation $\widetilde{Q}_i^{k-1}$ is updated using the TD-learning update in (31). This update is based on the Bellman equation [77]. The term $\left(r_i(s_i^{k-1}, a_i^{k-1}) + \gamma \widetilde{Q}_i^{k-1}(s_{\mathcal{N}_i^\kappa}^k, a_{\mathcal{N}_i^\kappa}^k)\right) - \widetilde{Q}_i^{k-1}(s_{\mathcal{N}_i^\kappa}^{k-1}, a_{\mathcal{N}_i^\kappa}^{k-1})$ is referred to as the temporal difference error, which can be viewed as a correction to the prior estimate after receiving a new reward. As described in Section 3.3, this subroutine serves as an example of how the truncated Q-function estimation can be computed and can be replaced by any other suitable approach that satisfies the observation and communication requirements (also see the discussion in Appendix F.1.4).

- **Line 7 (dual variable update):** The dual variable is updated by solving the sub-problem in (22), which is equivalent to

$$\mu^{t+1} = \underset{\mu \in \mathcal{U}}{\mathrm{argmin}} \langle \nabla_\mu \mathcal{L}(\theta^t, \mu^t), \mu - \mu^t \rangle + \frac{1}{2\eta_\mu} \|\mu\|_2^2,$$

by the linearity of $\mathcal{L}(\theta^t, \mu)$ in $\mu$. Thus, it is clear that the sub-problem yields the solution in (22). The regularization term $1/(2\eta_\mu) \cdot \|\mu\|_2^2$ helps to provide curvature to the problem and can also be substituted by $1/(2\eta_\mu) \cdot \|\mu - \mu_0\|_2^2/2$ for any fixed point $\mu_0 \in \mathcal{U}$. We assume the feasible region for $\mu$ is a high-dimensional box, denoted by $\mathcal{U} := U^n = [0, \overline{\mu}]^n$, where $\overline{\mu}$ is some fixed number. To optimize the convergence rate/sample complexity in the analysis, we will choose large values for $\eta_\mu$, resulting in an aggressive dual update. Empirically, the benefit of having a large $\eta_\mu$ is to

**Algorithm 2** Evaluation Subroutine Based on Temporal Difference Learning [43] (Eval)

---

1: **Input:** $\kappa$-hop policy $\pi_\theta$; local shadow rewards $\{r_i\}_{i \in \mathcal{N}}$; communication radius $\kappa$; initial truncated shadow Q-functions $\left\{ \widetilde{Q}_i^0 \in \mathbb{R}^{|\mathcal{S}_{\mathcal{N}_i^\kappa}| \times |\mathcal{A}_{\mathcal{N}_i^\kappa}|} \right\}_{i \in \mathcal{N}}$ as zero vectors; uniform initial distribution $\rho_0$; episode length $K$; step-sizes $\{\eta_Q^k\}_{k=0}^{K-1}$.

2: Sample the initial state $s^0 \sim \rho_0$. Each agent $i$ obtains the states of neighbors $s_{\mathcal{N}_i^\kappa}^0$, takes action according to $a_i^0 \sim \pi_{\theta_i}^i(\cdot|s_{\mathcal{N}_i^\kappa}^0)$, and receives the reward $r_i(s_i^0, a_i^0)$.

3: **for** iteration $k = 1, 2, \ldots, K$ **do**

4:    Sample state $s^k \sim \mathbb{P}(\cdot|s^{k-1}, a^{k-1})$. Each agent $i$ obtains the states of neighbors $s_{\mathcal{N}_i^\kappa}^k$, takes action according to $a_i^k \sim \pi_{\theta_i}^i(\cdot|s_{\mathcal{N}_i^\kappa}^k)$, and receives the reward $r_i(s_i^k, a_i^k)$.

5:    Each agent $i$ communicates with its neighbor agents $\mathcal{N}_i^\kappa$ and updates its truncated shadow Q-functions through

$$
\widetilde{Q}_i^k(s_{\mathcal{N}_i^\kappa}^{k-1}, a_{\mathcal{N}_i^\kappa}^{k-1}) \leftarrow \widetilde{Q}_i^{k-1}(s_{\mathcal{N}_i^\kappa}^{k-1}, a_{\mathcal{N}_i^\kappa}^{k-1}) + \eta_Q^{k-1} \Big[ \Big( r_i(s_i^{k-1}, a_i^{k-1}) + \gamma \widetilde{Q}_i^{k-1}(s_{\mathcal{N}_i^\kappa}^k, a_{\mathcal{N}_i^\kappa}^k) \Big)
$$
$$
- \widetilde{Q}_i^{k-1}(s_{\mathcal{N}_i^\kappa}^{k-1}, a_{\mathcal{N}_i^\kappa}^{k-1}) \Big], \quad (31)
$$
$$
\widetilde{Q}_i^k(s_{\mathcal{N}_i^\kappa}, a_{\mathcal{N}_i^\kappa}) \leftarrow \widetilde{Q}_i^{k-1}(s_{\mathcal{N}_i^\kappa}, a_{\mathcal{N}_i^\kappa}), \ \forall (s_{\mathcal{N}_i^\kappa}, a_{\mathcal{N}_i^\kappa}) \neq (s_{\mathcal{N}_i^\kappa}^{k-1}, a_{\mathcal{N}_i^\kappa}^{k-1}).
$$

6: **end for**

7: **Output:** Empirical truncated shadow Q-functions $\left\{ \widetilde{Q}_i^K \right\}_{i \in \mathcal{N}}$.

---

also ensure a relative low constraint violation during the training stage, which is essential in many safety-critical systems.

- **Line 8 (policy gradient evaluation):** The agents approximates their policy gradients through the truncated policy gradient defined in (15). By the equivalent forms of the policy gradient theorem (see Lemma D.1), (15) can be written as

$$
\widehat{\nabla}_{\theta_i} \mathcal{L}(\theta, \mu) = \mathbb{E}\left[ \sum_{k=0}^{\infty} \gamma^k \nabla_{\theta_i} \log \pi_{\theta_i}^i(a_i^k|s_{\mathcal{N}_i^\kappa}^k) \cdot \frac{1}{n} \sum_{j \in \mathcal{N}_i^\kappa} \left( \widehat{Q}_{f_j}^{\pi_\theta}(s_{\mathcal{N}_j^\kappa}^k, a_{\mathcal{N}_j^\kappa}^k) + \mu_j \widehat{Q}_{g_j}^{\pi_\theta}(s_{\mathcal{N}_j^\kappa}^k, a_{\mathcal{N}_j^\kappa}^k) \right) \right],
$$

where the expectation is taken over all possible trajectories under policy $\pi_\theta$. Thus, the truncated policy gradient $\widehat{\nabla}_{\theta_i} \mathcal{L}(\theta^t, \mu^{t+1})$ can be estimated through a REINFORCE-based mechanism [44] as shown in (20). It is important to note that, since all the batches $\{\mathcal{B}_i^t\}_{i \in \mathcal{N}}$ come from the same global trajectories sampled in Line 3, the values of each agent $i$'s empirical truncated Q-functions along the trajectories in its batch, i.e., $\left\{ \{\widetilde{Q}_{\diamond_i}^t(s_{\mathcal{N}_i^\kappa}^k, a_{\mathcal{N}_i^\kappa}^k)\}_{k=0}^{H-1} \right\}_{\tau \in \mathcal{B}_i^t}$ for $\diamond \in \{f, g\}$, are used in the computation of $\widetilde{\nabla}_{\theta_j} \mathcal{L}(\theta^t, \mu^{t+1})$ for all agents $j \in \mathcal{N}_i$. Therefore, it is sufficient for each agent $i$ to share this information and its updated dual variable $\mu_i^{t+1}$ with all other agents in its neighborhood $\mathcal{N}_i^\kappa$.

- **Line 9 (policy parameter update):** The policy update uses the vanilla projected gradient ascent with the estimated gradient in (20).

### D.1 Policy gradient theorem

In this section, we present the well-known policy gradient theorem [30].

**Lemma D.1** (Policy gradient under general parameterization). *Let $V^{\pi_\theta}(r)$ be a standard value function under policy $\pi_\theta$ with an arbitrary reward function $r : \mathcal{S} \times \mathcal{A} \to \mathbb{R}$, defined as*

$$
V^{\pi_\theta}(r) := \langle r, \lambda^{\pi_\theta} \rangle = \mathbb{E}\left[ \sum_{k=0}^{\infty} \gamma^k r\left(s^k, a^k\right) \Big| a^k \sim \pi_\theta(\cdot|s^k), s^0 \sim \rho \right].
$$

*The gradient of $V^{\pi_\theta}(r)$ with respect to $\theta$ can be given by the following three equivalent forms:*

$$\nabla_\theta V^{\pi_\theta}(r) = r^\top \cdot \nabla_\theta \lambda^{\pi_\theta} = \frac{1}{1-\gamma} \mathbb{E}_{s \sim d^{\pi_\theta}} \mathbb{E}_{a \sim \pi_\theta(\cdot|s)} \left[ \nabla_\theta \log \pi_\theta(a|s) \cdot Q^{\pi_\theta}(r; s, a) \right]$$

$$= \mathbb{E}\left[ \sum_{k=0}^\infty \gamma^k \nabla_\theta \log \pi_\theta(a^k|s^k) \cdot Q^{\pi_\theta}(r; s^k, a^k) \,\Big|\, a^k \sim \pi_\theta(\cdot|s^k), s^0 \sim \rho \right]$$

$$= \mathbb{E}\left[ \sum_{k=0}^\infty \gamma^k \cdot r(s^k, a^k) \cdot \left( \sum_{k'=0}^k \nabla_\theta \log \pi_\theta(a^{k'}|s^{k'}) \right) \,\Big|\, a^k \sim \pi_\theta(\cdot|s^k), s^0 \sim \rho \right],$$

*where $d^{\pi_\theta}(s) := (1-\gamma) \sum_{a \in \mathcal{A}} \lambda^{\pi_\theta}(s, a)$ is the discounted state occupancy measure, and $Q^{\pi_\theta}(r; \cdot, \cdot)$ is the state-action value function (Q-function) with reward $r$ defined in (8).*

## E   Supplementary materials for section 3

In this section, we provide the proofs of the results in Section 3.

### E.1   Proof of Proposition 3.4

*Proof.* For ease of notations, we treat the set of agents $\mathcal{N} = \{1, 2, \ldots, n\}$ and the set of numbers $[n] = \{1, 2, \ldots, n\}$ as equivalent when it is clear from the context. Given the decay condition $\left\| \pi_{\theta_i}^i(\cdot|s) - \pi_{\theta_i}^i(\cdot|s') \right\|_1 \le c\phi^\kappa$, it is clear that the induced $\kappa$-hop policy $\hat{\pi}_\theta$ satisfies that

$$\left\| \hat{\pi}_{\theta_i}^i(\cdot|s_{\mathcal{N}_i^\kappa}) - \pi_{\theta_i}^i(\cdot|s) \right\|_1 \le c\phi^\kappa, \ \forall s \in \mathcal{S}, i \in \mathcal{N}.$$

Below, we first bound the difference between the global policies $\pi_\theta$ and $\hat{\pi}_\theta$ by leveraging the policy factorization as follows

$$
\begin{aligned}
\|\hat{\pi}_\theta(\cdot|s) - \pi_\theta(\cdot|s)\|_1 &= \sum_{a \in \mathcal{A}} |\hat{\pi}_\theta(a|s) - \pi_\theta(a|s)| \\
&= \sum_{a \in \mathcal{A}} \left| \prod_{i=1}^n \hat{\pi}_{\theta_i}^i(a_i|s_{\mathcal{N}_i^\kappa}) - \prod_{i=1}^n \pi_{\theta_i}^i(a_i|s) \right| \\
&= \sum_{a \in \mathcal{A}} \left| \prod_{i=1}^n \hat{\pi}_{\theta_i}^i(a_i|s_{\mathcal{N}_i^\kappa}) - \hat{\pi}_{\theta_1}^1(a_1|s_{\mathcal{N}_1^\kappa}) \prod_{i=2}^n \pi_{\theta_i}^i(a_i|s) \right. \\
&\qquad \left. + \hat{\pi}_{\theta_1}^1(a_1|s_{\mathcal{N}_1^\kappa}) \prod_{i=2}^n \pi_{\theta_i}^i(a_i|s) - \prod_{i=1}^n \pi_{\theta_i}^i(a_i|s) \right| \\
&\le \sum_{a \in \mathcal{A}} \left| \prod_{i=1}^n \hat{\pi}_{\theta_i}^i(a_i|s_{\mathcal{N}_i^\kappa}) - \hat{\pi}_{\theta_1}^1(a_1|s_{\mathcal{N}_1^\kappa}) \prod_{i=2}^n \pi_{\theta_i}^i(a_i|s) \right| \\
&\qquad + \sum_{a \in \mathcal{A}} \left| \hat{\pi}_{\theta_1}^1(a_1|s_{\mathcal{N}_1^\kappa}) \prod_{i=2}^n \pi_{\theta_i}^i(a_i|s) - \prod_{i=1}^n \pi_{\theta_i}^i(a_i|s) \right| \\
&= \sum_{a \in \mathcal{A}} \hat{\pi}_{\theta_1}^1(a_1|s_{\mathcal{N}_1^\kappa}) \cdot \left| \prod_{i=2}^n \hat{\pi}_{\theta_i}^i(a_i|s_{\mathcal{N}_i^\kappa}) - \prod_{i=2}^n \pi_{\theta_i}^i(a_i|s) \right| \\
&\qquad + \sum_{a \in \mathcal{A}} \left| \hat{\pi}_{\theta_1}^1(a_1|s_{\mathcal{N}_1^\kappa}) - \pi_{\theta_1}^1(a_1|s) \right| \cdot \prod_{i=2}^n \pi_{\theta_i}^i(a_i|s),
\end{aligned}
\tag{32}
$$

where we used the triangular inequality. For the second term above, it holds that

$$
\begin{aligned}
& \sum_{a \in \mathcal{A}} \left| \hat{\pi}_{\theta_1}^1(a_1|s_{\mathcal{N}_1^\kappa}) - \pi_{\theta_1}^1(a_1|s) \right| \cdot \prod_{i=2}^n \pi_{\theta_i}^i(a_i|s) \\
&= \sum_{a_i \in \mathcal{A}_1} \left| \hat{\pi}_{\theta_1}^1(a_1|s_{\mathcal{N}_1^\kappa}) - \pi_{\theta_1}^1(a_1|s) \right| \cdot \sum_{a_{-1} \in \mathcal{A}_{-1}} \prod_{i=2}^n \pi_{\theta_i}^i(a_i|s) \\
&= \sum_{a_i \in \mathcal{A}_1} \left| \hat{\pi}_{\theta_1}^1(a_1|s_{\mathcal{N}_1^\kappa}) - \pi_{\theta_1}^1(a_1|s) \right| \\
&= \left\| \hat{\pi}_{\theta_1}^1(\cdot|s_{\mathcal{N}_1^\kappa}) - \pi_{\theta_1}^1(\cdot|s) \right\|_1.
\end{aligned}
\tag{33}
$$

The first term in the right-hand side of (32) can be further written as

$$\sum_{a \in \mathcal{A}} \hat{\pi}_{\theta_1}^1(a_1 | s_{\mathcal{N}_1^\kappa}) \cdot \left| \prod_{i=2}^n \hat{\pi}_{\theta_i}^i(a_i | s_{\mathcal{N}_i^\kappa}) - \prod_{i=2}^n \pi_{\theta_i}^i(a_i | s) \right|$$

$$= \sum_{a_1 \in \mathcal{A}_1} \hat{\pi}_{\theta_1}^1(a_1 | s_{\mathcal{N}_1^\kappa}) \sum_{a_{-1} \in \mathcal{A}_{-1}} \left| \prod_{i=2}^n \hat{\pi}_{\theta_i}^i(a_i | s_{\mathcal{N}_i^\kappa}) - \prod_{i=2}^n \pi_{\theta_i}^i(a_i | s) \right| \qquad (34)$$

$$= \sum_{a_{-1} \in \mathcal{A}_{-1}} \left| \prod_{i=2}^n \hat{\pi}_{\theta_i}^i(a_i | s_{\mathcal{N}_i^\kappa}) - \prod_{i=2}^n \pi_{\theta_i}^i(a_i | s) \right|.$$

Together, by substituting (33) and (34) into (32), we have that

$$\| \hat{\pi}_\theta(\cdot | s) - \pi_\theta(\cdot | s) \|_1$$

$$= \sum_{a \in \mathcal{A}} \left| \prod_{i=1}^n \hat{\pi}_{\theta_i}^i(a_i | s_{\mathcal{N}_i^\kappa}) - \prod_{i=1}^n \pi_{\theta_i}^i(a_i | s) \right|$$

$$\leq \left\| \hat{\pi}_{\theta_1}^1(\cdot | s_{\mathcal{N}_1^\kappa}) - \pi_{\theta_1}^1(\cdot | s) \right\|_1 + \sum_{a_{-1} \in \mathcal{A}_{-1}} \left| \prod_{i=2}^n \hat{\pi}_{\theta_i}^i(a_i | s_{\mathcal{N}_i^\kappa}) - \prod_{i=2}^n \pi_{\theta_i}^i(a_i | s) \right|$$

$$\leq \sum_{i \in \mathcal{N}} \left\| \hat{\pi}_{\theta_i}^i(\cdot | s_{\mathcal{N}_i^\kappa}) - \pi_{\theta_i}^i(\cdot | s) \right\|_1$$

$$\leq n c \phi^\kappa, \ \forall s \in \mathcal{S},$$

where the second inequality follows from recursively applying the derivations in (32).

Now, before showing the desired bound (13), we first derive an upper bound on $\left\| \lambda^{\hat{\pi}_\theta} - \lambda^{\pi_\theta} \right\|_1$ using the matrix representation of the occupancy measure. For a given policy $\pi$, let $\rho^\pi \in \mathbb{R}^{|\mathcal{S}||\mathcal{A}|}$ be the vector that represents the joint distribution of the state-action pair in the initial period, i.e., $\rho^\pi(s, a) := \rho(s)\pi(a|s)$. Let $\mathbb{P}^\pi \in \mathbb{R}^{|\mathcal{S}||\mathcal{A}| \times |\mathcal{S}||\mathcal{A}|}$ be the matrix of transition probability under policy $\pi$, where its $\big((s', a'), (s, a)\big)$-th entry is the probability of transiting from the state-action pair $(s, a)$ to $(s', a')$ in the next period, i.e.,

$$\mathbb{P}^\pi\big((s', a'), (s, a)\big) := \mathbb{P}(s'|s, a)\pi(a'|s').$$

As the occupancy measure $\lambda^\pi$ is defined as the discounted expected number of times that the agent will visit a particular state-action pair under policy $\pi$, we can represent $\lambda^\pi$ as

$$\lambda^\pi = \rho^\pi + \gamma \mathbb{P}^\pi \rho^\pi + (\gamma \mathbb{P}^\pi)^2 \rho^\pi + \cdots = \left[ \sum_{k=0}^\infty (\gamma \mathbb{P}^\pi)^k \right] \rho^\pi = (1 - \gamma \mathbb{P}^\pi)^{-1} \rho^\pi.$$

Thus, the difference $\left\| \lambda^{\hat{\pi}_\theta} - \lambda^{\pi_\theta} \right\|_1$ is equal to

$$\left\| \lambda_i^{\hat{\pi}_\theta} - \lambda_i^{\pi_\theta} \right\|_1 = \left\| (1 - \gamma \mathbb{P}^{\hat{\pi}_\theta})^{-1} \rho^{\hat{\pi}_\theta} - (1 - \gamma \mathbb{P}^{\pi_\theta})^{-1} \rho^{\pi_\theta} \right\|_1$$

$$= \left\| (1 - \gamma \mathbb{P}^{\hat{\pi}_\theta})^{-1} \rho^{\hat{\pi}_\theta} - (1 - \gamma \mathbb{P}^{\hat{\pi}_\theta})^{-1} \rho^{\pi_\theta} + (1 - \gamma \mathbb{P}^{\hat{\pi}_\theta})^{-1} \rho^{\pi_\theta} - (1 - \gamma \mathbb{P}^{\pi_\theta})^{-1} \rho^{\pi_\theta} \right\|_1$$

$$\leq \left\| (1 - \gamma \mathbb{P}^{\hat{\pi}_\theta})^{-1} \rho^{\hat{\pi}_\theta} - (1 - \gamma \mathbb{P}^{\hat{\pi}_\theta})^{-1} \rho^{\pi_\theta} \right\|_1 + \left\| (1 - \gamma \mathbb{P}^{\hat{\pi}_\theta})^{-1} \rho^{\pi_\theta} - (1 - \gamma \mathbb{P}^{\pi_\theta})^{-1} \rho^{\pi_\theta} \right\|_1$$

$$\leq \left\| (1 - \gamma \mathbb{P}^{\hat{\pi}_\theta})^{-1} \right\|_1 \left\| \rho^{\hat{\pi}_\theta} - \rho^{\pi_\theta} \right\|_1 + \left\| \left[ (1 - \gamma \mathbb{P}^{\hat{\pi}_\theta})^{-1} - (1 - \gamma \mathbb{P}^{\pi_\theta})^{-1} \right] \rho^{\pi_\theta} \right\|_1,$$

$$(35)$$

where $\|A\|_1 := \sup_x \frac{\|Ax\|_1}{\|x\|_1} = \sup_{\|x\|=1} \|Ax\|_1$ is the induced 1-norm for matrices and the last line follows from the norm inequality $\|Ax\|_1 \leq \|A\|_1 \|x\|_1$. Below, we separately bound the terms that appear on the right-hand side of (35). Note that the induced one norm is indeed the maximum absolute column sum of the matrix, i.e., $\|A\|_1 = \max_j \sum_i |a_{ij}|$. Then, for any policy $\pi_\theta$, since $\mathbb{P}^{\pi_\theta}$ is the transition matrix and has its column sum equal to 1, it holds that

$$\left\| (1 - \gamma \mathbb{P}^{\pi_\theta})^{-1} \right\|_1 = \sup_{(s,a) \in \mathcal{S} \times \mathcal{A}} \sum_{(s', a') \in \mathcal{S} \times \mathcal{A}} (1 - \gamma \mathbb{P}^{\pi_\theta})^{-1} \big((s', a'), (s, a)\big) = \sum_{k=0}^\infty \gamma^k = \frac{1}{1 - \gamma}.$$

Thus, by utilizing the definition of $\rho^\pi(s,a) = \rho(s)\pi(a|s)$, we have that

$$\left\|(1-\gamma\mathbb{P}^{\hat{\pi}_\theta})^{-1}\right\|_1 \left\|\rho^{\hat{\pi}_\theta} - \rho^{\pi_\theta}\right\|_1 = \frac{1}{1-\gamma}\left\|\rho^{\hat{\pi}_\theta} - \rho^{\pi_\theta}\right\|_1$$

$$= \frac{1}{1-\gamma}\sum_{s\in\mathcal{S}}\sum_{a\in\mathcal{A}}|\rho(s)\hat{\pi}_\theta(a|s) - \rho(s)\pi_\theta(a|s)|$$

$$= \frac{1}{1-\gamma}\sum_{s\in\mathcal{S}}\rho(s)\sum_{a\in\mathcal{A}}|\hat{\pi}_\theta(a|s) - \pi_\theta(a|s)| \qquad (36)$$

$$\leq \frac{1}{1-\gamma}\max_{s\in\mathcal{S}}\|\hat{\pi}_\theta(\cdot|s) - \pi_\theta(\cdot|s)\|_1$$

$$\leq \frac{nc\phi^\kappa}{1-\gamma},$$

where the first inequality holds since $\rho(\cdot)$ is a distribution. To bound the second term in (35), we can first derive that

$$\left\|\left[(1-\gamma\mathbb{P}^{\hat{\pi}_\theta})^{-1} - (1-\gamma\mathbb{P}^{\pi_\theta})^{-1}\right]\rho^{\pi_\theta}\right\|_1$$

$$= \left\|\left[(1-\gamma\mathbb{P}^{\pi_\theta})^{-1}\cdot\left[(1-\gamma\mathbb{P}^{\pi_\theta}) - (1-\gamma\mathbb{P}^{\hat{\pi}_\theta})\right]\cdot(1-\gamma\mathbb{P}^{\hat{\pi}_\theta})^{-1}\right]\rho^{\pi_\theta}\right\|_1$$

$$= \gamma\left\|\left[(1-\gamma\mathbb{P}^{\pi_\theta})^{-1}\cdot\left[\mathbb{P}^{\hat{\pi}_\theta} - \mathbb{P}^{\pi_\theta}\right]\cdot(1-\gamma\mathbb{P}^{\hat{\pi}_\theta})^{-1}\right]\rho^{\pi_\theta}\right\|_1 \qquad (37)$$

$$\leq \gamma\left\|(1-\gamma\mathbb{P}^{\pi_\theta})^{-1}\right\|_1\left\|\mathbb{P}^{\hat{\pi}_\theta} - \mathbb{P}^{\pi_\theta}\right\|_1\left\|(1-\gamma\mathbb{P}^{\hat{\pi}_\theta})^{-1}\right\|_1\left\|\rho^{\pi_\theta}\right\|_1$$

$$= \frac{\gamma}{(1-\gamma)^2}\left\|\mathbb{P}^{\hat{\pi}_\theta} - \mathbb{P}^{\pi_\theta}\right\|_1,$$

where we apply the norm equality again and use the fact that $\left\|\rho^{\pi_\theta}\right\|_1 = 1$. The term $\left\|\mathbb{P}^{\hat{\pi}_\theta} - \mathbb{P}^{\pi_\theta}\right\|_1$ can be further upper-bounded as follows

$$\left\|\mathbb{P}^{\hat{\pi}_\theta} - \mathbb{P}^{\pi_\theta}\right\|_1 = \sup_{(s,a)\in\mathcal{S}\times\mathcal{A}}\sum_{(s',a')\in\mathcal{S}\times\mathcal{A}}\left|\mathbb{P}^{\hat{\pi}_\theta}\big((s',a'),(s,a)\big) - \mathbb{P}^{\pi_\theta}\big((s',a'),(s,a)\big)\right|$$

$$= \sup_{(s,a)\in\mathcal{S}\times\mathcal{A}}\sum_{(s',a')\in\mathcal{S}\times\mathcal{A}}\left|\mathbb{P}(s'|s,a)\hat{\pi}_\theta(a'|s') - \mathbb{P}(s'|s,a)\pi_\theta(a'|s')\right|$$

$$= \sup_{(s,a)\in\mathcal{S}\times\mathcal{A}}\sum_{(s',a')\in\mathcal{S}\times\mathcal{A}}\mathbb{P}(s'|s,a)\left|\hat{\pi}_\theta(a'|s') - \pi_\theta(a'|s')\right| \qquad (38)$$

$$\leq \max_{s'\in\mathcal{S}}\|\hat{\pi}_\theta(\cdot|s') - \pi_\theta(\cdot|s')\|_1$$

$$\leq nc\phi^\kappa,$$

where we used the definition of $\mathbb{P}^{\pi_\theta}$ in the second line and the fact that $\mathbb{P}(\cdot|s,a)$ is a distribution in the fourth line. By substituting inequalities (36), (37), and (38) back into (35), we conclude that

$$\left\|\lambda^{\hat{\pi}_\theta} - \lambda^{\pi_\theta}\right\|_1 \leq \left\|\left[(1-\gamma\mathbb{P}^{\hat{\pi}_\theta})^{-1} - (1-\gamma\mathbb{P}^{\pi_\theta})^{-1}\right]\rho^{\pi_\theta}\right\|_1$$

$$\leq \frac{nc\phi^\kappa}{1-\gamma} + \frac{\gamma}{(1-\gamma)^2}\cdot nc\phi^\kappa$$

$$= \frac{nc\phi^k}{(1-\gamma)^2}.$$

Finally, recall that the local occupancy measure is the marginalization of the global occupancy measure (see the discussion below (4)). Therefore, for every agent $i\in\mathcal{N}$, it holds that

$$\left\|\lambda_i^{\hat{\pi}_\theta} - \lambda_i^{\pi_\theta}\right\|_1 = \sum_{(s_i,a_i)\in\mathcal{S}_i\times\mathcal{A}_i}\left|\lambda_i^{\hat{\pi}_\theta}(s_i,a_i) - \lambda_i^{\pi_\theta}(s_i,a_i)\right|$$

$$= \sum_{(s_i,a_i)\in\mathcal{S}_i\times\mathcal{A}_i}\left|\sum_{(s_{-i},a_{-i})\in\mathcal{S}_{-i}\times\mathcal{A}_{-i}}\lambda^{\hat{\pi}_\theta}(s,a) - \sum_{(s_{-i},a_{-i})\in\mathcal{S}_{-i}\times\mathcal{A}_{-i}}\lambda^{\pi_\theta}(s,a)\right|$$

$$\leq \sum_{(s,a)\in\mathcal{S}\times\mathcal{A}}\left|\lambda^{\hat{\pi}_\theta}(s,a) - \lambda^{\pi_\theta}(s,a)\right|$$

$$= \left\|\lambda^{\hat{\pi}_\theta} - \lambda^{\pi_\theta}\right\|_1$$

$$\leq \frac{nc\phi^k}{(1-\gamma)^2},$$

where the first inequality follows from the triangular inequality. This completes the proof. □

## E.2 Proof of Lemma 3.5

Firstly, when $\left\| r_{\diamond_i}^{\pi_\theta} \right\|_\infty \le M_r$ for every $\diamond \in \{f, g\}$ and $\theta \in \Theta$, it follows from [34, Proposition 6] that the shadow Q-functions satisfy the so-called *exponential decay property*. Specifically, for every $\theta \in \Theta$, agent $i \in \mathcal{N}$, and state-action pairs $(s, a), (s', a') \in \mathcal{S} \times \mathcal{A}$ such that $s_{\mathcal{N}_i^\kappa} = s'_{\mathcal{N}_i^\kappa}$, $a_{\mathcal{N}_i^\kappa} = a'_{\mathcal{N}_i^\kappa}$, it holds that

$$\left| Q_{\diamond_i}^{\pi_\theta}(s, a) - Q_{\diamond_i}^{\pi_\theta}(s', a') \right| \le c_0 \phi_0^\kappa, \ \forall \diamond \in \{f, g\}, \tag{39}$$

where $(c_0, \phi_0) = \left( \frac{2\gamma \chi M_r}{2 - \gamma \chi}, e^{-\omega} \right)$. Then, it is clear from the definition of the truncated Q-function that

$$\sup_{s,a} \left| \widehat{Q}_{\diamond_i}^{\pi_\theta}(s_{\mathcal{N}_i^\kappa}, a_{\mathcal{N}_i^\kappa}) - Q_{\diamond_i}^{\pi_\theta}(s, a) \right| \le c_0 \phi_0^\kappa, \ \forall \theta \in \Theta, i \in \mathcal{N}. \tag{40}$$

In the proof below, we write the expectation $\mathbb{E}_{s \sim d^{\pi_\theta}, a \sim \pi_\theta(\cdot|s)}$ simply as $\mathbb{E}^{\pi_\theta}$ to reduce the burden of notations. By the definitions of the truncated policy gradient in (15) and the true policy gradient with $\kappa$-hop policies in (10), we have that

$$n(1 - \gamma) \left[ \widehat{\nabla}_{\theta_i} \mathcal{L}(\theta, \mu) - \nabla_{\theta_i} \mathcal{L}(\theta, \mu) \right]$$

$$= \mathbb{E}^{\pi_\theta} \left[ \nabla_{\theta_i} \log \pi_{\theta_i}^i (a_i | s_{\mathcal{N}_i^\kappa}) \left[ \sum_{j \in \mathcal{N}_i^\kappa} \left( \widehat{Q}_{f_j}^{\pi_\theta}(s_{\mathcal{N}_j^\kappa}, a_{\mathcal{N}_j^\kappa}) + \mu_j \widehat{Q}_{g_j}^{\pi_\theta}(s_{\mathcal{N}_j^\kappa}, a_{\mathcal{N}_j^\kappa}) \right) - \sum_{j \in \mathcal{N}} \left( Q_{f_j}^{\pi_\theta}(s, a) + \mu_j Q_{g_j}^{\pi_\theta}(s, a) \right) \right] \right]$$

$$= \underbrace{\mathbb{E}^{\pi_\theta} \left[ \nabla_{\theta_i} \log \pi_{\theta_i}^i (a_i | s_{\mathcal{N}_i^\kappa}) \cdot \left[ \sum_{j \in \mathcal{N}} \left( \widehat{Q}_{f_j}^{\pi_\theta}(s_{\mathcal{N}_j^\kappa}, a_{\mathcal{N}_j^\kappa}) - Q_{f_j}^{\pi_\theta}(s, a) \right) + \mu_j \left( \widehat{Q}_{g_j}^{\pi_\theta}(s_{\mathcal{N}_j^\kappa}, a_{\mathcal{N}_j^\kappa}) - Q_{g_j}^{\pi_\theta}(s, a) \right) \right] \right]}_{\mathcal{T}_1}$$

$$\underbrace{- \mathbb{E}^{\pi_\theta} \left[ \nabla_{\theta_i} \log \pi_{\theta_i}^i (a_i | s_{\mathcal{N}_i^\kappa}) \cdot \sum_{j \in \mathcal{N}_{-i}^\kappa} \left( \widehat{Q}_{f_j}^{\pi_\theta}(s_{\mathcal{N}_j^\kappa}, a_{\mathcal{N}_j^\kappa}) + \mu_j \widehat{Q}_{g_j}^{\pi_\theta}(s_{\mathcal{N}_j^\kappa}, a_{\mathcal{N}_j^\kappa}) \right) \right]}_{\mathcal{T}_2},$$

$$\tag{41}$$

where we add and subtract the truncated shadow Q-functions of agents in $\mathcal{N}_{-i}^\kappa$ in the second equality. Below, we first show that the term $\mathcal{T}_2$ is actually equal to 0. For any given state $s \in \mathcal{S}$, one can write

$$\mathbb{E}_{a \sim \pi_\theta(\cdot|s)} \left[ \nabla_{\theta_i} \log \pi_{\theta_i}^i (a_i | s_{\mathcal{N}_i^\kappa}) \cdot \sum_{j \in \mathcal{N}_{-i}^\kappa} \left( \widehat{Q}_{f_j}^{\pi_\theta}(s_{\mathcal{N}_j^\kappa}, a_{\mathcal{N}_j^\kappa}) + \mu_j \widehat{Q}_{g_j}^{\pi_\theta}(s_{\mathcal{N}_j^\kappa}, a_{\mathcal{N}_j^\kappa}) \right) \right]$$

$$= \sum_{a \in \mathcal{A}} \pi_\theta(a|s) \cdot \frac{\nabla_{\theta_i} \pi_{\theta_i}^i (a_i|s)}{\pi_{\theta_i}^i (a_i|s)} \cdot \left[ \sum_{j \in \mathcal{N}_{-i}^\kappa} \left( \widehat{Q}_{f_j}^{\pi_\theta}(s_{\mathcal{N}_j^\kappa}, a_{\mathcal{N}_j^\kappa}) + \mu_j \widehat{Q}_{g_j}^{\pi_\theta}(s_{\mathcal{N}_j^\kappa}, a_{\mathcal{N}_j^\kappa}) \right) \right]$$

$$= \sum_{a \in \mathcal{A}} \left( \prod_{k \in \mathcal{N}} \pi_{\theta_k}^k (a_k|s) \right) \cdot \frac{\nabla_{\theta_i} \pi_{\theta_i}^i (a_i|s)}{\pi_{\theta_i}^i (a_i|s)} \cdot \left[ \sum_{j \in \mathcal{N}_{-i}^\kappa} \left( \widehat{Q}_{f_j}^{\pi_\theta}(s_{\mathcal{N}_j^\kappa}, a_{\mathcal{N}_j^\kappa}) + \mu_j \widehat{Q}_{g_j}^{\pi_\theta}(s_{\mathcal{N}_j^\kappa}, a_{\mathcal{N}_j^\kappa}) \right) \right]$$

$$= \sum_{a \in \mathcal{A}} \left( \prod_{k \in \mathcal{N} \backslash \{i\}} \pi_{\theta_k}^k (a_k|s) \right) \cdot \nabla_{\theta_i} \pi_{\theta_i}^i (a_i|s) \cdot \left[ \sum_{j \in \mathcal{N}_{-i}^\kappa} \left( \widehat{Q}_{f_j}^{\pi_\theta}(s_{\mathcal{N}_j^\kappa}, a_{\mathcal{N}_j^\kappa}) + \mu_j \widehat{Q}_{g_j}^{\pi_\theta}(s_{\mathcal{N}_j^\kappa}, a_{\mathcal{N}_j^\kappa}) \right) \right]$$

$$\overset{(\Delta)}{=} \left[ \sum_{a_i \in \mathcal{A}_i} \nabla_{\theta_i} \pi_{\theta_i}^i (a_i|s) \right] \underbrace{\sum_{a_{-i} \in \mathcal{A}_{-i}} \left( \prod_{k \in \mathcal{N} \backslash \{i\}} \pi_{\theta_k}^k (a_k|s) \right) \cdot \left[ \sum_{j \in \mathcal{N}_{-i}^\kappa} \left( \widehat{Q}_{f_j}^{\pi_\theta}(s_{\mathcal{N}_j^\kappa}, a_{\mathcal{N}_j^\kappa}) + \mu_j \widehat{Q}_{g_j}^{\pi_\theta}(s_{\mathcal{N}_j^\kappa}, a_{\mathcal{N}_j^\kappa}) \right) \right]}_{\mathcal{T}_3}$$

$$= \mathcal{T}_3 \cdot \nabla_{\theta_i} \left[ \sum_{a_i \in \mathcal{A}_i} \pi_{\theta_i}^i (a_i|s) \right]$$

$$= \mathcal{T}_3 \cdot \nabla_{\theta_i} 1$$

$$= 0,$$

$$\tag{42}$$

where we expand the summation $\sum_{a \in \mathcal{A}}$ to $\sum_{a_i \in \mathcal{A}_i} \sum_{a_{-i} \in \mathcal{A}_{-i}}$ in equality ($\Delta$). Since $j \in \mathcal{N}_{-i}^{\kappa}$ means that agent $j$ is not in the $\kappa$-hop neighborhood of agent $i$, which further implies that $i \notin \mathcal{N}_j^{\kappa}$, we know that the term $\mathcal{T}_3$ is not relevant to agent $i$. Thus, the expansion in ($\Delta$) is justified. The last two lines follow from the facts that $\pi_{\theta_i}^i(\cdot|s)$ is a distribution over $\mathcal{A}_i$ and the gradient of a constant is equal to 0.

Thus, it suffices to bound the term $\mathcal{T}_1$ only. Using the exponential decay property of shadow Q-functions, we can derive from (41) that

$$n(1-\gamma) \left\| \widehat{\nabla}_{\theta_i} \mathcal{L}(\theta, \mu) - \nabla_{\theta_i} \mathcal{L}(\theta, \mu) \right\|_2$$

$$= \left\| \mathbb{E}^{\pi_\theta} \left[ \nabla_{\theta_i} \log \pi_{\theta_i}^i(a_i|s_{\mathcal{N}_i^\kappa}) \cdot \left[ \sum_{j \in \mathcal{N}} \left( \widehat{Q}_{f_j}^{\pi_\theta}(s_{\mathcal{N}_j^\kappa}, a_{\mathcal{N}_j^\kappa}) - Q_{f_j}^{\pi_\theta}(s,a) \right) + \mu_j \left( \widehat{Q}_{g_j}^{\pi_\theta}(s_{\mathcal{N}_j^\kappa}, a_{\mathcal{N}_j^\kappa}) - Q_{g_j}^{\pi_\theta}(s,a) \right) \right] \right] \right\|_2$$

$$\leq \mathbb{E}^{\pi_\theta} \left[ \left\| \nabla_{\theta_i} \log \pi_{\theta_i}^i(a_i|s_{\mathcal{N}_i^\kappa}) \right\|_2 \cdot \left[ \sum_{j \in \mathcal{N}} \left\| \widehat{Q}_{f_j}^{\pi_\theta}(s_{\mathcal{N}_j^\kappa}, a_{\mathcal{N}_j^\kappa}) - Q_{f_j}^{\pi_\theta}(s,a) \right\|_2 \right. \right.$$

$$\left. \left. + |\mu_j| \left\| \widehat{Q}_{g_j}^{\pi_\theta}(s_{\mathcal{N}_j^\kappa}, a_{\mathcal{N}_j^\kappa}) - Q_{g_j}^{\pi_\theta}(s,a) \right\|_2 \right] \right]$$

$$\leq M_\pi \cdot \max_{(s,a) \in \mathcal{S} \times \mathcal{A}} \left[ \sum_{j \in \mathcal{N}} \left\| \widehat{Q}_{f_j}^{\pi_\theta}(s_{\mathcal{N}_j^\kappa}, a_{\mathcal{N}_j^\kappa}) - Q_{f_j}^{\pi_\theta}(s,a) \right\|_2 + |\mu_j| \left\| \widehat{Q}_{g_j}^{\pi_\theta}(s_{\mathcal{N}_j^\kappa}, a_{\mathcal{N}_j^\kappa}) - Q_{g_j}^{\pi_\theta}(s,a) \right\|_2 \right]$$

$$\leq M_\pi \left[ n \cdot c_0 \phi_0^\kappa + n \|\mu\|_\infty c_0 \phi_0^\kappa \right]$$

$$= M_\pi \cdot (1 + \|\mu\|_\infty) \cdot n c_0 \phi_0^\kappa,$$

where we use the assumption $\left\| \nabla_{\theta_i} \log \pi_{\theta_i}^i \right\|_2 \leq M_\pi$ in the second inequality. Thus, we conclude that

$$\left\| \widehat{\nabla}_{\theta_i} \mathcal{L}(\theta, \mu) - \nabla_{\theta_i} \mathcal{L}(\theta, \mu) \right\|_2 \leq \frac{(1 + \|\mu\|_\infty) M_\pi c_0 \phi_0^\kappa}{1 - \gamma},$$

which completes the proof.

# F    Supplementary materials for Section 4

In this appendix, we first provide a detailed explanation for the assumptions used Section 4. We then present a summary of the problem's properties under these assumptions in Appendix (F.1.5).

## F.1    Discussions about assumptions

Besides the boundedness of score functions, Assumptions 4.1 and 4.2 require that $f_i(\lambda_i^{\pi_\theta})$ and $g_i(\lambda_i^{\pi_\theta})$ be smooth w.r.t. both the local occupancy measure $\lambda_i^{\pi_\theta}$ and the parameter $\theta$. These assumptions are standard in the literature of reinforcement learning with general utilities [14, 23, 71, 61]. Assumption 4.3 mainly ensures the existence an FOSP within the search region. When Slater's condition is met and the general utilities are concave in the occupancy measure, Assumption 4.3 is naturally satisfied since the strong duality holds and the optimal dual variable is bounded (see Lemma F.3).

### F.1.1    Discussion about Assumption 4.1

Let $\Lambda$ be the set of all possible (global) occupancy measures. It is well-known that $\Lambda$ is a convex polytope [78] and can be defined as:

$$\Lambda := \left\{ \lambda \in \mathbb{R}^{|S||A|} \,\middle|\, \lambda \geq 0, \sum_{a \in \mathcal{A}} \lambda(s,a) = (1-\gamma) \cdot \rho(s) + \gamma \sum_{(s',a') \in \mathcal{S} \times \mathcal{A}} \mathbb{P}(s|s',a') \cdot \lambda(s',a'), \forall s \in S \right\},$$
(43)

where $\rho(\cdot)$ is the initial distribution. Since $\Lambda$ is a compact set, and if the general utilities $f_i(\cdot)$ and $g_i(\cdot)$ are twice continuously differentiable, the smoothness property required by Assumption 4.1 naturally holds on $\Lambda$.

### F.1.2 Discussion about Assumption 4.2

The assumption on the boundedness of the score function is standard in the study of RL with/without general utilities [79, 71, 23, 10, 43]. Specifically, this assumption is essential in quantifying the approximation error of REINFORCE-based gradient estimators [44]. Similarly, the assumption of the smoothness of general utilities with respect to the policy parameter is common in the literature [17, 80, 71, 23, 61]. Indeed, the following existing results show that Assumption 4.2 holds true for two classes of policies under mild conditions. For the ease of notations, we present these results in the centralized (single-agent) setting, while they naturally generalize to the distributed (multi-agent) setting.

**Proposition F.1** (Direct parameterization [61]). *Suppose that the general utility $f(\lambda)$ has a bounded and Lipschitz gradient in $\Lambda$, namely, there exist $\ell_{f,1}, \ell_{f,2} > 0$ such that*

$$\|\nabla_\lambda f(\lambda)\|_\infty \le \ell_{f,1}, \quad \|\nabla_\lambda f(\lambda) - \nabla_\lambda f(\lambda')\|_\infty \le \ell_{f,2}\|\lambda - \lambda'\|_2, \quad \forall\, \lambda, \lambda' \in \Lambda.$$

*Then, $f(\lambda^\pi)$ is $\ell_F$-smooth with respect to the policy $\pi$, where*

$$\ell_F = \frac{4\ell_{f,1}\gamma|\mathcal{A}| + \ell_{f,2}|\mathcal{A}|^{3/2}}{(1-\gamma)^2}.$$

**Proposition F.2** (General soft-max parameterization [71]). *Consider the general soft-max parameterization $\pi_\theta(\cdot|\cdot)$, defined as*

$$\pi_\theta(a|s) = \frac{\exp\{\psi(\theta; s, a)\}}{\sum_{a' \in \mathcal{A}} \exp\{\psi(\theta; s, a')\}}, \quad \forall\, (s, a) \in \mathcal{S} \times \mathcal{A}.$$

*Suppose that $\psi(\cdot; s, a)$ is twice differentiable for all $(s, a) \in \mathcal{S} \times \mathcal{A}$ and there exist $\ell_{\psi,1}, \ell_{\psi,2} > 0$ such that*

$$\max_{(s,a) \in \mathcal{S} \times \mathcal{A}} \sup_\theta \|\nabla_\theta \psi(\theta; s, a)\|_2 \le \ell_{\psi,1} \quad and \quad \max_{(s,a) \in \mathcal{S} \times \mathcal{A}} \sup_\theta \|\nabla_\theta^2 \psi(\theta; s, a)\|_2 \le \ell_{\psi,2}.$$

*Assume that $f(\lambda)$ has a bounded and Lipschitz gradient in $\Lambda$, namely, there exist $\ell_{f,1}, \ell_{f,2} > 0$ such that*

$$\|\nabla_\lambda f(\lambda)\|_\infty \le \ell_{f,1}, \quad \|\nabla_\lambda f(\lambda) - \nabla_\lambda f(\lambda')\|_\infty \le \ell_{f,2}\|\lambda - \lambda'\|_2, \quad \forall\, \lambda, \lambda' \in \Lambda.$$

*The following statements hold:*

*(I) For every $\theta \in \Theta$ and $(s, a) \in \mathcal{S} \times \mathcal{A}$, it holds that*

$$\begin{cases} \|\nabla_\theta \log \pi_\theta(a|s)\|_2 \le 2\ell_{\psi,1}, \\ \|\nabla_\theta^2 \log \pi_\theta(a|s)\|_2 \le 2\left(\ell_{\psi,2} + \ell_{\psi,1}^2\right), \end{cases} \quad and \quad \|\nabla_\theta f(\lambda^{\pi_\theta})\|_2 \le \frac{2\ell_{\psi,1} \cdot \ell_{f,1}}{(1-\gamma)^2}.$$

*(II) For every $\theta_1, \theta_2 \in \Theta$, it holds that*

$$\|\lambda^{\pi_{\theta_1}} - \lambda^{\pi_{\theta_2}}\|_1 \le \frac{2\ell_{\psi,1}}{(1-\gamma)^2} \cdot \|\theta_1 - \theta_2\|_2.$$

*(III) The function $f(\lambda^{\pi_\theta})$ is $\ell_F$-smooth with respect to the parameter $\theta$, where*

$$\ell_F = \frac{4\ell_{f,2} \cdot \ell_{\psi,1}^2}{(1-\gamma)^4} + \frac{8\ell_{\psi,1}^2 \cdot \ell_{f,1}}{(1-\gamma)^3} + \frac{2\ell_{f,1} \cdot \left(\ell_{\psi,2} + \ell_{\psi,1}^2\right)}{(1-\gamma)^2}.$$

### F.1.3 Discussion about Assumption 4.3

In constrained optimization, it is common to assume that the feasible region for the dual variable is bounded [25]. In particular, when all the utilities are concave in the occupancy measure $\lambda^\pi$, problem (5) becomes a convex program with respect to $\lambda^\pi$. Under this circumstance, if the feasible region contains an interior point, which is usually the case when no equality constraints are enforced, it can be proven that the strong duality holds and the optimal dual variable is bounded [81, 82, 61]. This assumption of having an interior point is also referred to as Slater's condition.

**Lemma F.3** (Strong duality and boundedness of the optimal dual variable [61])**.** *Consider the centralized reinforcement learning problem with general utilities*

$$\max_{\theta \in \Theta} f(\lambda^{\pi_\theta}) \quad s.t. \quad g(\lambda^{\pi_\theta}) \geq 0,$$

*where $f(\cdot)$ and $g(\cdot)$ are concave functions. Denote $\theta^\star$ and $\mu^\star$ as the optimal primal variable and dual variable, respectively. Suppose Slater's condition holds true, i.e., there exist $\widetilde{\theta} \in \Theta$ and $\xi > 0$ such that $g(\lambda^{\pi_{\widetilde{\theta}}}) \geq \xi$, and the set $\mathrm{cl}\left(\left\{\lambda^{\pi_\theta} \middle| \theta \in \Theta\right\}\right)$ is convex. Then we have:*

*(I) the strong duality holds, i.e.,*

$$f(\lambda^{\pi_{\theta^\star}}) = \mathcal{L}(\theta^\star, \mu^\star) = \max_{\theta \in \Theta} \mathcal{L}(\theta, \mu^\star),$$

*(II) the optimal dual variable is bounded, s.t.*

$$0 \leq \mu^\star \leq \frac{f(\lambda^{\pi_{\theta^\star}}) - f(\lambda^{\pi_{\widetilde{\theta}}})}{\xi}.$$

### F.1.4 Discussion about Assumption 4.5

The following proposition on the effectiveness of Algorithm 2 is adapted from [10].

**Proposition F.4** (Sample complexity of Algorithm 2 [10])**.** *Suppose that there exists positive integer $k_0$ and $\sigma \in (0, 1)$ such that for any policy $\pi_\theta$ and any initial state-action pair $(s, a) \in \mathcal{S} \times \mathcal{A}$, it holds that*

$$\mathbb{P}\left(\left(s^{k_0}_{\mathcal{N}_i^\kappa}, a^{k_0}_{\mathcal{N}_i^\kappa}\right) = \left(s'_{\mathcal{N}_i^\kappa}, a'_{\mathcal{N}_i^\kappa}\right) \middle| \left(s^0, a^0\right) = (s, a)\right) \geq \sigma, \ \forall \left(s'_{\mathcal{N}_i^\kappa}, a'_{\mathcal{N}_i^\kappa}\right) \in \mathcal{S}_{\mathcal{N}_i^\kappa} \times \mathcal{A}_{\mathcal{N}_i^\kappa}, i \in \mathcal{N}. \quad (44)$$

*Let $k_1$ and $h$ be two numbers such that $h \geq 1/\sigma \cdot \max\left\{2, 1/(1 - \sqrt{\gamma})\right\}$ and $k_1 \geq \max\{2h, 4\sigma h, k_0\}$. For given local shadow rewards $\{r_i\}_{i \in \mathcal{N}}$ such that $\max_{i \in \mathcal{N}} \|r_i\|_\infty \leq M_r$, denote $\left\{\widehat{Q}_i^{\pi_\theta}\right\}_{i \in \mathcal{N}}$ as the true truncated Q-functions and $\left\{\widetilde{Q}_i^K\right\}_{i \in \mathcal{N}}$ as the empirical truncated Q-functions output by Algorithm 2 with step-sizes $\left\{\eta_Q^k = h/(k + k_1)\right\}_{k=0}^{K-1}$. For each agent $i \in \mathcal{N}$, with probability at least $1 - \delta$, it holds that*

$$\left\|\widetilde{Q}_i^K - \widehat{Q}_i^{\pi_\theta}\right\|_\infty \leq \frac{C_i}{\sqrt{K + k_1}} + \frac{C_i'}{K + k_1}, \quad (45)$$

*where*

$$\begin{aligned} C_i &= \frac{6\bar{\epsilon}}{1 - \sqrt{\gamma}} \sqrt{\frac{hk_0}{\sigma} \left[\log\left(\frac{2k_0 K^2}{\delta}\right) + |\mathcal{N}_i^\kappa| \log\left(|\mathcal{S}_i|\|\mathcal{A}_i|\right)\right]} \\ C_i' &= \frac{2}{1 - \sqrt{\gamma}} \max\left(\frac{16\bar{\epsilon}hk_0}{\sigma}, \frac{2M_r}{1 - \gamma}(k_0 + k_1)\right), \end{aligned} \quad (46)$$

*with $\bar{\epsilon} = 4M_r/(1 - \gamma) + 2M_r$.*

The condition (44) requires every state-action pair in the $\kappa$-hop neighborhood to be visited with some probability $\sigma > 0$ after some period $k_0$. Intuitively, it means that the agents can quickly explore the environment no matter what the initial distribution is. Under this assumption, Proposition (F.4) implies that the error bound described in (26) can be achieved using $\mathcal{O}(1/(\epsilon_0)^2)$ samples with high probability. We remark that, since the error term on the right-hand side of (45) only logarithmically depends on the failure probability, the probabilistic version of Assumption 4.5 can be easily adapted to the proof by applying a similar argument as the the one before (72). The same order of the sample complexity would still hold true.

Besides the TD-learning method introduced in this work, various algorithms in the literature that enjoy faster convergence rates can be used in the truncated Q-function evaluations, such as the two timescale linear TD with gradient correction (TDC) [83, 84, 85] and the nonlinear TDC [86, 87].

### F.1.5 Direct consequences of Assumptions 4.1-4.3

The following properties are the direct consequence of Assumptions 4.1-4.3.

**Lemma F.5.** *Under Assumptions 4.1-4.3, it holds that*

**(I)** *the shadow rewards are bounded, i.e., $\exists M_r > 0$ s.t. $\left\| r_{\diamond_i}^{\pi_\theta} \right\|_2 \leq M_r$, $\forall \diamond \in \{f, g\}$, $\theta \in \Theta$, $i \in \mathcal{N}$.*

**(II)** *the Lagrangian and its gradient are bounded, i.e., $\exists M_L, M_\theta > 0$ s.t. $|\mathcal{L}(\theta, \mu)| \leq M_L$, $\left\| \nabla_\theta \mathcal{L}(\theta, \mu) \right\|_2 \leq M_\theta$, $\forall \theta \in \Theta, \mu \in \mathcal{U}$.*

**(III)** $\nabla_\theta \mathcal{L}(\theta, \mu)$ *is Lipschitz continuous w.r.t. $\theta$ and $\mu$, i.e., $\exists L_{\theta\theta}, L_{\theta\mu} > 0$ s.t. $\forall \theta, \theta' \in \Theta$ and $\mu, \mu' \in \mathcal{U}$*

$$\left\| \nabla_\theta \mathcal{L}(\theta, \mu) - \nabla_\theta \mathcal{L}(\theta', \mu) \right\|_2 \leq L_{\theta\theta} \left\| \theta - \theta' \right\|_2, \quad \left\| \nabla_\theta \mathcal{L}(\theta, \mu) - \nabla_\theta \mathcal{L}(\theta, \mu') \right\|_2 \leq L_{\theta\mu} \left\| \mu - \mu' \right\|_2.$$
(47)

*Proof of (I).* By Definition 3.1, the shadow rewards are the gradients of local utility functions w.r.t. the correponding local occupancy measures, i.e., $r_{f_i}^{\pi_\theta} \coloneqq \nabla_{\lambda_i} f_i(\lambda_i^{\pi_\theta})$ and $r_{g_i}^{\pi_\theta} \coloneqq \nabla_{\lambda_i} g_i(\lambda_i^{\pi_\theta})$, $\forall i \in \mathcal{N}$. Since the local occupancy measure $\lambda_i^{\pi_\theta}$ can be expressed by the global occupancy measure through $\lambda_i^\pi(s_i, a_i) = \sum_{s_{-i}, a_{-i}} \lambda^\pi(s, a)$, we can also view $f_i(\cdot)$ and $g_i(\cdot)$ as functions of $\lambda^{\pi_\theta}$.

Recall that the set of global occupancy measures, denoted as $\Lambda$, is compact (see (43)). When Assumption 4.1 holds, $r_{f_i}^{\pi_\theta}$ and $r_{g_i}^{\pi_\theta}$ are Lipschitz continuous functions on a compact set. Thus, there $\exists M_r > 0$ such that $\left\| r_{\diamond_i}^{\pi_\theta} \right\|_2$ is universally bounded by $M_r$ $\forall \diamond \in \{f, g\}$, $i \in \mathcal{N}$. $\qquad\square$

*Proof of (II).* Similarly, since utilities functions $f_i(\cdot)$ and $g_i(\cdot)$ are assumed to be continuously differentiable w.r.t. $\lambda_i^{\pi_\theta}$, they are continuous functions on the compact set $\Lambda$. Thus, there exists $M_f > 0$ and $M_g > 0$ such that $|f_i(\cdot)| \leq M_f$ and $|g_i(\cdot)| \leq M_g$ hold for all $\lambda^{\pi_\theta} \in \Lambda$. As the feasible region region for $\mu$ is assumed to be bounded according to Assumption 4.3, we have that

$$|\mathcal{L}(\theta, \mu)| \coloneqq \left| \frac{1}{n} \sum_{i \in \mathcal{N}} \left[ f_i(\lambda_i^{\pi_\theta}) + \mu_i g_i(\lambda_i^{\pi_\theta}) \right] \right| \leq M_f + \overline{\mu} M_g =: M_L.$$

The boundedness of $\left\| \nabla_\theta \mathcal{L}(\theta, \mu) \right\|_2$ follows from the boundedness of score functions, shadow rewards, and dual variables. Similar as (10), we can write that

$$
\begin{aligned}
\left\| \nabla_\theta \mathcal{L}(\theta, \mu) \right\|_2 &= \left\| \frac{1}{1-\gamma} \mathbb{E}_{s \sim d^{\pi_\theta}, a \sim \pi_\theta(\cdot|s)} \left[ \nabla_\theta \log \pi_\theta(a|s) \cdot \frac{1}{n} \sum_{i \in \mathcal{N}} \left( Q_{f_i}^{\pi_\theta}(s, a) + \mu_i Q_{g_i}^{\pi_\theta}(s, a) \right) \right] \right\|_2 \\
&\leq \frac{1}{1-\gamma} \cdot \max_{\theta \in \Theta, (s,a) \in \mathcal{S} \times \mathcal{A}} \left\{ \left\| \nabla_\theta \log \pi_\theta(a|s) \right\|_2 \right\} \cdot \max_{\theta \in \Theta, i \in \mathcal{N}} \left\{ \frac{\left\| r_{f_i}^{\pi_\theta} \right\|_\infty + |\mu_i| \left\| r_{g_i}^{\pi_\theta} \right\|_\infty}{1-\gamma} \right\} \\
&\leq \frac{(1+\overline{\mu}) M_r}{(1-\gamma)^2} \cdot \max_{\theta \in \Theta, (s,a) \in \mathcal{S} \times \mathcal{A}} \left\{ \left\| \nabla_\theta \log \pi_\theta(a|s) \right\|_2 \right\} \\
&\overset{(\Delta)}{=} \frac{(1+\overline{\mu}) M_r}{(1-\gamma)^2} \cdot \max_{\theta \in \Theta, (s,a) \in \mathcal{S} \times \mathcal{A}} \sqrt{\sum_{i \in \mathcal{N}} \left\| \nabla_{\theta_i} \log \pi_{\theta_i}^i(a_i|s) \right\|_2^2} \\
&\leq \frac{(1+\overline{\mu}) M_r}{(1-\gamma)^2} \cdot \sqrt{n M_\pi^2} \\
&= \frac{\sqrt{n}(1+\overline{\mu}) M_r M_\pi}{(1-\gamma)^2} =: M_\theta,
\end{aligned}
$$
(48)

where the first inequality is due to $|Q^{\pi_\theta}(r; s, a)| \leq \|r\|_\infty / (1-\gamma)$ for any reward function $r(\cdot, \cdot)$. Then, the subsequent inequality follows from the norm inequality $\|x\|_\infty \leq \|x\|_2$ for any vector $x$. In equality $(\Delta)$ above, we use the fact that the global policy can be factorized as the product of local policies, so that $\nabla_\theta \log \pi_\theta(a|s) = \sum_{i \in \mathcal{N}} \nabla_\theta \log \pi_{\theta_i}^i(a_i|s_{\mathcal{N}_i^\kappa}) = \sum_{i \in \mathcal{N}} \nabla_{\theta_i} \log \pi_{\theta_i}^i(a_i|s_{\mathcal{N}_i^\kappa})$. Thus, $\nabla_\theta \log \pi_\theta(a|s)$ can be viewed as the concatenation of $\nabla_{\theta_i} \log \pi_{\theta_i}^i(a_i|s_{\mathcal{N}_i^\kappa})$ for all $i \in \mathcal{N}$, which implies $(\Delta)$. $\qquad\square$

*Proof of (III).* By Assumption 4.2, the general utilities $F(\theta) = f(\lambda^{\pi_\theta})$ and $G_i(\theta) = g_i(\lambda_i^{\pi_\theta})$ are $L_\theta$-smooth w.r.t. $\theta$. Thus, when $\mu \in \mathcal{U} = [0, \overline{u}]^n$, we have that the Lagrangian function $\mathcal{L}(\theta, \mu) = F(\theta) + 1/n \cdot \sum_{i \in \mathcal{N}} \mu_i G_i(\theta)$ is $L_{\theta\theta} \coloneqq (1 + \overline{\mu}) L_\theta$-smooth, i.e.,

$$\left\| \nabla_\theta \mathcal{L}(\theta, \mu) - \nabla_\theta \mathcal{L}(\theta', \mu) \right\|_2 \leq L_{\theta\theta} \left\| \theta - \theta' \right\|_2, \; \forall \theta, \theta' \in \Theta \text{ and } \mu \in \mathcal{U}.$$

To show the second inequality, we can write that

$$\|\nabla_\theta \mathcal{L}(\theta,\mu) - \nabla_\theta \mathcal{L}(\theta,\mu')\|_2 = \left\| \nabla_\theta \left[ F(\theta) + \frac{1}{n}\sum_{i\in\mathcal{N}} \mu_i G_i(\theta) \right] - \nabla_\theta \left[ F(\theta) + \frac{1}{n}\sum_{i\in\mathcal{N}} \mu_i' G_i(\theta) \right] \right\|_2$$

$$= \frac{1}{n} \left\| \sum_{i\in\mathcal{N}} (\mu_i - \mu_i') \nabla_\theta G_i(\theta) \right\|_2$$

$$\leq \frac{1}{n} \max_{i\in\mathcal{N},\theta\in\Theta} \{\|\nabla_\theta G_i(\theta)\|_2\} \cdot \|\mu - \mu'\|_1$$

$$\leq \frac{1}{\sqrt{n}} \max_{i\in\mathcal{N},\theta\in\Theta} \{\|\nabla_\theta G_i(\theta)\|_2\} \cdot \|\mu - \mu'\|_2,$$

where the last line follows from the norm inequality that $\|x\|_1 \leq \sqrt{n}\,\|x\|_2$ for any vector $x \in \mathbb{R}^n$. Following the same derivation as (48), one can show that

$$\max_{i\in\mathcal{N},\theta\in\Theta} \{\|\nabla_\theta G_i(\theta)\|_2\} \leq \frac{\sqrt{n}M_r M_\pi}{(1-\gamma)^2},$$

which subsequently implies that $\|\nabla_\theta \mathcal{L}(\theta,\mu) - \nabla_\theta \mathcal{L}(\theta,\mu')\|_2 \leq L_{\theta\mu} \|\mu - \mu'\|_2$ with $L_{\theta\mu} := M_r M_\pi/(1-\gamma)^2$. This completes the proof. $\qquad\square$

## F.2 Implication of metric $\mathcal{E}(\theta,\mu)$

The following lemma states the relation of the metric $\mathcal{E}(\theta,\mu)$, defined in (23), to the first-order stationary point of problem (5).

**Lemma F.6.** *Given $\theta^\star \in \Theta$ and $\mu^\star \in \mathcal{U}$, if $\mathcal{E}(\theta^\star,\mu^\star) = 0$ and $u^\star$ is in the interior of $\mathcal{U}$, then $(\theta^\star,\mu^\star)$ is a pair of first-order stationary point of problem (5).*

*Proof.* We denote $g_i(\theta^\star) := g_i(\lambda_i^{\pi_{\theta^\star}}) = n \cdot [\nabla_\mu \mathcal{L}(\theta^\star,\mu^\star)]_i$ for the ease of notation. The first-order optimality condition for problem (5) is

$$\langle \nabla_\theta \mathcal{L}(\theta^\star,\mu^\star), \theta' - \theta^\star \rangle \leq 0, \forall \theta' \in \Theta, \tag{49a}$$

$$g_i(\theta^\star) \geq 0, \forall i \in \mathcal{N}, \tag{49b}$$

$$g_i(\theta^\star)\mu_i^\star = 0, \forall i \in \mathcal{N}, \tag{49c}$$

$$\mu_i^\star \geq 0, \forall i \in \mathcal{N}. \tag{49d}$$

By reformulation, we observe that (49a) is equivalent to

$$\max_{\theta'\in\Theta, \|\theta'-\theta^\star\|_2\leq 1} \langle \nabla_\theta \mathcal{L}(\theta^\star,\mu^\star), \theta' - \theta^\star \rangle = 0 \tag{50}$$

Then, we use a contradictory argument to show that (49b) and (49c) are implied by the equality $\min_{\mu'\in\mathcal{U}, \|\mu'-\mu^\star\|_2\leq 1} \langle \nabla_\mu \mathcal{L}(\theta^\star,\mu^\star), \mu' - \mu \rangle = 0$.

Firstly, if there exists an index $i$ such that $g_i(\theta^\star) < 0$, since $\mu^\star$ is in the interior of $\mathcal{U}$, there must exist some $\mu' \in \mathcal{U}$ with $\|\mu' - \mu\| \leq 1$ and $\mu_i' > \mu_i^\star$ as well as $\mu_j' = \mu_j^\star$ for all $i \neq j$ such that $[\nabla_\mu \mathcal{L}(\theta,\mu)]_i (\mu_i' - \mu_i) = g_i(\theta^\star)(\mu_i' - \mu_i^\star)/n < 0$. Then, we have that

$$\min_{\mu'\in\mathcal{U}, \|\mu'-\mu^\star\|_2\leq 1} \langle \nabla_\mu \mathcal{L}(\theta^\star,\mu^\star), \mu' - \mu \rangle \cdot n \leq g_i(\theta^\star)(\mu_i' - \mu_i^\star) + \sum_{j\neq i} g_j(\theta^\star)(\mu_j^\star - \mu_j^\star) < 0, \tag{51}$$

which violates the condition that $\mathcal{Y}(\theta^\star,\mu^\star) = 0$. Thus, it holds that $g_i(\theta^\star) \geq 0$ for all $i$.

Furthermore, if there exists an index $i$ such that $g_i(\theta^\star) > 0$ and $\mu_i^\star > 0$, then there must exist some $\mu' \in \mathcal{U}$, with $\|\mu' - \mu\| \leq 1$ and $0 \leq \mu_i' < \mu_i^\star$ such that $[\nabla_\mu \mathcal{L}(\theta,\mu)]_i (\mu_i' - \mu_i) = g_i(\theta^\star)(\mu_i' - \mu_i^\star)/n < 0$. By a similar argument as (51), we conclude that the condition $\mathcal{Y}(\theta^\star,\mu^\star) = 0$ is also violated. Thus, it holds that $g_i(\theta^\star)\mu_i = 0$ for all $i \in \mathcal{N}$. $\qquad\square$

# G Proof of Theorems 4.4 and 4.6

Before proving the main theorems, we first quantify the approximation errors of the estimators $\widetilde{\lambda}_i^t, \widetilde{r}_{\diamond_i}^t$, $\widetilde{Q}_{\diamond_i}^t, \widetilde{\nabla}_{\mu_i}\mathcal{L}(\theta^t, \mu^t)$, and $\widetilde{\nabla}_{\theta_i}\mathcal{L}(\theta^t, \mu^{t+1})$, which are computed in Algorithm 1 in the sampled-based setting. The results are summarized in the proposition below, whose proof can be found in Appendix G.1.

**Proposition G.1.** *Suppose that Assumptions 3.2, 3.3, 4.1-4.5 hold. Let $\delta_0 \in (0, 1/(2n))$ be the failure probability. Denote $\widetilde{\nabla}_\theta \mathcal{L}(\theta^t, \mu^t)$ and $\widetilde{\nabla}_\mu \mathcal{L}(\theta^t, \mu^t)$ as the concatenations of gradient estimators $\left\{\widetilde{\nabla}_{\theta_i}\mathcal{L}(\theta^t, \mu^t)\right\}_{i\in\mathcal{N}}$ and $\left\{\widetilde{\nabla}_\mu \mathcal{L}(\theta^t, \mu^t)\right\}_{\in\mathcal{N}}$, respectively. Then, for every period $t \geq 0$ in Algorithm 1, the following inequalities hold with probability at least $1 - 2n\delta_0$:*

*(Occupancy measures):* $\left\|\widetilde{\lambda}_i^t - \lambda_i^{\pi_{\theta^t}}\right\|_2 \leq \epsilon_1(\delta_0), \ \forall i \in \mathcal{N}$ (52a)

*(Shadow rewards):* $\left\|\widetilde{r}_{\diamond_i}^t - r_{\diamond_i}^{\pi_{\theta^t}}\right\|_\infty \leq L_\lambda \epsilon_1(\delta_0), \ \forall \diamond \in \{f, g\}, i \in \mathcal{N}.$ (52b)

*(Truncated Q-functions):* $\left\|\widetilde{Q}_{\diamond_i}^t - \widehat{Q}_{\diamond_i}^{\pi_{\theta^t}}\right\|_\infty \leq M_r \epsilon_0 + \dfrac{L_\lambda \epsilon_1(\delta_0)}{1-\gamma}, \ \forall \diamond \in \{f, g\}, i \in \mathcal{N}.$ (52c)

*(Dual gradient):* $\left\|\widetilde{\nabla}_\mu \mathcal{L}(\theta^t, \mu^t) - \nabla_\mu \mathcal{L}(\theta^t, \mu^t)\right\|_2^2 \leq \dfrac{(M_r \epsilon_1(\delta_0))^2}{n} =: \epsilon_\mu$ (52d)

*(Policy gradient):* $\left\|\widetilde{\nabla}_\theta \mathcal{L}(\theta^t, \mu^{t+1}) - \nabla_\theta \mathcal{L}(\theta^t, \mu^{t+1})\right\|_2^2 \leq \left(\sum_{i\in\mathcal{N}} \dfrac{|\mathcal{N}_i^\kappa|^2}{n^2}\right)\epsilon_2(\delta_0) + n\epsilon_3 =: \epsilon_\theta.$

(52e)

*where the constant $M_r$ is defined in Lemma F.5 and*

$$
\begin{aligned}
\epsilon_1(\delta_0) &:= \sqrt{\dfrac{4 + 2\gamma^{2H}B - 16\log\delta_0}{(1-\gamma)^2 B}} \\
\epsilon_2(\delta_0) &:= 4\left[\dfrac{(1+\overline{\mu})M_r M_\pi}{(1-\gamma)^2}\right]^2 \cdot \left[\left((1-\gamma)\epsilon_0 + \dfrac{L_\lambda \epsilon_1(\delta_0)}{M_r}\right)^2 + \dfrac{2 - 8\log\delta_0}{B} + \gamma^{2H}\right] \\
&= \mathcal{O}\left(\epsilon_0^2 + \dfrac{\log(1/\delta_0)}{B} + \gamma^{2H}\right) \\
\epsilon_3 &:= 4\left[\dfrac{(1+\overline{\mu})M_\pi c_0\phi_0^\kappa}{1-\gamma}\right]^2 = \mathcal{O}\left(\phi_0^{2\kappa}\right).
\end{aligned}
$$

(53)

**Remark G.2** (Exact setting). *Consider the exact setting where the agents have accurate estimates of their local occupancy measures, shadow Q-functions, and truncated policy gradients. In this case, it is evident that the error bounds (52d) and (52e) always hold with $\epsilon_\mu = 0$ and $\epsilon_\theta = n\epsilon_3$, where $\epsilon_3$, as defined in (53), represents the truncation error of the policy gradient.*

**Remark G.3** (Truncation error). *As stated in (52e), the error of the policy gradient estimator, $\epsilon_\theta$, is composed of two parts. The second part, $n\epsilon_3$, arises from the use of truncated Q-functions and truncated policy gradients. It it important to note that this error has the factor $n$ because we assume that the norm of each agent $i$'s local score function, $\left\|\nabla_{\theta_i} \log \pi_{\theta_i}^i(\cdot|\cdot)\right\|_2$, is individually bounded by the constant $M_\pi$. If we instead assume a constant upper bound on the norm of the global score function $\nabla_\theta \log \pi_\theta(\cdot|\cdot)$, then the factor $n$ would not be present (as in [10]).*

With the shorthand notations $\widetilde{\nabla}_\theta \mathcal{L}(\theta^t, \mu^t)$ and $\widetilde{\nabla}_\mu \mathcal{L}(\theta^t, \mu^t)$, we can express the updates in Algorithm 1 as

$$
\begin{cases}
\mu^{t+1} = \mathcal{P}_\mathcal{U}\left(-\eta_\mu \widetilde{\nabla}_\mu \mathcal{L}\left(\theta^t, \mu^t\right)\right) \\[2mm]
\theta^{t+1} = \mathcal{P}_\Theta\left(\theta^t + \eta_\theta \cdot \widetilde{\nabla}_\theta \mathcal{L}\left(\theta^t, \mu^{t+1}\right)\right)
\end{cases}
, \text{ for } t = 0, 1, 2, \ldots.
$$

(54)

Recall that the exact dual variable update rule is given by (22). For ease of the notation, we define $\mathcal{L}^t(\mu)$ as the exact objective function in sub-problem (22) and $\widetilde{\mathcal{L}}^t(\mu)$ as the empirical objective

function used in Algorithm 1, i.e.,

$$\mathcal{L}^t(\mu) \coloneqq \mathcal{L}(\theta^t, \mu) + \frac{1}{2\eta_\mu} \|\mu\|_2^2, \quad \widetilde{\mathcal{L}}^t(\mu) \coloneqq \langle \widetilde{\nabla}_\mu \mathcal{L}(\theta^t, \mu^t), \mu \rangle + \frac{1}{2\eta_\mu} \|\mu\|_2^2. \tag{55}$$

By definition, it is clear that $\mu^{t+1} = \operatorname{argmin}_{\mu \in \mathcal{U}} \widetilde{\mathcal{L}}^t(\mu)$. Also, we note that both $\mathcal{L}^t(\mu)$ and $\widetilde{\mathcal{L}}^t(\mu)$ are $1/\eta_\mu$-strongly convex quadratic functions.

*Proof of Theorem 4.4.* Throughout the proof below, we assume that the following error bounds hold for $t = 0, 1, \ldots, T - 1$.

$$\left\|\widetilde{\nabla}_\mu \mathcal{L}(\theta^t, \mu^t) - \nabla_\mu \mathcal{L}(\theta^t, \mu^t)\right\|_2^2 \le \epsilon_\mu, \quad \left\|\widetilde{\nabla}_\theta \mathcal{L}(\theta^t, \mu^{t+1}) - \nabla_\theta \mathcal{L}(\theta^t, \mu^{t+1})\right\|_2^2 \le \epsilon_\theta. \tag{56}$$

According to Remark G.2, this is always the case in the exact setting with $\epsilon_\mu = 0$ and $\epsilon_\theta = n\epsilon_3$, where $\epsilon_3$ is the approximation error of the truncated policy gradient estimator.

We begin with a general argument that applies to both the exact setting (Theorem 4.4) and the sample-based setting (Theorem 4.6). Since the feasible set $\Theta$ is convex, by the property of the projection operator, it holds that

$$\left\langle \left[\theta^t + \eta_\theta \cdot \widetilde{\nabla}_\theta \mathcal{L}\left(\theta^t, \mu^{t+1}\right)\right] - \theta^{t+1}, \theta - \theta^{t+1} \right\rangle \le 0, \ \forall \theta \in \Theta, \tag{57}$$

which thus implies that

$$\left\langle \widetilde{\nabla}_\theta \mathcal{L}\left(\theta^t, \mu^{t+1}\right), \theta - \theta^{t+1} \right\rangle \le \frac{1}{\eta_\theta} \left\langle \theta^{t+1} - \theta^t, \theta - \theta^{t+1} \right\rangle. \tag{58}$$

Therefore, for any $\theta \in \Theta$, we have that

$$
\begin{aligned}
\left\langle \widetilde{\nabla}_\theta \mathcal{L}\left(\theta^t, \mu^{t+1}\right), \theta - \theta^t \right\rangle &= \left\langle \widetilde{\nabla}_\theta \mathcal{L}\left(\theta^t, \mu^{t+1}\right), \theta - \theta^{t+1} \right\rangle + \left\langle \widetilde{\nabla}_\theta \mathcal{L}\left(\theta^t, \mu^{t+1}\right), \theta^{t+1} - \theta^t \right\rangle \\
&\le \frac{1}{\eta_\theta} \left\langle \theta^{t+1} - \theta^t, \theta - \theta^{t+1} \right\rangle + \left\langle \widetilde{\nabla}_\theta \mathcal{L}\left(\theta^t, \mu^{t+1}\right), \theta^{t+1} - \theta^t \right\rangle \\
&= \frac{1}{\eta_\theta} \left\langle \theta^{t+1} - \theta^t, \theta - \theta^t \right\rangle + \frac{1}{\eta_\theta} \left\langle \theta^{t+1} - \theta^t, \theta^t - \theta^{t+1} \right\rangle \\
&\quad + \left\langle \widetilde{\nabla}_\theta \mathcal{L}\left(\theta^t, \mu^{t+1}\right), \theta^{t+1} - \theta^t \right\rangle \\
&= \frac{1}{\eta_\theta} \left\langle \theta^{t+1} - \theta^t, \theta - \theta^t \right\rangle - \frac{1}{\eta_\theta} \left\|\theta^t - \theta^{t+1}\right\|_2^2 + \left\langle \widetilde{\nabla}_\theta \mathcal{L}\left(\theta^t, \mu^{t+1}\right), \theta^{t+1} - \theta^t \right\rangle.
\end{aligned}
\tag{59}
$$

By taking the maximum on both sides over all $\theta \in \Theta$ such that $\left\|\theta - \theta^t\right\|_2 \le 1$, the inequality (59) becomes

$$
\begin{aligned}
&\max_{\theta \in \Theta, \|\theta - \theta^t\| \le 1} \left\langle \widetilde{\nabla}_\theta \mathcal{L}\left(\theta^t, \mu^{t+1}\right), \theta - \theta^t \right\rangle \\
&\le \max_{\theta \in \Theta, \|\theta - \theta^t\| \le 1} \left\{ \frac{1}{\eta_\theta} \left\langle \theta^{t+1} - \theta^t, \theta - \theta^t \right\rangle \right\} - \frac{1}{\eta_\theta} \left\|\theta^t - \theta^{t+1}\right\|_2^2 + \left\langle \widetilde{\nabla}_\theta \mathcal{L}\left(\theta^t, \mu^{t+1}\right), \theta^{t+1} - \theta^t \right\rangle \\
&\le \frac{1}{\eta_\theta} \left\|\theta^{t+1} - \theta^t\right\|_2 - \frac{1}{\eta_\theta} \left\|\theta^t - \theta^{t+1}\right\|_2^2 + \left\|\widetilde{\nabla}_\theta \mathcal{L}\left(\theta^t, \mu^{t+1}\right)\right\|_2 \cdot \left\|\theta^{t+1} - \theta^t\right\|_2 \\
&\le \left( \frac{1}{\eta_\theta} + \left\|\widetilde{\nabla}_\theta \mathcal{L}\left(\theta^t, \mu^{t+1}\right)\right\|_2 \right) \left\|\theta^{t+1} - \theta^t\right\|_2,
\end{aligned}
\tag{60}
$$

where we apply the Cauchy's inequality $\langle x, y \rangle \leq \|x\|_2 \|y\|_2$ in the third line. Thus, it holds that

$$
\begin{aligned}
& \left[ \mathcal{X}\left(\theta^t, \mu^{t+1}\right) \right]^2 \\
&= \left[ \max_{\theta \in \Theta, \|\theta - \theta^t\| \leq 1} \left\langle \nabla_\theta \mathcal{L}\left(\theta^t, \mu^{t+1}\right), \theta - \theta^t \right\rangle \right]^2 \\
&= \left[ \max_{\theta \in \Theta, \|\theta - \theta^t\| \leq 1} \left\{ \left\langle \widetilde{\nabla}_\theta \mathcal{L}\left(\theta^t, \mu^{t+1}\right), \theta - \theta^t \right\rangle + \left\langle \nabla_\theta \mathcal{L}\left(\theta^t, \mu^{t+1}\right) - \widetilde{\nabla}_\theta \mathcal{L}\left(\theta^t, \mu^{t+1}\right), \theta - \theta^t \right\rangle \right\} \right]^2 \\
&\leq \left[ \max_{\theta \in \Theta, \|\theta - \theta^t\| \leq 1} \left\{ \left\langle \widetilde{\nabla}_\theta \mathcal{L}\left(\theta^t, \mu^{t+1}\right), \theta - \theta^t \right\rangle \right\} + \max_{\theta \in \Theta, \|\theta - \theta^t\| \leq 1} \left\{ \left\langle \nabla_\theta \mathcal{L}\left(\theta^t, \mu^{t+1}\right) - \widetilde{\nabla}_\theta \mathcal{L}\left(\theta^t, \mu^{t+1}\right), \theta - \theta^t \right\rangle \right\} \right]^2 \\
&\leq \left[ \max_{\theta \in \Theta, \|\theta - \theta^t\| \leq 1} \left\{ \left\langle \widetilde{\nabla}_\theta \mathcal{L}\left(\theta^t, \mu^{t+1}\right), \theta - \theta^t \right\rangle \right\} + \left\| \nabla_\theta \mathcal{L}\left(\theta^t, \mu^{t+1}\right) - \widetilde{\nabla}_\theta \mathcal{L}\left(\theta^t, \mu^{t+1}\right) \right\|_2 \right]^2 \\
&\leq 2 \left[ \max_{\theta \in \Theta, \|\theta - \theta^t\| \leq 1} \left\{ \left\langle \widetilde{\nabla}_\theta \mathcal{L}\left(\theta^t, \mu^{t+1}\right), \theta - \theta^t \right\rangle \right\} \right]^2 + 2 \left\| \nabla_\theta \mathcal{L}\left(\theta^t, \mu^{t+1}\right) - \widetilde{\nabla}_\theta \mathcal{L}\left(\theta^t, \mu^{t+1}\right) \right\|_2^2 \\
&\overset{(\Delta)}{\leq} 2 \left( \frac{1}{\eta_\theta} + \left\| \widetilde{\nabla}_\theta \mathcal{L}\left(\theta^t, \mu^{t+1}\right) \right\|_2 \right)^2 \left\| \theta^{t+1} - \theta^t \right\|_2^2 + 2\epsilon_\theta \\
&\leq 2 \left( \frac{1}{\eta_\theta} + M_\theta \right)^2 \left\| \theta^{t+1} - \theta^t \right\|_2^2 + 2\epsilon_\theta,
\end{aligned}
$$

$$(61)$$

where we apply (60) and (56) in $(\Delta)$. The last line follows from the fact that $\widetilde{\nabla}_\theta \mathcal{L}\left(\theta^t, \mu^{t+1}\right)$ is the Monte Carlo estimator for the true gradient $\nabla_\theta \mathcal{L}\left(\theta^t, \mu^{t+1}\right)$, thus enjoying the same upper bound $\left\| \widetilde{\nabla}_\theta \mathcal{L}\left(\theta^t, \mu^{t+1}\right) \right\|_2 \leq M_\theta$ (see Lemma F.5). Therefore, it is important to properly upper-bound the term $\left\| \theta^{t+1} - \theta^t \right\|_2^2$. We proceed by focusing on the dual variable update. By the definition of $\mathcal{L}^t(\mu)$ in (55), we can derive that

$$
\begin{aligned}
\mathcal{L}^{t+1}(\mu^{t+2}) - \mathcal{L}^t(\mu^{t+2}) &= \left[ \mathcal{L}(\theta^{t+1}, \mu^{t+2}) + \frac{1}{2\eta_\mu} \|\mu^{t+2}\|_2^2 \right] - \left[ \mathcal{L}(\theta^t, \mu^{t+2}) + \frac{1}{2\eta_\mu} \|\mu^{t+2}\|_2^2 \right] \\
&= \left[ \mathcal{L}(\theta^{t+1}, \mu^{t+2}) - \mathcal{L}(\theta^t, \mu^{t+2}) \right] \\
&\geq \left\langle \nabla_\theta \mathcal{L}(\theta^t, \mu^{t+2}), \theta^{t+1} - \theta^t \right\rangle - \frac{L_{\theta\theta}}{2} \left\| \theta^{t+1} - \theta^t \right\|_2^2,
\end{aligned}
$$

$$(62)$$

where we apply the $L_{\theta\theta}$-smoothness of $\mathcal{L}(\theta, \mu)$ w.r.t. $\theta$ (see Lemma F.5), i.e.,

$$
-\mathcal{L}(\theta^{t+1}, \mu^{t+2}) \leq -\mathcal{L}(\theta^t, \mu^{t+2}) + \left\langle -\nabla_\theta \mathcal{L}(\theta^t, \mu^{t+2}), \theta^{t+1} - \theta^t \right\rangle + \frac{L_{\theta\theta}}{2} \left\| \theta^{t+1} - \theta^t \right\|_2^2.
$$

Then, from (62), we further deduce that

$$
\begin{aligned}
\mathcal{L}^{t+1}(\mu^{t+2}) &\geq \mathcal{L}^t(\mu^{t+2}) + \left\langle \nabla_\theta \mathcal{L}(\theta^t, \mu^{t+2}), \theta^{t+1} - \theta^t \right\rangle - \frac{L_{\theta\theta}}{2} \left\| \theta^{t+1} - \theta^t \right\|_2^2 \\
&= \mathcal{L}^t(\mu^{t+2}) + \left\langle \nabla_\theta \mathcal{L}(\theta^t, \mu^{t+2}) - \nabla_\theta \mathcal{L}(\theta^t, \mu^{t+1}), \theta^{t+1} - \theta^t \right\rangle - \frac{L_{\theta\theta}}{2} \left\| \theta^{t+1} - \theta^t \right\|_2^2 \\
&\quad + \left\langle \nabla_\theta \mathcal{L}(\theta^t, \mu^{t+1}), \theta^{t+1} - \theta^t \right\rangle \\
&\overset{(\Delta)}{\geq} \mathcal{L}^t(\mu^{t+2}) - L_{\theta\mu} \left\| \mu^{t+2} - \mu^{t+1} \right\|_2 \left\| \theta^{t+1} - \theta^t \right\|_2 - \frac{L_{\theta\theta}}{2} \left\| \theta^{t+1} - \theta^t \right\|_2^2 \\
&\quad + \left\langle \nabla_\theta \mathcal{L}(\theta^t, \mu^{t+1}), \theta^{t+1} - \theta^t \right\rangle \\
&= \mathcal{L}^t(\mu^{t+1}) - L_{\theta\mu} \left\| \mu^{t+2} - \mu^{t+1} \right\|_2 \left\| \theta^{t+1} - \theta^t \right\|_2 - \frac{L_{\theta\theta}}{2} \left\| \theta^{t+1} - \theta^t \right\|_2^2 \\
&\quad + \left\langle \nabla_\theta \mathcal{L}(\theta^t, \mu^{t+1}), \theta^{t+1} - \theta^t \right\rangle + \left[ \mathcal{L}^t(\mu^{t+2}) - \mathcal{L}^t(\mu^{t+1}) \right],
\end{aligned}
$$

$$(63)$$

where $(\Delta)$ is due to Lemma F.5 and Cauchy's inequality. Then, we lower-bound the term $\left[ \mathcal{L}^t(\mu^{t+2}) - \mathcal{L}^t(\mu^{t+1}) \right]$ using the $1/\eta_\mu$-strong convexity of $\mathcal{L}^t(\cdot)$ as follows

$$
\begin{aligned}
\mathcal{L}^t(\mu^{t+2}) - \mathcal{L}^t(\mu^{t+1}) \; &\geq \left\langle \nabla_\mu \mathcal{L}^t(\mu^{t+1}), \mu^{t+2} - \mu^{t+1} \right\rangle + \frac{1}{2\eta_\mu} \left\| \mu^{t+2} - \mu^{t+1} \right\|_2^2 \\
&\stackrel{(\Delta_1)}{=} \left\langle \nabla_\mu \mathcal{L}(\theta^t, \mu^t) + \frac{1}{\eta_\mu} \mu^{t+1}, \mu^{t+2} - \mu^{t+1} \right\rangle + \frac{1}{2\eta_\mu} \left\| \mu^{t+2} - \mu^{t+1} \right\|_2^2 \\
&= \frac{1}{\eta_\mu} \left\langle \mu^{t+1} - \left( -\eta_\mu \widetilde{\nabla}_\mu \mathcal{L}(\theta^t, \mu^t) \right) - \eta_\mu \widetilde{\nabla}_\mu \mathcal{L}(\theta^t, \mu^t), \mu^{t+2} - \mu^{t+1} \right\rangle \\
&\quad + \left\langle \nabla_\mu \mathcal{L}(\theta^t, \mu^t), \mu^{t+2} - \mu^{t+1} \right\rangle + \frac{1}{2\eta_\mu} \left\| \mu^{t+2} - \mu^{t+1} \right\|_2^2 \\
&\stackrel{(\Delta_2)}{\geq} \left\langle \nabla_\mu \mathcal{L}(\theta^t, \mu^t) - \widetilde{\nabla}_\mu \mathcal{L}(\theta^t, \mu^t), \mu^{t+2} - \mu^{t+1} \right\rangle + \frac{1}{2\eta_\mu} \left\| \mu^{t+2} - \mu^{t+1} \right\|_2^2 \\
&\stackrel{(\Delta_3)}{\geq} -\eta_\mu \left\| \nabla_\mu \mathcal{L}(\theta^t, \mu^t) - \widetilde{\nabla}_\mu \mathcal{L}(\theta^t, \mu^t) \right\|_2^2 - \frac{1}{4\eta_\mu} \left\| \mu^{t+2} - \mu^{t+1} \right\|_2^2 \\
&\quad + \frac{1}{2\eta_\mu} \left\| \mu^{t+2} - \mu^{t+1} \right\|_2^2 \\
&\geq -\eta_\mu \epsilon_\mu + \frac{1}{4\eta_\mu} \left\| \mu^{t+2} - \mu^{t+1} \right\|_2^2 .
\end{aligned}
\tag{64}
$$

In the above inequality, $(\Delta_1)$ follows from the definition of $\mathcal{L}^t(\cdot)$ in (55). Next, we use the property of the projection operator in inequality $(\Delta_2)$, i.e.,

$$
\begin{aligned}
&\left\langle \left( -\eta_\mu \widetilde{\nabla}_\mu \mathcal{L}(\theta^t, \mu^t) \right) - \mu^{t+1}, \mu - \mu^{t+1} \right\rangle \\
&= \left\langle \left( -\eta_\mu \widetilde{\nabla}_\mu \mathcal{L}(\theta^t, \mu^t) \right) - \mathcal{P}_\mathcal{U} \left( -\eta_\mu \widetilde{\nabla}_\mu \mathcal{L} \left( \theta^t, \mu^t \right) \right), \mu - \mathcal{P}_\mathcal{U} \left( -\eta_\mu \widetilde{\nabla}_\mu \mathcal{L} \left( \theta^t, \mu^t \right) \right) \right\rangle \leq 0, \; \forall \mu \in \mathcal{U}.
\end{aligned}
$$

Finally, $(\Delta_3)$ is due to Cauchy's inequality $\langle x, y \rangle \geq -k/2 \cdot \|x\|_2^2 - 1/(2k) \cdot \|y\|_2^2$ for any $k > 0$ and the last inequality follows from the error bound in (56).

In addition, the term $\left\langle \nabla_\theta \mathcal{L}(\theta^t, \mu^{t+1}), \theta^{t+1} - \theta^t \right\rangle$ on the right-hand side of (63) can be lower-bounded as follows

$$
\begin{aligned}
&\left\langle \nabla_\theta \mathcal{L}(\theta^t, \mu^{t+1}), \theta^{t+1} - \theta^t \right\rangle \\
&= \left\langle \widetilde{\nabla}_\theta \mathcal{L}(\theta^t, \mu^{t+1}), \theta^{t+1} - \theta^t \right\rangle + \left\langle \nabla_\theta \mathcal{L}(\theta^t, \mu^{t+1}) - \widetilde{\nabla}_\theta \mathcal{L}(\theta^t, \mu^{t+1}), \theta^{t+1} - \theta^t \right\rangle \\
&\geq \frac{1}{\eta_\theta} \left\| \theta^{t+1} - \theta^t \right\|_2^2 + \left\langle \nabla_\theta \mathcal{L}(\theta^t, \mu^{t+1}) - \widetilde{\nabla}_\theta \mathcal{L}(\theta^t, \mu^{t+1}), \theta^{t+1} - \theta^t \right\rangle \\
&\geq \frac{1}{\eta_\theta} \left\| \theta^{t+1} - \theta^t \right\|_2^2 - \frac{\eta_\theta}{2} \left\| \nabla_\theta \mathcal{L}(\theta^t, \mu^{t+1}) - \widetilde{\nabla}_\theta \mathcal{L}(\theta^t, \mu^{t+1}) \right\|_2^2 - \frac{1}{2\eta_\theta} \left\| \theta^{t+1} - \theta^t \right\|_2^2 \\
&= \frac{1}{2\eta_\theta} \left\| \theta^{t+1} - \theta^t \right\|_2^2 - \frac{\eta_\theta}{2} \left\| \nabla_\theta \mathcal{L}(\theta^t, \mu^{t+1}) - \widetilde{\nabla}_\theta \mathcal{L}(\theta^t, \mu^{t+1}) \right\|_2^2 \\
&\geq \frac{1}{2\eta_\theta} \left\| \theta^{t+1} - \theta^t \right\|_2^2 - \frac{\eta_\theta}{2} \epsilon_\theta,
\end{aligned}
\tag{65}
$$

where the first inequality uses (58) by taking $\theta = \theta^t$, and the second inequality is again due to Cauchy's inequality.

Substituting (64) and (65) into the right-hand side of (63), we have that

$$
\mathcal{L}^{t+1}(\mu^{t+2}) - \mathcal{L}^t(\mu^{t+1})
$$

$$
\geq - L_{\theta\mu} \left\| \mu^{t+2} - \mu^{t+1} \right\|_2 \left\| \theta^{t+1} - \theta^t \right\|_2 - \frac{L_{\theta\theta}}{2} \left\| \theta^{t+1} - \theta^t \right\|_2^2 + \frac{1}{2\eta_\theta} \left\| \theta^{t+1} - \theta^t \right\|_2^2 + \frac{1}{4\eta_\mu} \left\| \mu^{t+2} - \mu^{t+1} \right\|_2^2
$$

$$
\quad - \frac{\eta_\theta}{2}\epsilon_\theta - \eta_\mu\epsilon_\mu
$$

$$
= - L_{\theta\mu} \left\| \mu^{t+2} - \mu^{t+1} \right\|_2 \left\| \theta^{t+1} - \theta^t \right\|_2 + \left( \frac{1}{2\eta_\theta} - \frac{L_{\theta\theta}}{2} \right) \left\| \theta^{t+1} - \theta^t \right\|_2^2 + \frac{1}{4\eta_\mu} \left\| \mu^{t+2} - \mu^{t+1} \right\|_2^2
$$

$$
\quad - \frac{\eta_\theta}{2}\epsilon_\theta - \eta_\mu\epsilon_\mu
$$

$$
\overset{(\Delta)}{\geq} - L_{\theta\mu} \left( \frac{1}{4L_{\theta\mu}\eta_\mu} \left\| \mu^{t+2} - \mu^{t+1} \right\|_2^2 + L_{\theta\mu}\eta_\mu \left\| \theta^{t+1} - \theta^t \right\|_2^2 \right) + \left( \frac{1}{2\eta_\theta} - \frac{L_{\theta\theta}}{2} \right) \left\| \theta^{t+1} - \theta^t \right\|_2^2
$$

$$
\quad + \frac{1}{4\eta_\mu} \left\| \mu^{t+2} - \mu^{t+1} \right\|_2^2 - \frac{\eta_\theta}{2}\epsilon_\theta - \eta_\mu\epsilon_\mu
$$

$$
= \left( \frac{1}{2\eta_\theta} - \frac{L_{\theta\theta}}{2} - L_{\theta\mu}^2\eta_\mu \right) \left\| \theta^{t+1} - \theta^t \right\|_2^2 - \frac{\eta_\theta}{2}\epsilon_\theta - \eta_\mu\epsilon_\mu
$$

$$
= L_{\theta\mu}^2\eta_\mu \left\| \theta^{t+1} - \theta^t \right\|_2^2 - \frac{\eta_\theta}{2}\epsilon_\theta - \eta_\mu\epsilon_\mu,
$$

(66)

where we apply the Cauchy's inequality to the term $\left\| \mu^{t+2} - \mu^{t+1} \right\|_2 \left\| \theta^{t+1} - \theta^t \right\|_2$ in $(\Delta)$ and substitute in the value $\eta_\theta = 1/\left( L_{\theta\theta} + 4L_{\theta\mu}^2\eta_\mu \right)$ in the last line. Therefore, by rearranging the terms in (66), we obtain the desired upper bound on $\left\| \theta^{t+1} - \theta^t \right\|_2^2$, i.e.,

$$
\left\| \theta^{t+1} - \theta^t \right\|_2^2 \leq \frac{1}{L_{\theta\mu}^2\eta_\mu} \cdot \left[ \mathcal{L}^{t+1}(\mu^{t+2}) - \mathcal{L}^t(\mu^{t+1}) + \frac{\eta_\theta}{2}\epsilon_\theta + \eta_\mu\epsilon_\mu \right]. \tag{67}
$$

We remark that (67) also implies the terms on the right-hand side must be strictly nonnegative. Returning back to (61) with the above inequality, we deduce that

$$
\left[ \mathcal{X}\left( \theta^t, \mu^{t+1} \right) \right]^2
$$

$$
\leq 2 \left( \frac{1}{\eta_\theta} + M_\theta \right)^2 \frac{1}{L_{\theta\mu}^2\eta_\mu} \cdot \left[ \mathcal{L}^{t+1}(\mu^{t+2}) - \mathcal{L}^t(\mu^{t+1}) + \frac{\eta_\theta}{2}\epsilon_\theta + \eta_\mu\epsilon_\mu \right] + 2\epsilon_\theta
$$

$$
= 2 \left( L_{\theta\theta} + 4L_{\theta\mu}^2\eta_\mu + M_\theta \right)^2 \frac{1}{L_{\theta\mu}^2\eta_\mu} \cdot \left[ \mathcal{L}^{t+1}(\mu^{t+2}) - \mathcal{L}^t(\mu^{t+1}) + \frac{\eta_\theta}{2}\epsilon_\theta + \eta_\mu\epsilon_\mu \right] + 2\epsilon_\theta
$$

$$
= 2 \left[ \frac{(L_{\theta\theta} + M_\theta)^2}{L_{\theta\mu}^2\eta_\mu} + 8(L_{\theta\theta} + M_\theta) + 16L_{\theta\mu}^2\eta_\mu \right] \left[ \mathcal{L}^{t+1}(\mu^{t+2}) - \mathcal{L}^t(\mu^{t+1}) + \frac{\eta_\theta}{2}\epsilon_\theta + \eta_\mu\epsilon_\mu \right] + 2\epsilon_\theta,
$$

(68)

where the first equality follows from substituting in the value of $\eta_\theta$ We sum the inequality (68) over $t = 0, 1, \ldots, T-1$ and divide it by $T$, which yields that

$$
\frac{1}{T} \sum_{t=0}^{T-1} \left[ \mathcal{X}\left( \theta^t, \mu^{t+1} \right) \right]^2
$$

$$
\leq 2 \left[ \frac{(L_{\theta\theta} + M_\theta)^2}{L_{\theta\mu}^2\eta_\mu} + 8(L_{\theta\theta} + M_\theta) + 16L_{\theta\mu}^2\eta_\mu \right] \cdot \left[ \frac{\sum_{t=0}^{T-1} \left[ \mathcal{L}^{t+1}(\mu^{t+2}) - \mathcal{L}^t(\mu^{t+1}) \right]}{T} + \frac{\eta_\theta}{2}\epsilon_\theta + \eta_\mu\epsilon_\mu \right]
$$

$$
\quad + 2\epsilon_\theta
$$

$$
= 2 \left[ \frac{(L_{\theta\theta} + M_\theta)^2}{L_{\theta\mu}^2\eta_\mu} + 8(L_{\theta\theta} + M_\theta) + 16L_{\theta\mu}^2\eta_\mu \right] \cdot \left[ \frac{\left[ \mathcal{L}^T(\mu^{T+1}) - \mathcal{L}^0(\mu^1) \right]}{T} + \frac{\eta_\theta}{2}\epsilon_\theta + \eta_\mu\epsilon_\mu \right] + 2\epsilon_\theta
$$

$$
\leq 2 \left[ \frac{(L_{\theta\theta} + M_\theta)^2}{L_{\theta\mu}^2\eta_\mu} + 8(L_{\theta\theta} + M_\theta) + 16L_{\theta\mu}^2\eta_\mu \right] \cdot \left[ \frac{2M_L}{T} + \frac{n\overline{\mu}^2}{2\eta_\mu T} + \frac{\epsilon_\theta}{2L_{\theta\theta} + 8L_{\theta\mu}^2\eta_\mu} + \eta_\mu\epsilon_\mu \right] + 2\epsilon_\theta,
$$

(69)

where the equality is due to a telescoping sum. The last line follows from the choice of $\eta_\theta$ and the boundedness of $\mathcal{L}^t(\cdot)$. Specifically, by Assumption 4.3 and Lemma F.5, we have that

$$\left|\mathcal{L}^T(\mu^{T+1}) - \mathcal{L}^0(\mu^1)\right| = \left|\mathcal{L}(\theta^T, \mu^{T+1}) + \frac{1}{2\eta_\mu}\left\|\mu^{T+1}\right\|_2^2 - \mathcal{L}(\theta^0, \mu^1) - \frac{1}{2\eta_\mu}\left\|\mu^1\right\|_2^2\right|$$

$$\leq \left|\mathcal{L}(\theta^T, \mu^{T+1})\right| + \left|\mathcal{L}(\theta^0, \mu^1)\right| + \frac{1}{2\eta_\mu}\max_{\mu \in \mathcal{U}}\|\mu\|_2^2$$

$$\leq 2M_L + \frac{1}{2\eta_\mu}n\overline{\mu}^2.$$

Now, we focus on evaluating the dual stationarity metric $\mathcal{Y}(\theta, \mu)$ defined in (23). Firstly, we recall that the dual gradient is equal to the values of constraint functions and is irrelevant to the value of the dual variable, i.e., $\nabla_\mu\mathcal{L}(\theta, \mu) = \nabla_\mu\mathcal{L}(\theta, \mu')$, $\forall \mu, \mu'$. Then, for any $t = 0, 1, \ldots, T-1$, we have that

$$\mathcal{Y}(\theta^t, \mu^{t+1})$$

$$= -\min_{\mu \in \mathcal{U}, \|\mu - \mu^{t+1}\|_2 \leq 1}\left\langle\nabla_\mu\mathcal{L}(\theta^t, \mu^t), \mu - \mu^{t+1}\right\rangle$$

$$= -\min_{\mu \in \mathcal{U}, \|\mu - \mu^{t+1}\|_2 \leq 1}\left\{\left\langle\widetilde{\nabla}_\mu\mathcal{L}(\theta^t, \mu^t), \mu - \mu^{t+1}\right\rangle + \left\langle\nabla_\mu\mathcal{L}(\theta^t, \mu^t) - \widetilde{\nabla}_\mu\mathcal{L}(\theta^t, \mu^t), \mu - \mu^{t+1}\right\rangle\right\}$$

$$= -\min_{\mu \in \mathcal{U}, \|\mu - \mu^{t+1}\|_2 \leq 1}\left\{\left\langle\nabla_\mu\widetilde{\mathcal{L}}^t(\mu^{t+1}) - \frac{1}{\eta_\mu}\mu^{t+1}, \mu - \mu^{t+1}\right\rangle + \left\langle\nabla_\mu\mathcal{L}(\theta^t, \mu^t) - \widetilde{\nabla}_\mu\mathcal{L}(\theta^t, \mu^t), \mu - \mu^{t+1}\right\rangle\right\},$$

$$(70)$$

where we use the definition of $\widetilde{\mathcal{L}}^t(\cdot)$ in the last equality. Since $\mu^{t+1}$ is the minimizer of the convex quadratic function $\widetilde{\mathcal{L}}^t(\cdot)$ in $\mathcal{U}$, it follows that

$$\left\langle\nabla_\mu\widetilde{\mathcal{L}}^t(\mu^{t+1}), \mu - \mu^{t+1}\right\rangle \geq 0, \ \forall \mu \in \mathcal{U}.$$

Substituting the above inequality into (70), we conclude that

$$\mathcal{Y}(\theta^t, \mu^{t+1})$$

$$\leq -\min_{\mu \in \mathcal{U}, \|\mu - \mu^{t+1}\|_2 \leq 1}\left\{-\frac{1}{\eta_\mu}\left\langle\mu^{t+1}, \mu - \mu^{t+1}\right\rangle + \left\langle\nabla_\mu\mathcal{L}(\theta^t, \mu^t) - \widetilde{\nabla}_\mu\mathcal{L}(\theta^t, \mu^t), \mu - \mu^{t+1}\right\rangle\right\}$$

$$= \max_{\mu \in \mathcal{U}, \|\mu - \mu^{t+1}\|_2 \leq 1}\left\{\frac{1}{\eta_\mu}\left\langle\mu^{t+1}, \mu - \mu^{t+1}\right\rangle + \left\langle\nabla_\mu\mathcal{L}(\theta^t, \mu^t) - \widetilde{\nabla}_\mu\mathcal{L}(\theta^t, \mu^t), \mu^{t+1} - \mu\right\rangle\right\}$$

$$\leq \max_{\mu \in \mathcal{U}, \|\mu - \mu^{t+1}\|_2 \leq 1}\left\{\frac{1}{\eta_\mu}\left\|\mu^{t+1}\right\|_2\left\|\mu^{t+1} - \mu\right\|_2 + \left\|\nabla_\mu\mathcal{L}(\theta^t, \mu^t) - \widetilde{\nabla}_\mu\mathcal{L}(\theta^t, \mu^t)\right\|_2\left\|\mu^{t+1} - \mu\right\|_2\right\}$$

$$(71)$$

$$\leq \frac{1}{\eta_\mu}\left\|\mu^{t+1}\right\|_2 + \left\|\nabla_\mu\mathcal{L}(\theta^t, \mu^t) - \widetilde{\nabla}_\mu\mathcal{L}(\theta^t, \mu^t)\right\|_2$$

$$\leq \frac{1}{\eta_\mu}\sqrt{n}\overline{\mu} + \sqrt{\epsilon_\mu}.$$

In the exact setting, according to Remark G.2, inequality(69) can be simplified by taking $\epsilon_\mu = 0$ and $\epsilon_\theta = n\epsilon_3$:

$$\frac{1}{T}\sum_{t=0}^{T-1}\left[\mathcal{X}\left(\theta^t, \mu^{t+1}\right)\right]^2$$

$$\leq 2\left[\frac{(L_{\theta\theta} + M_\theta)^2}{L_{\theta\mu}^2\eta_\mu} + 8(L_{\theta\theta} + M_\theta) + 16L_{\theta\mu}^2\eta_\mu\right]\cdot\left[\frac{2M_L}{T} + \frac{n\overline{\mu}^2}{2\eta_\mu T} + \frac{n\epsilon_3}{2L_{\theta\theta} + 8L_{\theta\mu}^2\eta_\mu}\right] + 2n\epsilon_3$$

$$= 2\left[\mathcal{O}\left(T^{-1/3}\right) + \mathcal{O}(1) + \mathcal{O}\left(T^{1/3}\right)\right]\cdot\left[\mathcal{O}\left(T^{-1}\right) + \mathcal{O}\left(T^{-4/3}\right) + \frac{n\epsilon_3}{\mathcal{O}\left(1 + T^{1/3}\right)}\right] + 2n\epsilon_3$$

$$\leq \mathcal{O}\left(T^{1/3}\right)\cdot\left[\mathcal{O}\left(T^{-1}\right) + n\epsilon_3\cdot\mathcal{O}\left(T^{-1/3}\right)\right] + 2n\epsilon_3$$

$$= \mathcal{O}\left(T^{-2/3}\right) + \mathcal{O}(\epsilon_3)$$

$$= \mathcal{O}\left(T^{-2/3}\right) + \mathcal{O}(\phi_0^{2\kappa}),$$

where the last equality follows from the definition of $\epsilon_3$ in (53). Since $1/T \cdot \sum_{t=0}^{T-1} \left[ \mathcal{X} \left( \theta^t, \mu^{t+1} \right) \right]^2$ is the average of $T$ non-negative numbers, there must exist $t^* \in \{0, 1, \ldots, T-1\}$ such that

$$\left[ \mathcal{X} \left( \theta^{t^*}, \mu^{t^*+1} \right) \right]_2^2 = \mathcal{O} \left( T^{-2/3} \right) + \mathcal{O}(\phi_0^{2\kappa}).$$

Therefore, it follows from inequality (71) that

$$\mathcal{E} \left( \theta^{t^*}, \mu^{t^*+1} \right) = \left[ \mathcal{X} \left( \theta^{t^*}, \mu^{t^*+1} \right) \right]_2^2 + \left[ \mathcal{Y} \left( \theta^{t^*}, \mu^{t^*+1} \right) \right]^2$$

$$\leq \mathcal{O} \left( T^{-2/3} \right) + \mathcal{O}(\phi_0^{2\kappa}) + \left( \frac{1}{\eta_\mu} \sqrt{n\bar{\mu}} \right)^2$$

$$= \mathcal{O} \left( T^{-2/3} \right) + \mathcal{O}(\phi_0^{2\kappa}),$$

which completes the proof. $\qquad\square$

*Proof of Theorem 4.6.* As stated in Proposition G.1, for any fixed $t \geq 0$, the empirical gradient estimators have the error bounds (52d) and (52e) with probability $1 - 2n\delta_0$. By applying the union bound, we have that the error bounds are met for all $t = 0, 1, \ldots, T$ with probability $1 - (T+1) \cdot (2n\delta_0) = 1 - \delta$. Assuming that the error bounds hold true, the previously derived inequalities (69) and (71) are applicable. In particular, for the primal stationarity metric $\mathcal{X}(\theta, \mu)$, we have that

$$\frac{1}{T} \sum_{t=0}^{T-1} \left[ \mathcal{X} \left( \theta^t, \mu^{t+1} \right) \right]^2$$

$$\leq 2 \left[ \frac{(L_{\theta\theta} + M_\theta)^2}{L_{\theta\mu}^2 \eta_\mu} + 8(L_{\theta\theta} + M_\theta) + 16 L_{\theta\mu}^2 \eta_\mu \right] \cdot \left[ \frac{2M_L}{T} + \frac{n\bar{\mu}^2}{2\eta_\mu T} + \frac{\epsilon_\theta}{2L_{\theta\theta} + 8L_{\theta\mu}^2 \eta_\mu} + \eta_\mu \epsilon_\mu \right] + 2\epsilon_\theta. \tag{72}$$

With the choice of the batch size $B = \mathcal{O} \left( \log(1/\delta_0) \epsilon^{-2} \right)$ and episode length $H = \log(1/\epsilon)$ as stated in Theorem 4.6, it holds that

$$\epsilon_\mu = \frac{(M_r \epsilon_1(\delta_0))^2}{n} = \frac{M_r^2}{n} \cdot \frac{4 + 2\gamma^{2H} B - 16\log\delta_0}{(1-\gamma)^2 B} = \mathcal{O} \left( \frac{1}{B} + \gamma^{2H} + \frac{\log(1/\delta_0)}{B} \right) = \mathcal{O}(\epsilon^2). \tag{73}$$

Similarly, since $\epsilon_0 = \sqrt{\epsilon}$, the size of the policy gradient approximation error can be evaluated as

$$\epsilon_\theta = \left( \sum_{i \in \mathcal{N}} \frac{|\mathcal{N}_i^\kappa|^2}{n^2} \right) \epsilon_2(\delta_0) + n\epsilon_3$$

$$= \mathcal{O} \left( \epsilon_0^2 + \frac{\log(1/\delta_0)}{B} + \gamma^{2H} + \phi_0^{2\kappa} \right) = \mathcal{O} \left( \epsilon + \epsilon^2 + \phi_0^{2\kappa} \right) = \mathcal{O} \left( \epsilon + \phi_0^{2\kappa} \right). \tag{74}$$

Therefore, since the step-sizes are chosen as $\eta_\mu = \mathcal{O} \left( \epsilon^{-0.5} \right)$ and $\eta_\theta = 1/\left( L_{\theta\theta} + 4L_{\theta\mu}^2 \eta_\mu \right)$, and the number of periods is $T = \mathcal{O} \left( \epsilon^{-1.5} \right)$, we deduce from (72) that

$$\frac{1}{T} \sum_{t=0}^{T-1} \left[ \mathcal{X} \left( \theta^t, \mu^{t+1} \right) \right]^2$$

$$= \left[ \mathcal{O} \left( \sqrt{\epsilon} \right) + \mathcal{O} \left( 1 \right) + \mathcal{O} \left( \epsilon^{-0.5} \right) \right] \cdot \left[ \mathcal{O} \left( \epsilon^{1.5} \right) + \mathcal{O} \left( \epsilon^2 \right) + \frac{\mathcal{O} \left( \epsilon + \phi_0^{2\kappa} \right)}{\mathcal{O} \left( \epsilon^{-0.5} \right)} + \mathcal{O} \left( \epsilon^{1.5} \right) \right] + \mathcal{O} \left( \epsilon + \phi_0^{2\kappa} \right)$$

$$= \mathcal{O} \left( \epsilon^{-0.5} \right) \cdot \left[ \mathcal{O} \left( \epsilon^{1.5} \right) + \frac{\mathcal{O} \left( \epsilon + \phi_0^{2\kappa} \right)}{\mathcal{O} \left( \epsilon^{-0.5} \right)} \right] + \mathcal{O} \left( \epsilon + \phi_0^{2\kappa} \right)$$

$$= \mathcal{O} \left( \epsilon + \phi_0^{2\kappa} \right). \tag{75}$$

As a result, there must exist $t^* \in \{0, 1, \ldots, T-1\}$ that satisfies

$$\left[ \mathcal{X} \left( \theta^{t^*}, \mu^{t^*+1} \right) \right]_2^2 = \mathcal{O} \left( \epsilon \right) + \mathcal{O}(\phi_0^{2\kappa}).$$

At the meanwhile, it follows from (71) that

$$\mathcal{Y}\left(\theta^{t^\star}, \mu^{t^\star+1}\right) \le \frac{1}{\eta_\mu}\sqrt{n\bar{\mu}} + \sqrt{\epsilon_\mu} = \mathcal{O}\left(\sqrt{\epsilon}+\epsilon\right) = \mathcal{O}\left(\sqrt{\epsilon}\right).$$

Thus, we conclude that

$$\mathcal{E}\left(\theta^{t^\star}, \mu^{t^\star+1}\right) = \left[\mathcal{X}\left(\theta^{t^\star}, \mu^{t^\star+1}\right)\right]_2^2 + \left[\mathcal{Y}\left(\theta^{t^\star}, \mu^{t^\star+1}\right)\right]^2 = \mathcal{O}\left(\epsilon\right) + \mathcal{O}(\phi_0^{2\kappa}) + \mathcal{O}(\epsilon) = \mathcal{O}\left(\epsilon\right) + \mathcal{O}(\phi_0^{2\kappa}). \tag{76}$$

In each period, the number of samples required is

$$\begin{aligned}
B \times H + \mathcal{O}\left(1/(\epsilon_0)^2\right) &= \mathcal{O}\left(\log(1/\delta_0)\epsilon^{-2}\right) \cdot \log(1/\epsilon) + \mathcal{O}\left(1/\epsilon\right) \\
&= \mathcal{O}\left(\log(T/\delta)\epsilon^{-2}\right) \cdot \log(1/\epsilon) + \mathcal{O}\left(1/\epsilon\right) \\
&= \mathcal{O}\left(\log(\epsilon^{-1.5}/\delta)\epsilon^{-2}\right) \cdot \log(1/\epsilon) + \mathcal{O}\left(1/\epsilon\right) \\
&= \widetilde{\mathcal{O}}\left(\epsilon^{-2}\right),
\end{aligned} \tag{77}$$

where the first part comes from the trajectory sampling and the second part comes from the truncated shadow Q-function evaluation. Therefore, the total number of samples required is $T \cdot \widetilde{\mathcal{O}}\left(\epsilon^{-2}\right) = \widetilde{\mathcal{O}}\left(\epsilon^{-3.5}\right)$. This completes the proof. □

## G.1 Proof of Proposition G.1

*Proof of* (52a) *and* (52b). The proof can be found in [23, Appendix D.1]. For the sake of completeness, we will also provide it here.

Let $\mathcal{F}^{t-1}$ denote the $\sigma$-algebra generated by all trajectories sampled at $0, 1, \ldots, t-1$-th periods. For any trajectory $\tau = \{(s^0, a^0), \cdots, (s^{H-1}, a^{H-1})\}$ of length $H$, we use the shorthand notation $\lambda_i(\tau) := \sum_{k=0}^{H-1} \gamma^k \cdot \mathbb{1}_i\left(s_i^k, a_i^k\right)$ to denote the empirical occupancy measure estimation along trajectory $\tau$. Then, by the definition of $\widetilde{\lambda}_i^t$ in (17), we have that $\widetilde{\lambda}_i^t = 1/B \cdot \sum_{\tau \in \mathcal{B}_i^t} \lambda_i(\tau)$ and thus

$$\left\| \mathbb{E}\left[\widetilde{\lambda}_i^t \big| \mathcal{F}^{t-1}\right] - \lambda_i^{\pi_{\theta^t}} \right\|_1 = \left\| \mathbb{E}\left[\frac{1}{B}\sum_{\tau \in \mathcal{B}_i^t}\lambda_i(\tau)\bigg|\mathcal{F}^{t-1}\right] - \lambda_i^{\pi_{\theta^t}} \right\|_1 \le \frac{\gamma^H}{1-\gamma}. \tag{78}$$

Additionally, since it always holds that $\|\lambda_i(\tau)\|_2^2 \le \|\lambda_i(\tau)\|_1^2 \le 1/(1-\gamma)^2$, by [88, Lemma 18], we have that for an agent $i \in \mathcal{N}$,

$$\mathbb{P}\left(\left\|\widetilde{\lambda}_i^t - \mathbb{E}\left[\widetilde{\lambda}_i^t \big| \mathcal{F}^{t-1}\right]\right\|_2^2 \ge \epsilon\right) \le \exp\left(-\frac{2+(1-\gamma)^2\epsilon B}{8}\right).$$

By setting $\epsilon = \frac{2-8\log\delta_0}{(1-\gamma)^2 B}$, the above equation becomes

$$\mathbb{P}\left(\left\|\widetilde{\lambda}_i^t - \mathbb{E}\left[\widetilde{\lambda}_i^t\big|\mathcal{F}^{t-1}\right]\right\|_2^2 \ge \frac{2-8\log\delta_0}{(1-\gamma)^2 B}\right) \le \delta_0.$$

Together with (78), we derive that with probability at least $1 - \delta_0$, it holds that

$$\begin{aligned}
\left\|\widetilde{\lambda}_i^t - \lambda_i^{\pi_{\theta^t}}\right\|_2^2 &\le 2\left\|\widetilde{\lambda}_i^t - \mathbb{E}\left[\widetilde{\lambda}_i^t\big|\mathcal{F}^{t-1}\right]\right\|_2^2 + 2\left\|\mathbb{E}\left[\widetilde{\lambda}_i^t\big|\mathcal{F}^{t-1}\right] - \lambda_i^{\pi_{\theta^t}}\right\|_2^2 \\
&\le \frac{2\gamma^{2H}}{(1-\gamma)^2} + \frac{4-16\log\delta_0}{(1-\gamma)^2 B} \\
&= \frac{4+2\gamma^{2H}B - 16\log\delta_0}{(1-\gamma)^2 B} =: \left(\epsilon_1(\delta_0)\right)^2,
\end{aligned} \tag{79}$$

where the first inequality follows from the fact that $\|x+y\|_2^2 \le 2\|x\|_2^2 + 2\|y\|_2^2$ for any two vectors $x, y$. Thus, by applying the union bound, we know that with probability $1 - n\delta_0$, (79) holds for every agent $i \in \mathcal{N}$.

When (79) holds for all agents, by the Lipschitz continuity of $\nabla_{\lambda_i} f_i(\cdot)$ and $\nabla_{\lambda_i} g_i(\cdot)$ (see Assumption 4.1), we have that

$$\left\|\widetilde{r}_{f_i}^t - r_{f_i}^{\pi_{\theta^t}}\right\|_\infty = \left\|\nabla_{\lambda_i} f_i(\widetilde{\lambda}_i^t) - \nabla_{\lambda_i} f_i(\lambda_i^{\pi_{\theta^t}})\right\|_\infty \le L_\lambda \left\|\widetilde{\lambda}_i^t - \lambda_i^{\pi_{\theta^t}}\right\|_2 \le L_\lambda \epsilon_1(\delta_0). \tag{80}$$

This also holds for the constraint shadow rewards $r_{g_i}$, which completes the proof of (52a) and (52b). □

*Proof of* (52c). We note that Line 6 of Algorithm 1 aims at estimating the truncated Q-function $\widehat{Q}^{\pi_{\theta^t}}_{\diamond_i}$ with the empirical shadow reward $\widetilde{r}^t_{\diamond_i}$. Thus, the approximation error in this step can be attributed to two factors: the imprecision of the empirical shadow reward and the evaluation subroutine used. Recall that we denote $\widehat{Q}^{\pi_\theta}_i(r_i; \cdot, \cdot)$ as the true truncated local Q-function with reward $r_i(\cdot, \cdot)$ under policy $\pi_\theta$. Then, we can decompose the approximation error as follows

$$
\begin{aligned}
\left\| \widetilde{Q}^t_{\diamond_i} - \widehat{Q}^{\pi_{\theta^t}}_{\diamond_i} \right\|_\infty &\leq \left\| \widetilde{Q}^t_{\diamond_i} - \widehat{Q}^{\pi_{\theta^t}}_i(\widetilde{r}^t_{\diamond_i}; \cdot, \cdot) \right\|_\infty + \left\| \widehat{Q}^{\pi_{\theta^t}}_i(\widetilde{r}^t_{\diamond_i}; \cdot, \cdot) - \widehat{Q}^{\pi_{\theta^t}}_{\diamond_i} \right\|_\infty \\
&\leq \left\| \widetilde{r}^t_{\diamond_i} \right\|_\infty \epsilon_0 + \left\| \widehat{Q}^{\pi_{\theta^t}}_i(\widetilde{r}^t_{\diamond_i} - r^{\pi_{\theta^t}}_{\diamond_i}; \cdot, \cdot) \right\|_\infty \\
&\leq M_r \epsilon_0 + \frac{\left\| \widetilde{r}^t_{\diamond_i} - r^{\pi_{\theta^t}}_{\diamond_i} \right\|_\infty}{1 - \gamma},
\end{aligned}
\tag{81}
$$

where the second inequality follows from Assumption 4.5 and the third inequality is due to $\left\| \widehat{Q}^{\pi_\theta}_i(r_i; \cdot, \cdot) \right\| \leq \|r_i\|_\infty / (1 - \gamma)$ for any reward function $r_i(\cdot, \cdot)$. Therefore, when (52b) holds, which happens with probability $1 - n\delta_0$, it also holds that

$$
\left\| \widetilde{Q}^t_{\diamond_i} - \widehat{Q}^{\pi_{\theta^t}}_{\diamond_i} \right\|_\infty \leq M_r \epsilon_0 + \frac{\left\| \widetilde{r}^t_{\diamond_i} - r^{\pi_{\theta^t}}_{\diamond_i} \right\|_\infty}{1 - \gamma} \leq M_r \epsilon_0 + \frac{L_\lambda \epsilon_1(\delta_0)}{1 - \gamma}, \ \forall \diamond \in \{f, g\}, i \in \mathcal{N},
$$

which completes the proof of (52c). $\qquad \square$

*Proof of* (52d). The dual gradient is equal to the constraint function value, i.e., $\nabla_{\mu_i} \mathcal{L}(\theta^t, \mu^t) = G_i(\theta^t)/n = g_i(\lambda^{\pi_\theta}_i)/n$, and the empirical estimator we use has the expression $\widetilde{\nabla}_{\mu_i} \mathcal{L}(\theta^t, \mu^t) = \widetilde{g}^t_i/n = g_i(\widetilde{\lambda}^t_i)/n$. By Lemma F.5 *(I)*, the shadow rewards are bounded by the constant $M_r$, which is equivalent to

$$
\max_{i \in \mathcal{N}, \theta \in \Theta} \left\| \nabla_{\lambda_i} g(\lambda^{\pi_\theta}_i) \right\|_2 = \max_{i \in \mathcal{N}, \theta \in \Theta} \left\| r^{\pi_\theta}_{g_i} \right\|_2 \leq M_r.
\tag{82}
$$

Thus, for every $i \in \mathcal{N}$, the constraint utility $g_i(\cdot)$ is $M_r$-Lipschitz continuous w.r.t. its local occupancy measure. Therefore, it holds that

$$
\begin{aligned}
\left\| \widetilde{\nabla}_\mu \mathcal{L}(\theta^t, \mu^t) - \nabla_\mu \mathcal{L}(\theta^t, \mu^t) \right\|^2_2 &= \sum_{i \in \mathcal{N}} \left| \widetilde{\nabla}_{\mu_i} \mathcal{L}(\theta^t, \mu^t) - \nabla_{\mu_i} \mathcal{L}(\theta^t, \mu^t) \right|^2_2 \\
&= \frac{1}{n^2} \sum_{i \in \mathcal{N}} \left| g_i(\widetilde{\lambda}^t_i) - g_i(\lambda^{\pi_{\theta^t}}_i) \right|^2_2 \\
&\leq \frac{M_r^2}{n^2} \sum_{i \in \mathcal{N}} \left\| \widetilde{\lambda}^t_i - \lambda^{\pi_{\theta^t}}_i \right\|^2_2,
\end{aligned}
$$

where we use the mean value theorem and (82) in the last line. Thus, when (52a) holds, which happens with probability $1 - n\delta_0$, we have that

$$
\left\| \widetilde{\nabla}_\mu \mathcal{L}(\theta^t, \mu^t) - \nabla_\mu \mathcal{L}(\theta^t, \mu^t) \right\|^2_2 \leq \frac{M_r^2}{n^2} \cdot n \big( \epsilon_1(\delta_0) \big)^2 = \frac{(M_r \epsilon_1(\delta_0))^2}{n},
$$

which completes the proof or (52d). $\qquad \square$

*Proof of* (52e). Similar to the proof of (52a), we denote $\mathcal{F}^{t-1}$ as the $\sigma$-algebra generated by all trajectories sampled at $0, 1, \ldots, t - 1$-th periods. For each $i \in \mathcal{N}$, let

$$
\mathcal{L}^t_i := \frac{1}{B} \sum_{\tau \in \mathcal{B}^t_i} \left[ \sum_{k=0}^{H-1} \gamma^k \nabla_{\theta_i} \log \pi^i_{\theta_i}(a^k_i | s^k_{\mathcal{N}^\kappa_i}) \cdot \frac{1}{n} \sum_{j \in \mathcal{N}^\kappa_i} \left[ \widehat{Q}^{\pi_{\theta^t}}_{f_j}(s^k_{\mathcal{N}^\kappa_j}, a^k_{\mathcal{N}^\kappa_j}) + \mu^{t+1}_j \widehat{Q}^{\pi_{\theta^t}}_{g_j}(s^k_{\mathcal{N}^\kappa_j}, a^k_{\mathcal{N}^\kappa_j}) \right] \right].
\tag{83}
$$

Note that the only distinction between $\mathcal{L}^t_i$ and $\widetilde{\nabla}_{\theta_i} \mathcal{L}(\theta^t, \mu^{t+1})$ is in the Q-function term, where we use the true truncated Q-functions in the definition of $\mathcal{L}^t_i$. Then, the difference

$\left\| \widetilde{\nabla}_{\theta_i} \mathcal{L}(\theta^t, \mu^{t+1}) - \nabla_{\theta_i} \mathcal{L}(\theta^t, \mu^{t+1}) \right\|_2^2$ can be decomposed into the following four parts

$$\left\| \widetilde{\nabla}_{\theta_i} \mathcal{L}(\theta^t, \mu^{t+1}) - \nabla_{\theta_i} \mathcal{L}(\theta^t, \mu^{t+1}) \right\|_2^2 \le 4 \bigg[ \underbrace{\left\| \widetilde{\nabla}_{\theta_i} \mathcal{L}(\theta^t, \mu^{t+1}) - \mathcal{L}_i^t \right\|_2^2}_{\mathcal{T}_1} + \underbrace{\left\| \mathcal{L}_i^t - \mathbb{E}\left[ \mathcal{L}_i^t | \mathcal{F}^{t-1} \right] \right\|_2^2}_{\mathcal{T}_2}$$

$$+ \underbrace{\left\| \mathbb{E}\left[ \mathcal{L}_i^t | \mathcal{F}^{t-1} \right] - \widehat{\nabla}_{\theta_i} \mathcal{L}(\theta^t, \mu^{t+1}) \right\|_2^2}_{\mathcal{T}_3} \tag{84}$$

$$+ \underbrace{\left\| \widehat{\nabla}_{\theta_i} \mathcal{L}(\theta^t, \mu^{t+1}) - \nabla_{\theta_i} \mathcal{L}(\theta^t, \mu^{t+1}) \right\|_2^2}_{\mathcal{T}_4} \bigg],$$

where we used the inequality that $\left\| \sum_{j=1}^J x_j \right\|_2^2 \le J \sum_{j=1}^J \|x_j\|_2^2$. Below, we separately upper-bound the terms $\mathcal{T}_1$ - $\mathcal{T}_4$. Firstly, by the boundedness of score function (see Assumption 4.2) and the dual variable, we can write that

$$\mathcal{T}_1 \le \left\| \frac{1}{B} \sum_{\tau \in \mathcal{B}_i^t} \left[ \sum_{k=0}^{H-1} \gamma^k \nabla_{\theta_i} \log \pi_{\theta_i}^i (a_i^k | s_{\mathcal{N}_i^\kappa}^k) \cdot \frac{1}{n} \sum_{j \in \mathcal{N}_i^\kappa} \left[ \left\| \widetilde{Q}_{f_j}^t - \widehat{Q}_{f_j}^{\pi_{\theta^t}} \right\|_\infty + |\mu_j^{t+1}| \left\| \widetilde{Q}_{g_j}^t - \widehat{Q}_{g_j}^{\pi_{\theta^t}} \right\|_2 \right] \right] \right\|_2^2$$

$$\le \left\| \frac{1}{B} \sum_{\tau \in \mathcal{B}_i^t} \left[ \sum_{k=0}^{H-1} \gamma^k \nabla_{\theta_i} \log \pi_{\theta_i}^i (a_i^k | s_{\mathcal{N}_i^\kappa}^k) \right] \right\|_2^2 \cdot \left[ \frac{|\mathcal{N}_i^\kappa|(1+\overline{\mu})}{n} \left( M_r \epsilon_0 + \frac{L_\lambda \epsilon_1(\delta_0)}{1-\gamma} \right) \right]^2$$

$$\le \left( \frac{M_\pi}{1-\gamma} \right)^2 \cdot \left[ \frac{|\mathcal{N}_i^\kappa|(1+\overline{\mu})}{n} \left( M_r \epsilon_0 + \frac{L_\lambda \epsilon_1(\delta_0)}{1-\gamma} \right) \right]^2$$

$$= \left[ \frac{|\mathcal{N}_i^\kappa|(1+\overline{\mu}) M_\pi}{n(1-\gamma)} \left( M_r \epsilon_0 + \frac{L_\lambda \epsilon_1(\delta_0)}{1-\gamma} \right) \right]^2, \tag{85}$$

where we assume the upper bound in (52c) holds in the second inequality, which happens with probability $1 - n\delta_0$.

To upper-bound $\mathcal{T}_2$, we use a similar argument as (78). For any trajectory $\tau = \{(s^0, a^0), \cdots, (s^{H-1}, a^{H-1})\}$ of length $H$, we define the shorthand notation $\mathcal{G}_i(\tau)$ as

$$\mathcal{G}_i(\tau) := \sum_{k=0}^{H-1} \gamma^k \nabla_{\theta_i} \log \pi_{\theta_i}^i (a_i^k | s_{\mathcal{N}_i^\kappa}^k) \cdot \frac{1}{n} \sum_{j \in \mathcal{N}_i^\kappa} \left[ \widehat{Q}_{f_j}^{\pi_{\theta^t}} (s_{\mathcal{N}_j^\kappa}^k, a_{\mathcal{N}_j^\kappa}^k) + \mu_j^{t+1} \widehat{Q}_{g_j}^{\pi_{\theta^t}} (s_{\mathcal{N}_j^\kappa}^k, a_{\mathcal{N}_j^\kappa}^k) \right]. \tag{86}$$

Then, it is clear from (83) that $\mathcal{L}_i^t = 1/B \cdot \sum_{\tau \in \mathcal{B}_i^t} \mathcal{G}_i(\tau)$. By the boundedness of the score function, dual variable, and the shadow Q-function, we have that

$$\|\mathcal{G}_i(\tau)\|_2^2 \le \left( \frac{|\mathcal{N}_i^\kappa|(1+\overline{\mu}) M_r}{n(1-\gamma)} \right)^2 \cdot \left\| \sum_{k=0}^{H-1} \gamma^k \nabla_{\theta_i} \log \pi_{\theta_i}^i (a_i^k | s_{\mathcal{N}_i^\kappa}^k) \right\|_2^2$$

$$\le \left( \frac{|\mathcal{N}_i^\kappa|(1+\overline{\mu}) M_r}{n(1-\gamma)} \right)^2 \cdot \frac{M_\pi^2}{(1-\gamma)^2} \tag{87}$$

$$= \left( \frac{|\mathcal{N}_i^\kappa|(1+\overline{\mu}) M_r M_\pi}{n(1-\gamma)^2} \right)^2,$$

where we bound the Q-function terms in the first step and apply the boundedness of the score function in the second step. Again, it follows from [88, Lemma 18] that with probability $1 - \delta_0$

$$\mathcal{T}_2 = \left\| \mathcal{L}_i^t - \mathbb{E}\left[ \mathcal{L}_i^t | \mathcal{F}^{t-1} \right] \right\|_2^2 \le \frac{2 - 8 \log \delta_0}{B} \left( \frac{|\mathcal{N}_i^\kappa|(1+\overline{\mu}) M_r M_\pi}{n(1-\gamma)^2} \right)^2. \tag{88}$$

To upper-bound $\mathcal{T}_3$, which is the error due to trajectory truncation, we have that

$$
\begin{aligned}
\mathcal{T}_3 &= \left\| \mathbb{E}\left[ 1/B \cdot \sum_{\tau \in \mathcal{B}_i^t} \mathcal{G}_i(\tau) \middle| \mathcal{F}^{t-1} \right] - \widehat{\nabla}_{\theta_i} \mathcal{L}(\theta^t, \mu^{t+1}) \right\|_2^2 \\
&= \left\| \mathbb{E}\left[ \mathcal{G}_i(\tau) \middle| \mathcal{F}^{t-1} \right] - \widehat{\nabla}_{\theta_i} \mathcal{L}(\theta^t, \mu^{t+1}) \right\|_2^2 \\
&\overset{(\triangle)}{=} \left\| \mathbb{E}\left[ \sum_{k=H}^{\infty} \gamma^k \nabla_{\theta_i} \log \pi_{\theta_i}^i(a_i^k | s_{\mathcal{N}_i^\kappa}^k) \cdot \frac{1}{n} \sum_{j \in \mathcal{N}_i^\kappa} \left( \widehat{Q}_{f_j}^{\pi_\theta}(s_{\mathcal{N}_j^\kappa}^k, a_{\mathcal{N}_j^\kappa}^k) + \mu_j \widehat{Q}_{g_j}^{\pi_\theta}(s_{\mathcal{N}_j^\kappa}^k, a_{\mathcal{N}_j^\kappa}^k) \right) \right] \right\|_2^2 \quad (89) \\
&\leq \left[ \frac{M_\pi \gamma^H}{1-\gamma} \cdot \frac{|\mathcal{N}_i^\kappa|(1+\overline{\mu})M_r}{n(1-\gamma)} \right]^2 \\
&= \left[ \frac{\gamma^H |\mathcal{N}_i^\kappa|(1+\overline{\mu})M_r M_\pi}{n(1-\gamma)^2} \right]^2,
\end{aligned}
$$

where $(\triangle)$ follows from the definition of the truncated policy gradient $\widehat{\nabla}_{\theta_i} \mathcal{L}(\theta^t, \mu^{t+1})$ (see (D)) and the inequality is due to a similar argument as (87).

Finally, the upper bound of the last term $\mathcal{T}_4$ is provided in Lemma 3.5, and it holds that

$$
\mathcal{T}_4 = \left\| \widehat{\nabla}_{\theta_i} \mathcal{L}(\theta^t, \mu^{t+1}) - \nabla_{\theta_i} \mathcal{L}(\theta^t, \mu^{t+1}) \right\|_2^2 \leq \left[ \frac{(1+\|\mu\|_\infty)M_\pi c_0 \phi_0^\kappa}{1-\gamma} \right]^2 \leq \left[ \frac{(1+\overline{\mu})M_\pi c_0 \phi_0^\kappa}{1-\gamma} \right]^2. \quad (90)
$$

Together, by substituting (85), (88), (89), and (90) into (84), we derive that

$$
\begin{aligned}
&\left\| \widetilde{\nabla}_{\theta_i} \mathcal{L}(\theta^t, \mu^{t+1}) - \nabla_{\theta_i} \mathcal{L}(\theta^t, \mu^{t+1}) \right\|_2^2 \\
&\leq 4 \Bigg\{ \left[ \frac{|\mathcal{N}_i^\kappa|(1+\overline{\mu})M_\pi}{n(1-\gamma)} \left( M_r \epsilon_0 + \frac{L_\lambda \epsilon_1(\delta_0)}{1-\gamma} \right) \right]^2 + \frac{2 - 8 \log \delta_0}{B} \left( \frac{|\mathcal{N}_i^\kappa|(1+\overline{\mu})M_r M_\pi}{n(1-\gamma)^2} \right)^2 \\
&\qquad + \left[ \frac{\gamma^H |\mathcal{N}_i^\kappa|(1+\overline{\mu})M_r M_\pi}{n(1-\gamma)^2} \right]^2 + \left[ \frac{(1+\overline{\mu})M_\pi c_0 \phi_0^\kappa}{1-\gamma} \right]^2 \Bigg\} \\
&= \frac{|\mathcal{N}_i^\kappa|^2}{n^2} \cdot 4 \left[ \frac{(1+\overline{\mu})M_r M_\pi}{(1-\gamma)^2} \right]^2 \cdot \underbrace{\left[ \left( (1-\gamma)\epsilon_0 + \frac{L_\lambda \epsilon_1(\delta_0)}{M_r} \right)^2 + \frac{2-8\log\delta_0}{B} + \gamma^{2H} \right]}_{\epsilon_2(\delta_0)} \quad (91) \\
&\quad + \underbrace{4\left[ \frac{(1+\overline{\mu})M_\pi c_0 \phi_0^\kappa}{1-\gamma} \right]^2}_{\epsilon_3} \\
&= \frac{|\mathcal{N}_i^\kappa|^2}{n^2} \epsilon_2(\delta_0) + \epsilon_3.
\end{aligned}
$$

Note that, when (52c) is satisfied (which has probability $1 - n\delta_0$), (91) has a failure probability of $\delta_0$ due to the probabilistic bound (88). Thus, by applying the union bound, we can conclude that (91) holds for all agent $i \in \mathcal{N}$ with probability $1 - 2n\delta_0$. Recall that $\widetilde{\nabla}_\theta \mathcal{L}(\theta^t, \mu^{t+1})$ is defined as the concatenation of local estimators $\{\widetilde{\nabla}_{\theta_i} \mathcal{L}(\theta^t, \mu^{t+1})\}_{i \in \mathcal{N}}$. We conclude that with probability $1 - 2n\delta_0$, it holds that

$$
\begin{aligned}
\left\| \widetilde{\nabla}_\theta \mathcal{L}(\theta^t, \mu^{t+1}) - \nabla_\theta \mathcal{L}(\theta^t, \mu^{t+1}) \right\|_2^2 &= \sum_{i \in \mathcal{N}} \left\| \widetilde{\nabla}_{\theta_i} \mathcal{L}(\theta^t, \mu^{t+1}) - \nabla_{\theta_i} \mathcal{L}(\theta^t, \mu^{t+1}) \right\|_2^2 \\
&\leq \left( \sum_{i \in \mathcal{N}} \frac{|\mathcal{N}_i^\kappa|^2}{n^2} \right) \epsilon_2(\delta_0) + n\epsilon_3.
\end{aligned} \quad (92)
$$

Finally, we remark that by definitions $\epsilon_3 = \mathcal{O}\left( \phi_0^{2\kappa} \right)$ and

$$
\epsilon_2(\delta_0) = \mathcal{O}\left( \epsilon_0^2 + (\epsilon_1(\delta_0))^2 + \frac{\log(1/\delta_0)}{B} + \gamma^{2H} \right) = \mathcal{O}\left( \epsilon_0^2 + \frac{\log(1/\delta_0)}{B} + \gamma^{2H} \right), \quad (93)
$$

which completes the proof.

$\square$

# H   Numerical experiments

In this section, we provide details on the experimental results. First, in Appendices H.1-H.3, we separately introduce the three environments considered in this work and discuss the performance of Algorithm 1 on these environments. Then, in Appendix H.4, we compare Algorithm 1 with three baselines based on the MAPPO-Lagrangian method by [31] in two standard safe MARL problems. Finally, in Appendix H.5, we illustrate the effectiveness of employing general utilities.

## H.1   Synthetic environment

Consider an environment similar to that of [24, Section 5.1], where the agents are placed along a line, i.e., $1 - 2 - \cdots - n$. The local state and action spaces of every agent $i$ are binary, i.e., $\mathcal{S}_i = \mathcal{A}_i = \{0, 1\}$, with the transition dynamics specified as follows:

- For agent 1, $s_1^{t+1} = 1$ if and only if $s_2^t = 1$.
- For agent $n$, $s_n^{t+1} = 1$ if and only if $a_n^t = 1$.
- For every agent $i \in \mathcal{N} \backslash \{1, n\}$, the local transition probability $\mathbb{P}_i$ is specified by

$$\mathbb{P}_i(s_i^{t+1} = 1 | s^t, a^t) = \begin{cases} 1, & \text{if } a_i^t = 1, s_{i+1}^t = 1 \\ 0.8, & \text{if } a_i^t = 1, s_{i+1}^t = 0 \\ 0, & \text{otherwise.} \end{cases}$$

The goal of the agents is to jointly maximize a cumulative reward while complying with the exploration requirements, which can be formulated as

$$\max_{\theta \in \Theta} \sum_{i \in \mathcal{N}} \left\langle \lambda_i^{\pi_\theta}, r_i \right\rangle, \text{ s.t. } \text{Entropy}(\lambda_i^{\pi_\theta}) \geq c, \ \forall i \in \mathcal{N}, \tag{94}$$

where the local rewards only depend on the states of the agents with $r_1(1) = 1$ and $r_i(1) = 0.1$, $\forall i \in \mathcal{N} \backslash \{1\}$. In all other cases, the reward is 0. The function $\text{Entropy}(\lambda_i^{\pi_\theta}) = -\sum_{s \in \mathcal{S}} d_i^\pi(s) \cdot \log \left( d_i^\pi(s) \right)$ refers to the local entropy. Without the constraint, it is clear that the optimal policy is that all agents take action 1 regardless of their states. However, the optimality of this policy is compromised in the presence of the entropy constraint, since the agents are restricted from taking the same action all the time.

### H.1.1   Experimental results

In the experiment, we consider $n = 10$ and $\gamma = 0.99$. The results are plotted in Figure 3, where we evaluate the performance of Algorithm 1 using episodic return and total constraint violation as metrics. The agents are initialized with random policies, resulting in a high entropy during the early stages of training. As a result, the constraints are always being strictly satisfied at the beginning. As the agents strive to increase their cumulative reward, they gradually begin to take action 1 more frequently and spend more time in state 1, which results in a decrease in entropy. Eventually, the agents find a balance between maximizing the cumulative reward and satisfying the entropy constraint.

Below, we discuss the experiment results shown in Figure 3.

**Different communication ranges**   In this experiment, we test the algorithm with communication radius $\kappa \in \{0, 1, 2, 5\}$. We note that the case $\kappa = 5$ is close to global observation for agents in the middle. The results demonstrate that $\kappa = 1, 2, 5$ exhibit comparable performances, i.e., a stricter restriction on the communication range does not compromise the optimality in this environment, which is due to the simplicity of the environment. The case with no communication ($\kappa = 0$) is slightly worse than others and suffers from a higher constraint violation during training.

**Different constraint RHS values**   In addition, we also vary the values for the threshold of the entropy constraints. A larger threshold value implies a stronger requirement for exploration, which subsequently results in a lower cumulative reward since the agents only receive rewards when their

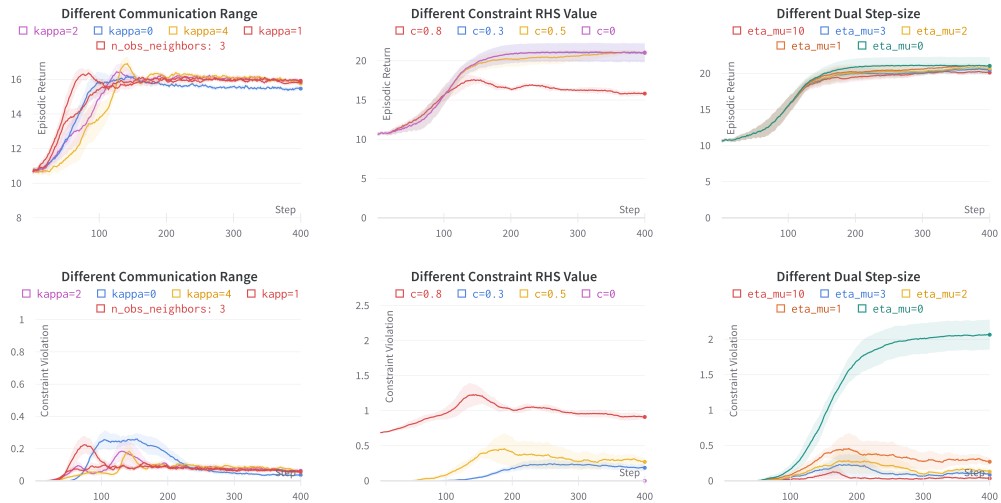

Figure 3: Performance of Algorithm 1 in synthetic environment with 10 agents under entropy constraints. **Left**: different communication ranges. **Middle**: different constraint right-hand side (RHS) values. **Right**: different dual step-sizes.

states are equal to 1. As seen in the two middle plots of Figure 3, the experimental results uphold this argument.

**Different dual step-sizes**   Furthermore, we test how the size of the dual step-size/regularization-weight $\eta_\mu$ influences the learning process. The results show that when $\eta_\mu$ is reasonably large, e.g., $\eta_\mu \geq \epsilon \{2, 3, 10\}$, the performances of the algorithm are roughly the same. Notably, we observe that having a large $\eta_\mu$ not only ensures a low constraint violation for the learned policy, but also guarantees a low violation during the training stage. On the other side, a small $\eta_\mu$, e.g., $\eta_\mu \in \{0, 1\}$ may not provide enough incentive to offset the violated constraints.

## H.2   Pistonball environment

The Pistonball [49] is a physics-based cooperative game where each piston at the bottom represents an agent (see Figures 5 and 6). The agents naturally form a network where there is an edge between two adjacent pistons. The agents' goal is to collaboratively move the ball from the right wall to the left while satisfying the exploratory constraint defined by an entropy function. The action space $\mathcal{A}_i$ of each agent $i$ contains three elements: moving four pixels up, moving four pixels down, and remaining still. The local state space $\mathcal{S}_i$ consists of two components: the $y$-position of agent $i$ and its observed information of the ball, which is a five tuple, namely the ball's $x$-position, $y$-position, $x$-velocity, $y$-velocity, and angular velocity. Each agent $i$ can only observe the ball when it enters the space above itself, otherwise the agent receives a binary value indicating whether the ball is to its left or to its right.

The local reward function $r_i$ is constructed such that agent $i$ can receive a non-zero reward (penalty) only if any part of the ball is above itself at the current or the last time step. The size of reward (penalty) depends on the change in the ball's x-position, where a rightwards move receives a penalty of twice the size of the reward for a leftwards move. When the ball stays at the same place for over three steps, the agents below will receive a negative time penalty. Mathematically, the problem can be formulated as:

$$\max_{\theta \in \Theta} \frac{1}{n} \sum_{i \in \mathcal{N}} \langle \lambda_i^{\pi_\theta}, r_i \rangle, \ \text{s.t. Entropy}(\lambda_i^{\pi_\theta}) \geq c, \ \forall i \in \mathcal{N}, \tag{95}$$

where we use a common constraint threshold $c$ for all agents.

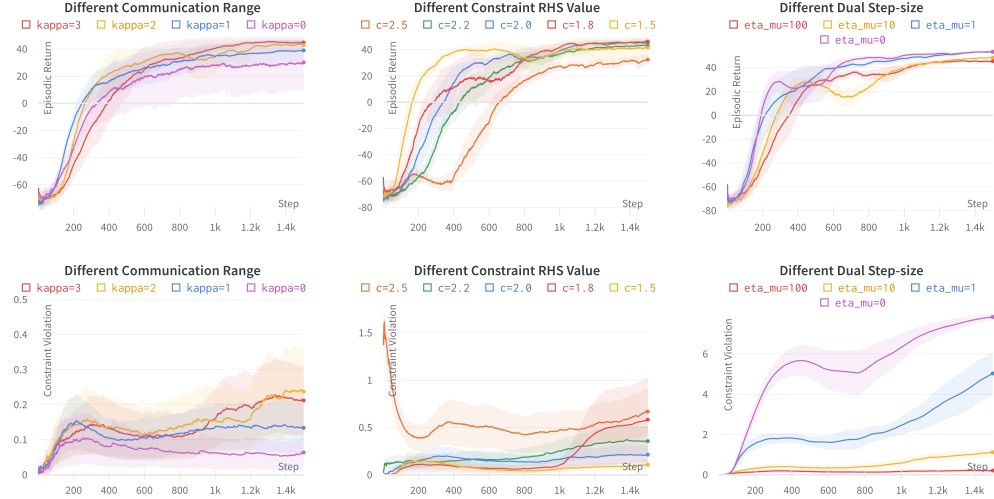

Figure 4: Performance of Algorithm 1 in the Pistonball environment with 10 agents under entropy constraints. **Left**: different communication range. **Middle**: different constraint RHS value. **Right**: different dual step-size. The total constraint violation is defined as the sum of absolute violations for each local constraints.

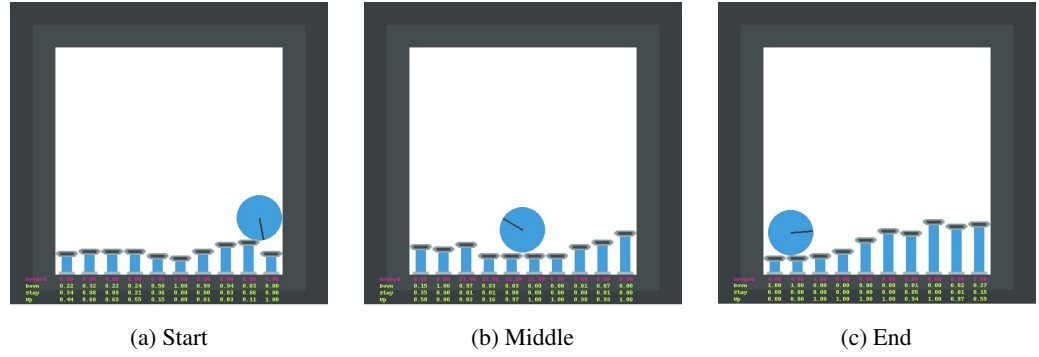

(a) Start                    (b) Middle                    (c) End

Figure 5: Visualization of Pistonball environment at three different stages when executing the learned policy.

### H.2.1    Experimental results

We consider the scenario of 10 agents and label them by $\{1, 2, \ldots, 10\}$ from right to left. The experimental results are summarized in Figure 4, where we test the proposed algorithm based on different communication ranges, constraint RHS values, and dual step-sizes. The algorithm's performance is evaluated using two metrics: the cumulative reward and absolute constraint violation of the learned policy. Below, we first present some general observations, followed by individual discussions of three comparisons.

**General observations.**

- The safety constraint plays an important role in this environment. Without the constraint, sometimes the learning process is trapped in sub-optimal policies. A common local optimum is that all agents move to the lowest position and keep staying there. In this situation, the ball can still move to the left at a significantly slower pace, driven by its initial velocity, and agents will receive some rewards. By incorporating a mild entropy constraint (e.g., $c = 1.8$), agents are encouraged to explore the environment and escape sub-optimal policies. However, our comparison of different right-hand side values below also reveals that the optimality of the learned policy can be compromised if the

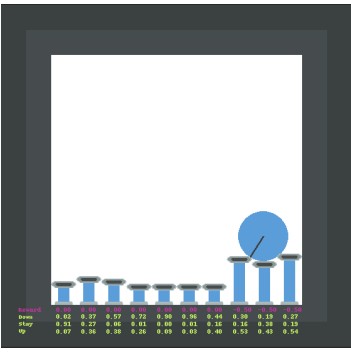

Figure 6: Illustration of the benefit of having a relatively-larger communication range ($\kappa = 2$). The agents on the right make a sacrifice by intentionally raised the ball all the way up to provide more flexibility for agents on their left.

exploration requirement is too strict. These two findings highlight the trade-off between *exploration* and *exploitation*.

It is worth noting that, although encouraging exploration is a common practice in RL, our formulation allows for the direct incorporation of the entropy of the occupancy measure since we allow the objective and constraint to be general utilities. Compared with standard approaches, such as adding a discounted entropy with respect to the policy in the objective [89], our approach provides a more explicit characterization for the exploration requirement.

- We visualize the learned policy with $\kappa = 3$, $c = 2$, and $\eta_\mu = 100$ in Figure 5, considering three different time points where the ball is located in the right-most region, middle region, and left-most region, respectively. The policy (action probability) of agents for the given state is displayed by the text at the bottom of the figure.

As shown in Figure 5a, agents' positions are initialized randomly at the beginning. To facilitate the ball's leftward movement, agent 1 must move upwards, while agent 2 should move downwards. This is confirmed by the current policies of the two agents, where the upward probability of agent 1 is one, and the downward probability of agent 2 is 0.83. Subsequently, Figure 5b demonstrates that agents $1 - 4$ have created a slope for the ball to move leftward rapidly. After the ball passes, we can see that the upward probabilities of agents $1 - 4$ are very close to one, meaning that they move upwards to eliminate the possibility of the ball moving back to the right. However, agents $8 - 10$ still obstruct the ball's path, as they have not detected the arrival of the ball due to the limited communication range and move mostly randomly to satisfy the entropy constraint. Finally, in Figure 5c, we observe that when the ball approaches, the downward probabilities of agents $9 - 10$ become one, and the upward probabilities of agents $5 - 8$ also increase to one.

**Different communication ranges.** We alter the communication radius from $\kappa = 0$ to 3, keeping other parameters constant. The results indicate that $\kappa = 2$ or 3 yields better performance, while disallowing any communication ($\kappa = 0$) results in lower rewards. This outcome can be attributed to the fact that, for the efficient movement of the ball from right to left, each agent must maintain the correct position before the ball's arrival. With no communication, agents are unaware of the ball's arrival in advance, and they mostly move randomly to fulfill the exploration requirement. Additionally, when agents cannot share their local shadow Q-functions with neighbors, they may end up learning a "selfish" policy focused solely on their own objectives. A larger communication radius not only enables agents to observe the ball earlier but also allows some agents to perform actions that assist other agents. In Figure 6, we observe that agents $1 - 3$ decide to move the ball all the way up. Despite incurring a time penalty for themselves, this provides more flexibility for agent 4 and allows more time to take random actions in order to satisfy the safety constraint. However, as a trade-off, a larger communication range also implies a larger input size. Therefore, the convergence rate is generally slower for a larger communication range when the same hyperparameters are used (see the comparison of $\kappa = 1$ and $\kappa = 3$ in Figure 4 at early stages).

In contrast to the objective, we find the constraint violation remains relatively low in all cases. This is because the entropy constraint encourages each individual agent to actively explore the environment,

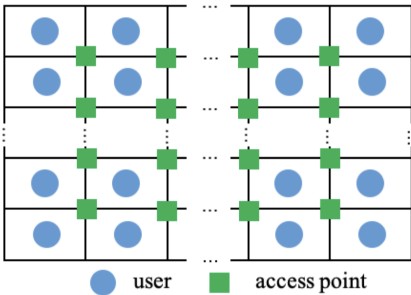

Figure 7: Wireless communication network with $n^2$ agents and $(n-1)^2$ access points [24].

enabling the agents to find ways to keep the constraint violations low under different communication ranges.

**Different constraint RHS values.** Here, we run the algorithm with different constraint RHS values $c \in \{1.5, 1.8, 2.0, 2.2, 2.5\}$. In the middle two plots of Figure 4, we observe that increasing $c$ from 1.5 to 2.0 yields a policy with higher rewards. This occurs because a slightly stricter exploration requirement helps the algorithm avoid sub-optimal stationary points and discover a superior policy (as explained in general observations). However, further increment of the constraint right-hand side value (from 2.0 to 2.5) forces the agents to make many unnecessary moves to meet the constraint, hindering the effective transfer of the ball to the left.

**Different dual step-sizes.** Finally, we test four different values $\{0, 1, 10, 100\}$ of dual step-size $\eta_\mu$. The result in the lower two plots in Figure 4 demonstrates that a larger value of $\eta_\mu$ yields a smaller constraint violation. Since $\eta_\mu$ serves as the weight of penalization of the constraint violation in the Lagrangian function, this observation is consistent with the developed theory in this paper. On the other side, we can observe that the objective is slightly lower for larger values of $\eta_\mu$, which can be viewed as a compromise in exchange for a better-satisfied constraint.

### H.3 Wireless communication environment

Consider an access control problem with safety constraints in wireless communication, following a similar network setup and transition dynamics as presented in [24, 50]. Specifically, we consider a grid with $n^2$ users (agents) $\mathcal{N} = [n] \times [n]$ and $(n-1)^2$ access points $Y$, as illustrated in Figure 7. The goal of the users is to successfully transmit their packets to access points for processing. Each user $i$ is connected to a set $Y_i \subset Y$ of access points located at the corner of the block it resides in. Two users are considered direct neighbors if they share a common access point. In every period, user $i$ receives a new packet by deadline $d_i$ with probability $p_i \in (0, 1)$. The user can then choose to send the earliest packet in its queue to an access point $y \in Y_i$ or not send any packet at all. User $i$ receives a reward 1 if and only if access point $y$ does not receive transmissions from other users and successfully processes the packet from $i$, which occurs with probability $q_y \in (0, 1)$.

When formulated as a standard RL problem, the state of each user $i$ is defined by a $d_i$-dimensional vector with binary values, i.e., $s_i \in \{0, 1\}^{d_i}$. The $k$-th entry of $s_i$ takes the value 1 when user $i$ currently has a packet with $k$ days remaining until the deadline. The action space of user $i$ is defined as $\mathcal{A}_i = Y_i \cup \{\text{null}\}$, which means agent $i$ can choose to send the packet to an access point $y \in Y_i$ or do nothing. It is important to note that the local transition dynamic and local reward function of each user depend on the states and actions of other users in its neighborhood. This slightly differs from the setting presented in our paper, as we assume $r_i : \mathcal{S}_i \times \mathcal{A}_i \to \mathbb{R}$. However, since this objective takes the form of cumulative reward (rather than general utilities), our analysis can be extended to settings where the local reward $r_i$ depends on $(s_{\mathcal{N}_i}, a_{\mathcal{N}_i})$.

In this experiment, safety is a critical concern. More specifically, potential risks may arise when agents learn overly randomized policies, causing the neighbors failing to know which access points will be occupied and thereby resulting in a collision. This resonates with real-life applications such as autonomous driving and human-AI collaboration, where *an agent's policy needs to be predictable to other agents*. In light of this, we introduce an additional safety constraint, $1/2 \cdot (1-\gamma)^2 \cdot \|\lambda^{\pi_\theta}\|_2^2 \geq c$,

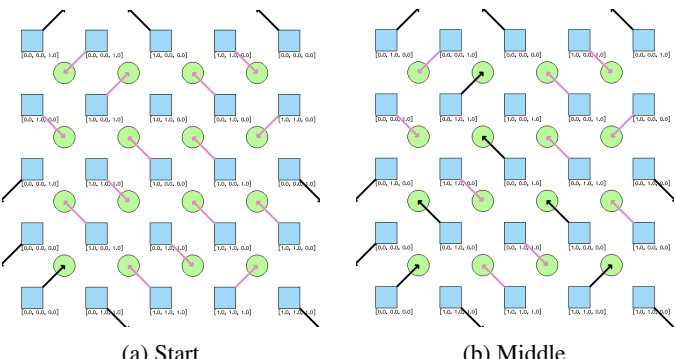

| (a) Start | (b) Middle |

Figure 8: Two consecutive frames from the wireless communication experiment when $\eta = 100$, rhs = 0.3. The agents are encouraged to take deterministic actions. Pink arrows indicate successful transmissions, and the binary integers below each agent indicates its state.

to encourage agents to learn less randomized policies. The term $(1 - \gamma)^2$ serves as a normalization constant. In summary, the problem can be formulated as:

$$\max_{\theta \in \Theta} \frac{1}{n} \sum_{i \in \mathcal{N}} V^{\pi_\theta}(r_i), \text{ s.t. } \frac{(1-\gamma)^2}{2} \left\| \sum_{s_i \in \mathcal{S}_i} \lambda_i^{\pi_\theta} \right\|_2^2 \geq c, \; \forall i \in \mathcal{N}. \tag{96}$$

### H.3.1 Experimental results

In our experiments, we consider a setting with $n = 5$ (comprising 25 agents and 16 access points) and $d_i = 3$. Probabilities $p_i$ and $q_y$ are randomly generated. We perform the same set of comparisons based on various communication ranges, constraint RHS values, and dual step-sizes. The experimental results are illustrated in Figure 9, with key findings summarized as follows:

- The performance of the algorithm with $\kappa = 1$ clearly surpasses that of $\kappa = 0$. This highlights the critical role of communication in situations where potential conflicts between neighbors can occur.

- Unlike the Pistonball environment, discouraging exploration via constraints leads to an improved performance in this example ($c = 0.3$ yields higher return than $c = 0.2$). This can be explained by the fact that when all agents strive to learn less-randomized policies, their actions become more predictable for other agents, thus minimizing the conflicts. As shown in Figure 8, agents always take the same actions. Some agents even choose to forfeit their own packets, either by not taking actions or selecting non-existent access points, as a strategy to minimize the overall collisions within the environment.

  Our model lets the agents learn about the behaviors of other agents, thereby facilitating understanding of the collective interplay between the locations and actions of those agents. Remarkably, the agents are able to collaboratively identify a plan so that each access point is only used by one agent in order to avoid collision in Figure 8.

- The final two plots in Figure 9 confirm that a relatively large dual step-size is needed in order to achieve a good performance. It is important to note that the policy learned with $\eta = 1$ significantly violates the constraint, while the policy learned with $\eta = 10$ gets trapped in some sub-optimal points.

### H.4 Baseline comparison

We emphasize that our method distinguishes itself from existing approaches like MAPPO-Lagrangian (MAPPO-L) [31], as we allow for the objective and constraints to take the form of general utilities, i.e., nonlinear functions of the occupancy measure. The adoption of general utilities enable our formulation cover a wider range of problems (as discussed in Section 2 and Appendix H.5), but also renders the existing analysis inapplicable.

To make fair comparisons, we consider two standard safe MARL problems, where both objectives and constrains are defined using cumulative rewards, i.e., the problem can be formulated as (30).

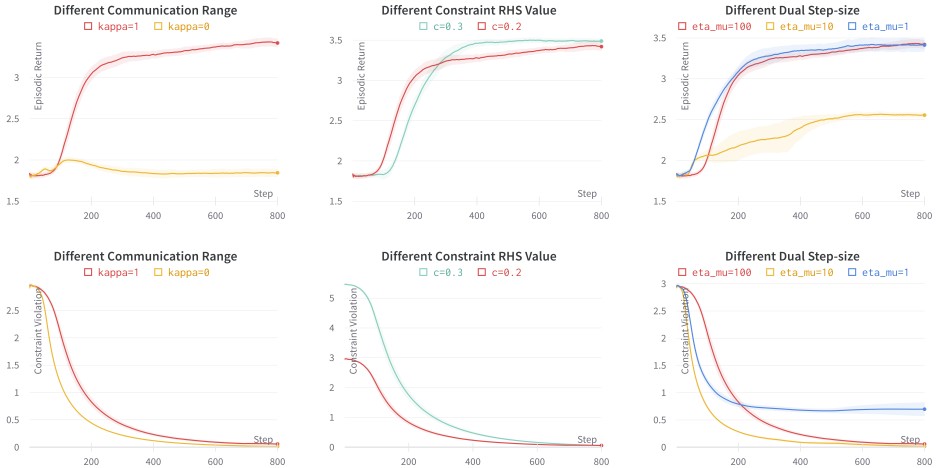

Figure 9: Performance of Algorithm 1 in wireless communication environment with 25 agents under $\ell_2$-constraints. **Left**: different communication ranges. **Middle**: different constraint RHS values. **Right**: different dual step-sizes.

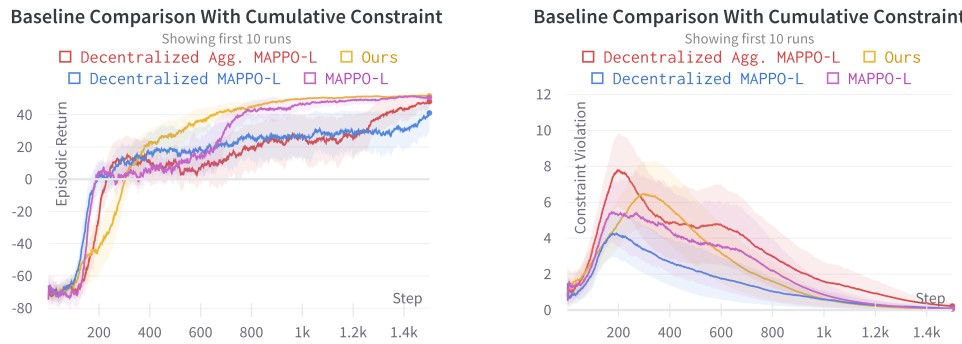

Figure 10: Comparison between Scalable Primal-Dual Actor-Critic method in our work with MAPPO-L by [31] in Pistonball.

The experiment results are illustrated in Figures 10 and 11 and Table 1. The two experiments are respectively conducted within the contexts of the Pistonball (10 agents) and wireless communication (25 agents) environments. In Pistonball, the constraints are designed to keep the pistons away from high positions: each agent $i$ receives an additional reward $u_i$ (constraint reward), proportional to its current height, and we enforce a upper bound for the cumulative reward. In wireless communication, the constraints are designed to encourage agents only taking actions when necessary: each agent $i$ receives a negative reward once it chooses to send out a packet, and we enforce a lower bound for the cumulative reward.

The original MAPPO-L is not designed for decentralized (distributed) training, as it assumes that each agent has access to global information. Therefore, we introduced three baselines based on MAPPO-L and studied their performances in the distributed settings, namely MAPPO-L, Decentralized MAPPO-L, and Decentralized Aggregate MAPPO-L (see Section 5).

From Figures 10 and 11 and Table 1, we observe that our method consistently outperforms the baselines while maintaining a satisfying constraint violation. MAPPO-L is the closest baseline in terms of performance, but it requires centralized training and access to global information. If we adapt MAPPO-L to the decentralized case, the performance quickly drops, since in Decentralized MAPPO-L, each agent is only trained to maximize its individual rewards. This is especially problematic in scenarios such as wireless communication where some agents need to make sacrifices. Even if we

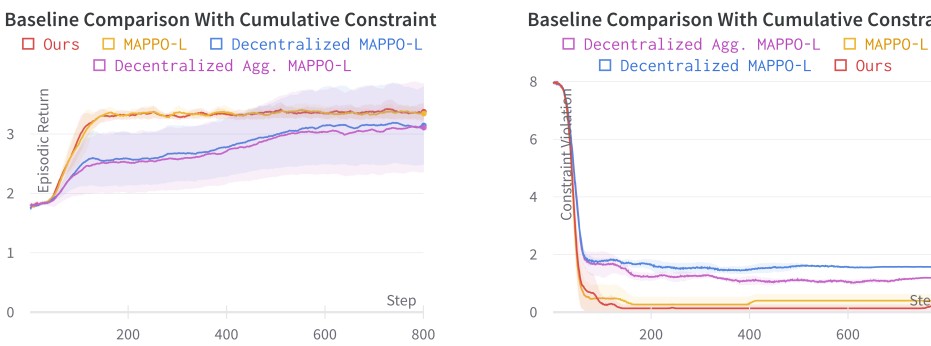

Figure 11: Comparison between Scalable Primal-Dual Actor-Critic method in our work with MAPPO-L by [31] in wireless communication.

aggregate all the rewards in the local neighborhood as in Decentralized Aggregate MAPPO-L, the overall returns are still inferior to our algorithm.

## H.5 Benefits of general utilities

Finally, we present an experiment underscoring the advantages of using general utilities. We remark that prior works such as [17, 23] have also demonstrated the power of RL with general utilities compared to traditional cumulative rewards. It is noteworthy that *using occupancy measures is not guaranteed to get higher returns*. Rather, it allows for a more extensive range of objectives and constraints and simplifies the design of cumulative reward-based schemes in certain scenarios. This versatility is particularly useful in tasks like imitation learning (where the agent's actions need to align closely with expert trajectories) and pure exploration, where designing suitable reward schemes can be challenging. Indeed, [16, Lemma 1] demonstrated that for certain MDPs, no stationary reward function could equate to a general utility.

To better illustrate the benefits of general utilities, we focus on a scenario where the constraint conflicts with the objective. We use the wireless communication environment, which requires less-randomized policies to achieve a good objective value, but this time we instead enforce a high entropy constraint (see (96)). A potential alternative for achieving this is introducing a gradient penalty term during critic training by directly deducting the next step action entropy from the policy gradient loss [90]. However, this approach suffers from the ambiguity in selecting the penalty coefficient: while a small coefficient fails to enforce the constraint, an excessively large one can impede the objective. In our experiments, we find it challenging to identify a single penalty coefficient $\lambda$ that can achieve high return while keeping the total constraint violation under one.

Figure 12 compares the performance of our method with the gradient penalty approach. Under a simple grid search, our method (shown in blue) can readily obtain a satisfactory performance while meeting the safety constraint with $\eta_\mu = 200$. Conversely, even after an extensive search for the appropriate penalty coefficient, the baseline performances are still unsatisfying. When $\lambda = 0.025$, the gradient penalty baseline is unable to match the return of our method and exceeds the constraint violation requirement. The baseline performance only begins to exceed our method at $\lambda = 0.024$, but at this point, the constraint violation significantly exceeds the threshold of one.

## H.6 Network Architecture and Hyperparameters

In this section, we specify the network architecture and hyperparameters for our algorithm. We refer the readers to our public Github repository for more details: https://github.com/CDSAC-MARL/CDSAC, where we also present the animated GIF figures of the experiments.

The hyperparameters used are summarized in Table 2. For all experiments, we perform a grid search by randomly sampling from the above list of hyperparameters for each experiment setting and choose the combination that offers the best trade-off between episodic return and constraint violation. We

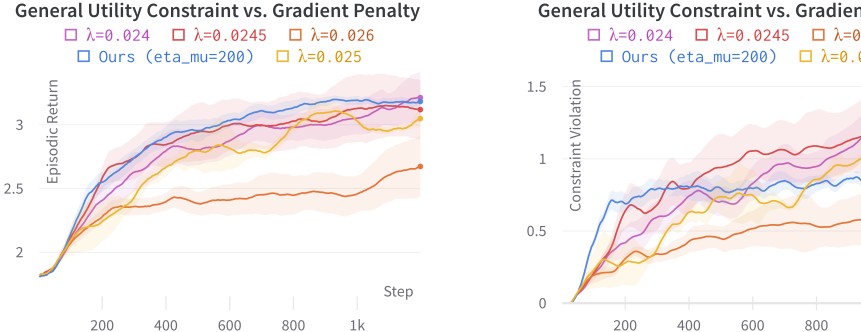

Figure 12: General utility constraint versus gradient penalty in wireless communication.

Table 2: Hyperparameters for Algorithm 1.

| Hyperparameter | Synthetic | Pistonball | Wireless Comm. |
|---|---|---|---|
| Total iterations (T) | 400 | 1500 | 800 |
| Horizon (H) | 125 | 200 | 12 |
| Number of agents | 10 | 10 | $5 \times 5$ |
| Frame stack size | 0 | 4 | 0 |
| Actor lr. | $10^{-3}$ | $\in \{10, 5, 2, 1\} \times 10^{-4}$ | $\in \{5, 1\} \times 10^{-4}$ |
| Critic lr. | $10^{-3}$ | $\in \{10, 5, 2, 1\} \times 10^{-4}$ | $\in \{10, 5\} \times 10^{-4}$ |
| Batch size (B) | 5 | $\in \{5, 8, 16\}$ | 30 |
| Q-evaluation step | 500 | $\in \{512, 1024, 1600\}$ | 512 |
| Target Q polyak | 0.95 | 0.995 | $\in \{0.95, 0.99, 0.995\}$ |
| Discount ($\gamma$) | 0.99 | $\in \{0.8, 0.9, 0.95, 0.99\}$ | $\in \{0.7, 0.8, 0.9\}$ |

then run 3-6 seeds on the chosen set of hyperparameters to produce the confidence intervals in the above figures. The experiments are produced on Tesla V100s and NVIDIA 3090s.

Below, we separately introduce the network architecture for the three environments.

**Synthetic environment**  The actor network is defined as follows:

$$(2\kappa + 1, 1) \xrightarrow{Embedding} (2\kappa + 1, 4) \xrightarrow{flatten} (4 \times (2\kappa + 1)) \xrightarrow{linear} (32) \xrightarrow{linear} (\text{num\_actions}).$$

For each agent actor, we first project the states of its $2\kappa$ neighbors along with its own state each into a vector of size $4$. We flatten the resulting embedding and additionally process it with two linear layers. The critic is defined similarly, except that we also include the actions of its $2\kappa$ neighbors along with its own action, so that the resulting vector is of size $8 \times (2\kappa + 1)$.

**Pistonball**  The actor network is defined as follows:

$$(\text{frame\_stack\_size}, 2\kappa + 6) \xrightarrow{linear} (\text{frame\_stack\_size}, h_1) \xrightarrow{flatten} (\text{frame\_stack\_size} \times h_1)$$

$$\xrightarrow{linear} (h_2) \xrightarrow{linear} (h_3) \xrightarrow{linear} (\text{num\_actions}).$$

We first process each frame with a linear layer of hidden_dim $h_1 \in \{32, 64\}$ and flatten the result as input into a stack of linear layers, where $h_2 \in \{128, 256, 512\}$ and $h_3 \in \{32, 64\}$. We use ReLu as the activation function. The critic network is defined similarly, except we additionally embed each neighbor action into a vector of size $8$ and concatenate the result together with (frame\_stack\_size $\times h_1$).

**Wireless communication**  The actor network is defined as follows:

$$(d_i, (2\kappa + 1)^2) \xrightarrow{linear} (d_i, h_1) \xrightarrow{flatten} (d_i \times h_1) \xrightarrow{linear} (h_2) \xrightarrow{linear} (h_3) \xrightarrow{linear} (\text{num\_actions}).$$

Here, $h_1, h_2, h_3$ take the same values as in the Pistonball experiment.