# OpenReview forum: "Scalable Primal-Dual Actor-Critic Method for Safe Multi-Agent RL with General Utilities"
_NeurIPS.cc/2023/Conference — NeurIPS 2023 poster_

### Official Review · Reviewer_bKHc · 2023-07-05

**Soundness:** 3 good
**Presentation:** 2 fair
**Contribution:** 1 poor
**Rating:** 5
**Confidence:** 3

**Summary:**

The work extends constrained single-agent general utilities RL (where the objective is a non-convex function of occupancy measure) to a constrained multi-agent setting. Naturally, similar to single-agent, the constraints are satisfied in expectations. They assume the combined (global) objective to be additive in terms of agents. The major novelty part of the paper is the handling of constraints (unconstrained multi-agent is a prior work). In a multi-agent setting, communication is a challenge, and to tackle this, authors utilise a k-hop type method under which agents can communicate with nearby agents.

**Strengths:**

- Extension to multi-agent constrained setting (constrained are of general utility type)
- k-hop policies (I am not sure if it is a novel contribution or from earlier work)
- Convergence analysis for both exact and stochastic setting


**Weaknesses:**

Objective: The considered global objective function is additive in agents. For a framework of general utilities RL, I would expect to consider general dependence between the agent, especially since they motivate the problem for exploration/imitation. If the objective is exploration, then the current multi-agent framework is still performing exploration individually only and it might perform differently on a combined exploration objective.

Contributions: The contributions were not so clear from [72]. If a multi-agent extension with truncated PG utilizing shadow rewards was already done and if a constrained part for a single agent existed, then I am not sure if the paper has a novel contribution.

Experiments: Since the proposed algorithm is PG type, I would expect authors to level PG tools and perform experiments with high dim state action space and complicated dynamics, e.g. car or Ant (may be continuous space, not sure if easily can be done). Since the motivation of the problem is done using exploration and imitation tasks, I think an essential component in the paper should be some experiments on these tasks and comparing performance with the case when the additivity of agents is relaxed.


**Questions:**

- I would suggest authors to have a subsection on challenges, wrt to a single agent case. It might be worth mentioning here that are not trivial single-agent to multi-agent extensions.
- Is K-hop strategy and analysis existing in prior work?
- Maybe bring related works in the main paper
- It will be good to have an algorithm block in the main paper as well.

**Limitations:**

no potential negative impact

---

> ### Author Rebuttal · Authors · 2023-08-09
>
> We thank the reviewer for the insightful questions. Please find our responses below. We highly appreciate your re-evaluation of our work and a kind reconsideration of the score.
> ### Weaknesses
> > Objective: ... exploration objective.
>
> A: We admit that our current formulation still requires structural assumptions. Yet, we believe the separable objective has its own merit and captures a wide range of applications. We refer the reviewer to the 2nd point of **General Response** for a detailed discussion.
>
> We are aware that exploration based on our formulation is not equivalent to that based on the combined objective. We attach a simple example in Figure 2 of the rebuttal PDF for showing the inequivalence. Yet, besides such marginal scenarios, maximizing the sum of local entropies usually also implies a high global entropy. In Figure 2 of the rebuttal PDF, we compare the exploration based on the separable objective and combined objective. The result shows that both methods ultimately learn a high-entropy policy.
>
> > Contributions: ... novel contribution.
>
> A: Our paper pioneers a safe MARL formulation beyond cumulative rewards, accompanied by numerical results and code release. We distinguish our contributions from [23,72] (unconstrained MARL) and [60] (CMDP with general utilities), underscoring that our results aren't merely a fusion of these works.
>
> **Compare to [23]**: Our local constraints challenge the decentralized actor-critic approach of [23], where each agent maintains a local estimation for the global objective and exchanges its estimation with neighbors through weight averaging (Eq. (12) in [23]). Due to the time-varying Lagrangian function, agents can't use multiplier-independent critics, requiring each to maintain $\mathcal{O}(n)$ critics. In addition, unlike [23]'s full observability assumption, we focus on partial observability.
>
> **Compare to [72]** While [72] employs exponential decay, our constrained setting demands novel theoretical analysis. We introduce a new primal-dual update scheme where the updated dual variable is used to update the policy (Eq. (18) and (19)). In Lagrangian, the objective and constraint functions are intertwined, necessitating decoupling to derive separate upper bounds for the primal and dual stationarity metrics (Eq. (71) and (73)).
>
> **Compare to [60]**: We refer the reviewer to our response in the **Questions** section.
> > Experiments: ... easily can be done.
>
> A: We appreciate the feedback. Indeed, we aim to show our method's efficacy in high-dimensional spaces. The complexity of Pistonball, a cooperative physics-based game, is comparable to the mentioned environments. Each local state $s_i$ is a high-dimensional vector, capturing the positions of local pistons and five continuous attributes of the ball.
>
> Classical environments like Ant/CartPole were initially tailored for single-agent, which is why we didn't opt for them. We'll explore more suitable environments and, given the chance, incorporate the findings into the final version.
>
> > Since the motivation ... additivity of agents is relaxed.
>
> A: In our experiments, we incorporated general utilities for exploration and imitation, but retained the environments' original reward design and added these utilities as constraints. The entropy constraint stems from exploration, while the $\ell_2$ constraint is the inverse of imitation. We agree that evaluating a pure exploration/imitation task and contrasting our approach with the combined objective would be insightful. Figure 2 contains some preliminary results and we plan to include more results in our updated paper.
> ### Questions
> > I would suggest ... multi-agent extensions.
>
> A: Thanks for the suggestion. We clarify the challenges and will include the discussion in the revision.
>
> First, the multi-agent setting, with its emphasis on scalability and partial observability, demands a distinct algorithmic approach compared to the single-agent scenario (more details are in Section 3 and Appendix D).
>
> Compared with [60], which addresses single-agent safe RL with general utilities, our work has several technical innovations:
> * While both use gradient-based algorithms, [60] relies on assumptions that are restrictive for general utility problems, e.g. exact evaluations of policy gradient and constraint. Our approach is more versatile, considering a sample-based setting and addressing both iteration and sample complexities.
> * Our choice of dual step-size differs from [60]. They use a very small step-size, slowing dual variable updates (see Eq. (44)---Eq. (47) in [60]). Thus, it can not quickly adapt to the violated constraints during learning. We opt for a more aggressive update, ensuring constraint violations remain minimal (see Eq. (18), lines 1004--1022 in our paper).
> * Unlike [60], where updates are sequential, i.e. $(\theta^{t+1},\mu^{t+1})\leftarrow Alg(\theta^t,\mu^t)$, our primal and dual updates are interlinked, i.e. $\mu^{t+1}\leftarrow Alg(\theta^t,\mu^t),\theta^{t+1}\leftarrow Alg(\theta^t,\mu^{t+1})$, necessitating new convergence analyses.
>
> > Is $\kappa$-hop strategy ... in prior work?
>
> A: The $\kappa$-hop policy originates from classical control theory for spatially distributed systems [R]. In MARL, we are among the first to exploit this concept. We initially name it the "$\kappa$-localized policy" as it strictly generalizes the localized policy studied in [10]. Then, we renamed it to align with concurrent works [38,60]. Our analyses, such as in Proposition 3.4, offer fresh insights into the property of $\kappa$-hop policies.
>
> The truncated Q-function concept is attributed to [10], which focused on standard value functions without safety constraints. We've expanded this by applying it to general utilities using shadow rewards. Notably, our work is the first to use the truncated Q-function within a constrained setting, differing significantly from prior unconstrained approaches.
> ### Reference
> [R] Optimal Control of Spatially Distributed Systems

---

### Official Review · Reviewer_pEsM · 2023-07-06

**Soundness:** 4 excellent
**Presentation:** 4 excellent
**Contribution:** 4 excellent
**Rating:** 7
**Confidence:** 4

**Summary:**

This paper studies the safe MARL with general utilities and focuses on the setting of distributed training without global observability. The authors propose a primal-dual actor-critic method with shadow reward and \kappa-hop policy to tackle the problems of scalability and incorporating general utilities. They provide the theoretical analysis for the convergence rate and sampling complexity. In addition, they also demonstrate the effectiveness of their method via comprehensive experiment results.


**Strengths:**

(1) Novel formulation: This paper provides a clear mathematical formulation for the MARL with general utility problems. This formulation will help the real-world multi-agent problem modeling in the general case, and encourage more discussion on the intersected domain of Safe RL and MARL.

(2) Solid theoretical analysis: This work provides solid mathematical analysis for the convergence guarantee in general-utilities settings. The analysis is rigorous as well as easy to follow. In addition, they provide a detailed discussion of the properties and show how the convergence can be guaranteed in common multi-agent scenarios.

(3) Comprehensive experimental results: The authors validate their proposed method using three experiment tasks of different categories. It is a pleasure to see the detailed task description, code implementation, and results corresponding to different settings (e.g. number of neighbors, constraint thresholds) in the submission.


**Weaknesses:**

(1) Lack of baseline comparison in the main context: I understand that the authors want to display more theoretical analysis so they may move the baseline comparison into the appendix. However, I would suggest adding this section back to the main content as it is one of the most convincing parts to show the strengths of this work.


**Questions:**

(1) Experiment settings: I am wondering how you design the experiment settings. Based on your description in section H.4, (1.1) what is the motivation to set an upper bound for the cumulative reward in the Pistonball task? (1.2) Do you mean additional cost u_i in line 1287, or this term is directly added to the reward as the penalization?


**Limitations:**

The authors adequately addressed the limitations.

---

> ### Author Rebuttal · Authors · 2023-08-09
>
> We thank the reviewer for appreciating our work. We hope our responses below provide further clarity.
> ### Weakness
> > Lack of baseline comparison in the main context: I understand that the authors want to display more theoretical analysis so they may move the baseline comparison into the appendix. However, I would suggest adding this section back to the main content as it is one of the most convincing parts to show the strengths of this work.
>
> A: We thank the reviewer for the suggestion. As we discuss in the first point of our **General Response**, we plan to shift the contents of the paper and put more experiments in the main body of the paper.
> ### Questions
> > Experiment settings: I am wondering how you design the experiment settings.
>
> A: The general purpose of our experiment design focus on the following three aspects:
> * Can our algorithm truly offer scalability to relatively large systems, e.g., over 20 agents? We consider two realistic environments, the Pistonball with 20 agents and the wireless communication with 25 agents. The results demonstrate that even a truncation with $\kappa = 1$ offers a satisfying approximation to the problem.
> * Can our algorithm truly ensures safety, even for tasks where the constraint has a direct conflict with the objective? In our experiments, the constraint is designed to compromise the unconstrained optimal policy. We have plotted the sum of absolute constraint violations across agents for all our experiments, which demonstrate that the learned policies only have a small degree of violation.
> * How does our algorithm compare with existing methods that rely on global observations? In the appendix, we compare our method with three baselines based on MAPPO-L.  The results demonstrate that our method consistently outperforms both the centralized and decentralized variants of MAPPO-L.
>
> > Based on your description in section H.4, what is the motivation to set an upper bound for the cumulative reward in the Pistonball task? Do you mean additional cost u_i in line 1287, or this term is directly added to the reward as the penalization?
>
> A: We are sorry for the confusion. Yes. Because we want to design the constraints to keep the pistons away from high positions, the right sentence in line 1287 should be "each agent $i$ receives an additional **cost** $u_i$, proportional to its current height". Then, an upper bound is enforced for the cumulative cost of each agent. We thank the reviewer for pointing out this typo, and we will correct it in the revised paper.

---

> > ### Comment · Reviewer_pEsM · 2023-08-11
> >
> > Thank you for your detailed response. Most of my concerns have been addressed. After reading others’ reviews, I would like to keep my score in favor of acceptance.

---

> > > ### Author Response · Authors · 2023-08-14
> > >
> > > Thank you very much for appreciating our work!

---

### Official Review · Reviewer_Vbe7 · 2023-07-07

**Soundness:** 4 excellent
**Presentation:** 4 excellent
**Contribution:** 4 excellent
**Rating:** 6
**Confidence:** 4

**Summary:**

This paper considers the safe multi-agent rl in the fully-decentralized setting. In the system, agents need to satisfy local safety constraints, and they also have to maximize the joint objectives. Moreover, the objectives are general utilities, which have broader applications than common discounted reward. Authors utilize shadow reward and kappa-hop neighbor truncation to limit the communication to be only among neighboring agents, which removes the need of global state and global Q function when updating each local agent. Both convergence and the sample complexity are analyzed. The experiment results also validate the effectiveness of the proposed algorithm.

**Strengths:**

This paper has the following merits.
1. The paper is well-written generally and easy to follow.
2. The kappa-hop truncation is intuitive and interesting for many real-world applications, and it also successfully eliminates the need of global state and global Q of all agents.
3. Authors establish detailed convergence and sample complexity analysis, with thorough proofs and discussions.


**Weaknesses:**

While promising, I have some concerns on the proposed algorithm. See Questions part for more details.

**Questions:**

I have the following concerns on the paper.
1.	The potential scope and real-world applications for such fully-distributed MARL are not clear. As we all know, rl usually requires heavy exploration of the environment, and it also needs lots of samples to converge. Thus, it becomes adventurous and high-cost to deploy these policies directly to the environment, and let them train in a distributed way.
2.	The communication protocol for two unconnected but neighboring agents is not clear. They can communicate directly through P2P, or they can only transmit the data agent-by-agent if the communication is limited to connected agents. Such difference is essential for the algorithm design, as the former one may generate high-cost for communication, while the latter one may bring data latency to each agent.
3.	Assumption 3.2 is too strong, as it involves the summation over all agents. The system that satisfies the assumption is nearly independent, as each agent's action has little influence on other's state transition.
4.	The algorithm has 3 major steps but relies heavily on communication. The communication complexity is not analyzed in the work, which is in fact a major metric for decentralized optimization algorithms [1,2]. Further, the scalability is not experimented/discussed in detail in the paper. In fact, I have doubts on the scalability of the algorithm, especially for dense agents, as the communication overhead becomes extremely heavy. One related work is missing [3].

[1]. Optimal and Practical Algorithms for Smooth and Strongly Convex Decentralized Optimization, NIPS, 2020
[2]. Lower Bounds and Optimal Algorithms for Smooth and Strongly Convex Decentralized Optimization Over Time-Varying Networks, NIPS, 2021
[3]. Decom: decomposed policy for constrained cooperative multi-agent reinforcement learning, AAAI, 2023


**Limitations:**

Authors discussed the future works on the scalable safe MARL algorithms with adaptive communication of agents’ state/action information and intelligent sampling of agents’ trajectories. I personally prefer to focus more on the time-varying networks [2], where the connection is unstable and the communication has some unignored latency.

---

> ### Author Rebuttal · Authors · 2023-08-09
>
> We thank the reviewer for appreciating our work. We hope our responses below provide further clarity.
> ### Questions
> > The potential scope ... distributed way.
>
> A: We would like to provide more concrete motivations for studying decentralized MARL.
> * First, while global state/action spaces are decomposable and each agent has its own utilities, the problem in Eq. (5) isn't agent-separable. Different agents' utilities are closely linked through their occupancy measures.
> * Second, as highlighted in the 2nd point of **General Response**, many real-world problems exhibit a decentralized structure with primarily local agent interactions. Such problems align with our paper's focus, allowing agents to learn effective policies using only local interactions. For more examples, we refer the reviewer to [24].
> * Lastly, we acknowledge our framework applies to problems with a special additive structure in occupancy measure space, which presumes a degree of cooperation and special statistical structure between agents' occupancy measures; however, it is still meaningful to study more general problems when these structural assumptions don't hold, as in competitive or general-sum Markov games, as well as dec-POMDPs. The proposed framework does not apply to these more general settings, although it may be possible to generalize the notion of locality to these settings. We leave this avenue for future work.
>
> > The communication protocol ... each agent.
>
> A: In this work, we assume the underlying graph $\mathcal{G}$ to be a distance measure for different agents. Under $\mathcal{G}$, the $\kappa$-hop neighborhood $\mathcal{N}_i^\kappa$ servers as the allowable ranges for both observation and communication for agent $i$. Each agent $i$ can observe the states/actions of agents in $\mathcal{N}_i^\kappa$ and communicate with them in a P2P way. We will elaborate on this setting in the revised paper.
>
> We understand that another practical scenario is limiting the data transmission to connected agents. Yet, we hope the reviewer understands that our main focus in the paper is showing that agents can jointly learn a provably efficient safe policy without global observation.
>
> > Assumption 3.2 ... state transition.
>
> A: We understand the reviewer's concern. Assumption 3.2, while seeming stringent, is milder than many prevalent assumptions in existing theoretical works. It extends the conditions set by [10,24,38,43], where $\mathbb{P}_i$ is solely determined by $s\_{N_i^1}$ and $a_i$. Also, our approach offers a notable relaxation from prior MARL studies that mandate global observability. It's important to note that many MARL studies with complete partial observability lack a convergence theory and lean on centralized training decentralized execution (CTDE).
>
> Lastly, we note that the exponential decay in Assumption 3.2 mirrors wireless mesh networks, where signal strength typically decays exponentially with distance, plus noise [37].
>
> > The algorithm has ... algorithms [R1,R2].
>
> A: We appreciate the reviewer's insightful observation and are happy to clarify the communication complexity. In our algorithm, agents coordinate with neighbors during trajectory sampling (line 3), Q-function estimation (line 6), and gradient estimation (line 8). Specifically, each agent $i$ needs to **observe** states and actions of agents in $\mathcal{N}_i^\kappa$ in lines 3 and 6, and it needs to **share** the estimated dual variable and Q-functions in line 8. If we focus on direct message sharing under P2P, the communication per iteration is $1/2\cdot\sum_i |\mathcal{N}_i^\kappa|\leq n\cdot\max_i|\mathcal{N}_i^\kappa|\leq\mathcal{O}(n^2)$. For sparse networks where $\max_i|\mathcal{N}_i^\kappa|=\mathcal{O}(1)$, it's $\mathcal{O}(n)$.
>
> Considering "observation" as part of the communication complexity or assuming only adjacent agent communication, as in [R1,R2], would indeed increase the required communications per round. However, we highlight that, compared with works assuming CTDE or global observation (where communications are implicitly required), our work still provides a strong relaxation by only requiring local communications.
>
> >Further, the scalability ... extremely heavy.
>
> A: We'd like to clarify that our experiments validate the algorithm's scalability without sacrificing optimality. In Pistonball and Wireless communication, our algorithm effectively learns policies with $\kappa= 1$ (2 neighbors for Pistonball and 4 for Wireless), despite having 20 and 25 agents in total, respectively. Moreover, it performs comparably to MAPPO-L, which demands global observation.
>
> For scenarios with densely connected agents, our method aligns with standard primal-dual policy gradient techniques and doesn't present significant advantages over centralized algorithms (we assume $\mathcal{G}$ is not densely connected in Section 2). In fact, when interactions between agents are crucial, omitting any agent's data can lead to suboptimal or unsafe policies. In such cases, unless the problem has a unique structure, some level of communication remains essential.
>
> > One related work is missing [R3].
>
> A: We thank the reviewer for the additional reference. We will include it in the revised paper.
> ### Limitations
>
> > Authors discussed ... unignored latency.
>
> A: We thank the reviewer for the insightful comments, and we kindly refer the reviewer to the last point in our **General Response** for a discussion on this extension.
> ### Reference
> [R1] Optimal and Practical Algorithms for Smooth and Strongly Convex Decentralized Optimization
>
> [R2] Lower Bounds and Optimal Algorithms for Smooth and Strongly Convex Decentralized Optimization Over Time-Varying Networks
>
> [R3] Decom: decomposed policy for constrained cooperative multi-agent reinforcement learning

---

### Official Review · Reviewer_fkzS · 2023-07-11

**Soundness:** 3 good
**Presentation:** 3 good
**Contribution:** 3 good
**Rating:** 7
**Confidence:** 3

**Summary:**

This paper considers the problem of safe/constrained multi-agent reinforcement learning (MARL) with general utilities and constraints, which could be generic functions of agents' occupancy measures. A distributed method is proposed for solving the constrained MARL problem with local constraints, where each agent can communicate to its few-hop neighbors in the underlying network graph topology. The proposed algorithm is theoretically analyzed in terms of convergence behavior and sample complexity, and numerical results are provided to demonstrate the efficacy of the proposed method in multiple constrained MARL environments.

**Strengths:**

- The problem of constrained/safe MARL is of great practical interest and importance, so the paper focuses on a timely topic of interest to the NeurIPS audience.
- Distributed training of safe RL policies in multi-agent settings is of significance, and the paper makes important contributions in this direction.
- Extensive theoretical and numerical analysis is provided to showcase the benefits of the proposed approach.

**Weaknesses:**

- The presentation and structure of the paper could be improved, especially since many of the important parts of the paper (especially the algorithm pseudo-code, flowchart, and numerical results) are deferred to the appendix. Mindful of the space limitations, it would be great if the authors could shuffle the contents in their revision to include the pseudo-code/flowchart and more numerical results in the main body of the paper, with at least one result for each environment introduced in Section 5.
- The authors mention [25] (Lu et al., AAAI'21) in their discussion of related work, but I would have liked a more extensive discussion on the similarities and differences between this work and [25], especially since they both consider an underlying graph topology, as well as numerical comparisons with [25] in addition to MAPPO-Lagrangian.
- The paper mostly focuses on tabular MARL settings with a finite state space. It is unclear how the proposed method generalizes to continuous state spaces.

**Questions:**

- I believe the Transition decomposition assumption in lines 106-108 is very strong. Could you please comment on how you justify this assumption in practical settings and whether it could be relaxed to a weaker assumption?
- How does the proposed method extend to the case where the objective in Eq. (5) is not separable?
- Intuitively, the upper bound in Eq. (13) should depend on $\omega$ and $\chi$ in Assumption 3.2, so please discuss why they do not appear here.
- In Eq. (15), could $1/n$ be replaced with $1/|\mathcal{N}_i^{\kappa}|$? I understand that this is simply a scaling factor, but truncating the policies to $\kappa$-hop neighbors could also justify the aforementioned replacement, where the Lagrangian is always *averaged* over the $\kappa$-hop neighborhood of each agent.
- Does Fig. 1-d show that the constraints are always violated? Does this mean that no feasible policy in this environment might exist?
- In the Appendix, it is shown that MAPPO-L performs worse than the proposed method. Could you provide a more in-depth discussion of why this happens? As the authors allude to, MAPPO-L has access to the global state during the centralized training phase, so it is okay (and probably expected) if the proposed method (which is fully distributed) underperforms MAPPO-L.
- Since a graph structure and distributed implementation are the underpinnings of the proposed method, how does the proposed method related to graph neural networks (GNNs)? The agents could perform message passing to their neighbors through a GNN architecture, so it would be helpful if the authors could discuss potential connections of their method to MARL methods based on GNNs, such as the following reference [A]:
[A] Jiang, Jiechuan, Chen Dun, Tiejun Huang, and Zongqing Lu. "Graph Convolutional Reinforcement Learning." In International Conference on Learning Representations. 2019.

**Limitations:**

As mentioned under **Weaknesses**, the extension to non-tabular MARL settings is unclear. Furthermore, the robustness of the proposed method to imperfect communication among agents (e.g., noise perturbations, random communication link drops, delays, time/frequency resource constraints, etc.) needs to be studied.

---

> ### Author Rebuttal · Authors · 2023-08-09
>
> We thank the reviewer for the positive feedback. We hope our responses address your concerns and provide further clarity.
> > (Weakness) The presentation ... in Section 5.
>
> > (Weakness) The paper mostly focuses ... continuous state spaces.
>
> > (Question) How does the proposed method extend to the case where the objective in Eq. (5) is not separable?
>
> > (Limitation) The robustness of the proposed method ... needs to be studied.
>
> A: We kindly refer the reviewer to our **General Response** for detailed discussions on your comments.
> ### Weaknesses
> > The authors mention [25] ... in addition to MAPPO-L.
>
> A: We are pleased to offer a more detailed comparison.
> * [25] assumes global observability, where all agents share a common state and each agent's actions and states are universally known. They use a consensus-based algorithm, where each agent maintains an estimation for the global objective and averages it with its neighbors. The averaging mechanism (consensus update) is determined by the underlying graph. Our approach is more practical: agents have local states and only observe nearby agents, with distances defined by the associated graph.
> * [25] focuses on cumulative return and doesn't address general utilities like safety or risk-sensitive objectives, which our work covers. We direct the reviewer to Appendix H.5 for more benefits of using general utilities.
>
> We'll also include an experiment comparing with [25] in our revisions.
>
> ### Questions
> > I believe the Transition ... a weaker assumption?
>
> A: We'd like to clarify:
> * The transition decomposition doesn't suggest all agents' states and actions are independent. It assumes that new local states are independently generated based on current global information. This is a common assumption in networked multi-agent studies (e.g., [10,24,38,43]) and is consistent with our experiments.
> * A potential relaxation could consider coupled local state generation for adjacent agents. Then, it is important to analyze its impact on the system's correlation decay property.
>
> > Intuitively, the upper bound ... do not appear here.
>
> A: We are sorry for the confusion. Prop 3.4 is used to justify Assumption 3.3 ($\kappa$-hop policies), and it is independent of Assumption 3.2. Eq. (13) shows that if the local policy itself enjoys the decay property, then the occupancy measure also has the same rate of decay.
>
> > In Eq. (15), could $1/n$ ... neighborhood of each agent.
>
> A: Substituting $1/n$ with $1/|\mathcal{N}_i^\kappa|$ in gradient estimation might introduce bias. Indeed, the estimation in Eq. (15) can be obtained from the exact gradient in Eq. (10) by replacing $Q$-functions with their truncated estimators, as the truncated Q-functions of agents outside $\mathcal{N}_i^\kappa$ yield 0 after taking expectation (see Eq. (44)).
>
> However, this change ultimately is equivalent to scaling the common primal step-size $\eta_\theta$ by an agent-specific factor $n/|\mathcal{N}_i^\kappa|$ (see Eq. (19)). We'll assess this alteration's impact and detail it in the revised paper.
>
> > Does Fig. 1-d show ... might exist?
>
> A: We apologize for the confusion. We think this curve can be explained by three factors.
> * We display the sum of absolute violations for all 20 agents, which amplifies the perceived violation.  The curve for a single agent is shown in Figure 1 of the rebuttal PDF.
> * By our experiment design, the objective and constraint conflict. Hence, the constraint is very likely to hold with equality at the optimal solution. Under the sample-based setting, minor constraint violations are expected.
> * Primal-dual methods typically ensure averaged iterates' feasibility (e.g., [51,60,62]), and wandering in and out of the feasible set is typical numerical behavior. To achieve strict feasibility in the last iteration, one way is to introduce pessimism by adjusting the right-hand side of the constraints [60,62].
>
> In cases with no feasible policy, constraint violations are always positive. As per Eq. (18), this results in the algorithm greatly prioritizing constraint satisfaction over objective maximization.
>
> > In the Appendix, it is ... underperforms MAPPO-L.
>
> A: We believe MAPPO-L's underperformance compared to our algorithm is due to:
> * The full-observability of MAPPO-L, which significantly enlarges state/action spaces. Though we intentionally enlarged the neural network size for MAPPO-L, it might still be under-parameterized.
> * Potential inaccuracies in Q-function approximation. The inaccurate Q-function of distant agents might serve as noises. While truncated gradients can mitigate this issue, MAPPO-L, with its full observability, could be more affected.
>
> We will also discuss this issue in the revised paper.
>
> > Since a graph structure ... such as the following reference [R].
>
> A: Thank you for the insight. In DGN [R], agents aggregate local observations via graph convolution with an attention kernel, feeding the result into their Q-functions. DGN differs from our distributed approach as it shares weights across agents and has a dynamic adjacency matrix. Our work, however, assumes a static connectivity graph (unless we suppose Assumption 3.2 always holds for the time-varying graph). This allows us to concatenate all features from neighbors together, which has proven effective in experiments. For more complex, dynamic graphs, we admit architectures like a weight-independent DGN might be advantageous. We will discuss this as future work in the revised paper.
> ### Reference
> [R] Graph Convolutional Reinforcement Learning

---

> > ### Comment · Reviewer_fkzS · 2023-08-14
> > **Thank you!**
> >
> > Thank you very much for the responses to my comments, which have resolved my concerns to a great extent. I have updated my score accordingly.

---

> > > ### Author Response · Authors · 2023-08-14
> > >
> > > Thank you very much for appreciating our work and raising the score!

---

### Author Rebuttal · Authors · 2023-08-08

## General Response
We would like to express our sincere gratitude to the reviewers for reading our paper and providing valuable feedback. Below, we answer some common questions of the reviewers, including the **paper organization**, the **separability of the objective**, the **extension to continuous space**, and the **robustness of the algorithm**. Please find our responses to other questions in the personalized rebuttals.

**Remark: Without further specification, we use "[number]" to refer to the corresponding reference in our paper.**

> Paper organization (Reviewers fkzS, pEsM, bKHc).

A: As suggested by many of our reviewers, we will try to shift the contents in the revised paper, and present (at least some part of) the related work, algorithm pseudo-code, and numerical experiments/baseline comparisons in the main body of the paper. To make room for these important contents, we plan to defer the auxiliary results such as Proposition 3.4 and Lemma 3.5 (as well as the associated discussions) to the appendix. Moreover, we will try to introduce the problem formulation and technical assumptions with fewer words, with complementary discussions placed in the appendix. We believe this reorganization will significantly improve the presentation of the paper.

> The separability/additivity of the objective function in Eq. (5) (Reviewers fkzS, bKHc).

A: In our work, we assume the global objective is the average of agents' local objectives (possibly nonlinear functions of their local occupancy measures). While this formulation exhibits limitations, we want to justify it from four perspectives.
* First, assuming a separable objective is one of the basic motivations for our focus on distributed training without global observability. Without separability, partial observability may preclude the possibility of both decentralized training and even in the case of centralization, would limit the ability to ensure convergence without special conditions on the statistical dependence of how agents' states and actions are related.
* Second, under the separable setup, each agent can privately evaluate both its objective and constraint and has the choice of sharing them with its neighbors or not. In addition, we want to highlight that, despite the separability in the expression, the local utilities $f_i(\lambda_i^{\pi_\theta})$ are still coupled among agents. This is because the local occupancy measure $\lambda_i^{\pi_\theta}$ of each agent $i$ depends on all other agents' policies.
* Third, this separable objective still incentivizes cooperation among agents, ensuring that learning is collaborative.  Without this property, the resulting problem often could not be considered a MDP. Instead, it would be more properly defined as a Markov game. This is a significant research area unto itself.
* Lastly, this formulation covers a wide range of real-world applications. For example, temperature control in smart building systems [R1] often requires decentralization due to its good scalability and reduced communication burdens. Decentralized training is also critical for disease control, especially in underdeveloped areas where local officials need to make decisions based on limited local information [R2]. Other examples include wireless communication networks [R3], queueing networks [R4], and smart transportation [R5]. We refer to [24] for more applications. By incorporating general utilities and safety constraints, our formulation can adapt to many practical scenarios.

Given the above reasons, we believe the separable objective studied in this work is also a practical formulation and has its own value. In the meanwhile, we acknowledge that the extension of our work to the non-separable objective will be an exciting and meaningful future research.

> Generalization to Continuous Space (Reviewers fkzS, bKHc).

A: Since our algorithm allows policy parameterization, a natural approach to deal with the continuous space is to discretize it to a finite discrete space. As a drawback, there will be an additional error term arising from the discretization.

When discretization is inappropriate, the bottleneck of our algorithm lies in the estimation of occupancy measures in continuous space. Empirically, this can be done through kernel density estimation [R6] or Bayesian neural network [R7]. Yet, these approaches can be computationally inefficient, and it is beyond the scope of this work to analyze how the bias of such method figures into our convergence analysis.

> Robustness of the algorithm and time-varying networks (Reviewers fkzS and Vbe7).

A: We thank the reviewer for the suggestions. Our current analysis assumes perfect communication among agents, and it can be generalized to simple time-varying networks, provided that Assumption 3.2 always holds. The main message we want to convey is that even without global observation, the agents can learn a provably efficient policy for safe RL problems with general utilities. Yet, we admit that accounting for the data latency and considering robustness in communications are very meaningful future works.

For unstable networks, our current experiments suggest that dropped/perturbed communication may bring little influence to distant agents (since $\kappa=1$ already performs well). We are running new experiments to test the impact of dropped/perturbed neighboring Q values at step 8 of Algorithm 1 and will update the results in the PDF.
### Reference
[R1] Decentralized and distributed temperature control via HVAC systems in energy efficient buildings.

[R2] Health care reform, decentralization, prevention and control of vector-borne diseases.

[R3] Temporal starvation in multi-channel CSMA networks: an analytical framework.

[R4] The complexity of optimal queuing network control.

[R5] Control of robotic mobility-on-demand systems: a queueing-theoretical perspective.

[R6] Deconvolving kernel density estimators.

[R7] Estimating continuous distributions in Bayesian classifiers.

---

### Decision · Program_Chairs · 2023-09-21

**Decision:**

Accept (poster)

**Comment:**

**Summary:**
This paper explores the issue of safe and constrained multi-agent reinforcement learning (MARL), focusing on general utility and constraint functions based on agents' occupancy measures. The authors present a decentralized solution that accommodates local constraints and allows for limited communication between each agent and its immediate neighbors in the network graph. The algorithm's convergence and sample complexity are theoretically examined, and its effectiveness is empirically validated through numerical tests in various constrained MARL settings.

**Reasons for Acceptance:**
The reviewers unanimously agree that the paper is well-written, the theory is sound, and the experiments are adequate. The reviewers have raised a number of technical concerns, the authors have addressed them satisfactorily in the rebuttal phase.